# A human electrophysiological signature of Fragile X pathophysiology is shared in V1 of *Fmr1-/y* mice

Sara S. Kornfeld-Sylla [1,2] ✉, Cigdem Gelegen[3], Jordan E. Norris [4,5], Francesca A. Chaloner [3,6], Maia Lee[1,2], Michael Khela [7], Maxwell J. Heinrich[1,2], Peter S. B. Finnie[1,2,8], Lauren E. Ethridge[4,9], Craig A. Erickson[10,11], Lauren M. Schmitt[12], Sam F. Cooke[3,6], Carol L. Wilkinson [13,14] & Mark F. Bear [1,2] ✉

Predicting clinical therapeutic outcomes from animal studies using conserved electrophysiological phenotypes could facilitate developing treatments for neuropsychiatric disorders. Alpha oscillations in human resting-state electroencephalogram recordings are altered in many disorders, but whether these disruptions exist in mouse models is unknown. Here, we employed a uniform analytical method to show in males with fragile X syndrome (FXS) that alpha oscillations in humans and alpha-like oscillations in the visual cortex of *Fmr1-/y* mice are slowed, with a stronger phenotype in adults than juveniles and a juvenile-specific power phenotype in both species. We find that alpha-like oscillations are disrupted by deletion of *Fmr1* in cortical excitatory neurons and glia, reflect differential activity of two classes of GABAergic interneurons, and are more sensitive to activation of GABA_B receptors by Arbaclofen in wild-type than *Fmr1-/y* mice. Our framework reveals evolutionary conservation of alpha oscillation disruptions, enables a deeper understanding of FXS pathophysiology, and narrows the gap between treatment promise and practice.

Fragile X syndrome (FXS) is the leading inherited cause of intellectual disability (ID) and autism spectrum disorder (ASD)[1,2]. In most cases, FXS is caused by transcriptional silencing of the *FMR1* gene and loss of the protein product, fragile X messenger ribonucleoprotein (FMRP)[3–5]. Because it is monogenic, genetically engineered animal models of FXS are available for preclinical research[5–7]. Studies in *Fmr1-/y* mice have revealed key aspects of neural pathophysiology and provided an opportunity to test candidate therapeutics[8–10]. To date, however, the molecules nominated to treat FXS have failed to meet prospectively defined clinical trial endpoints in humans[11]. A mismatch between preclinical promise and clinical outcomes is certainly not unique to FXS[11,12]. Given the wide differences in animal and human behavior, a bridge between the preclinical and clinical domains that measures shared alterations in

[1]The Picower Institute for Learning and Memory, Massachusetts Institute of Technology, Cambridge, MA, USA. [2]Department of Brain and Cognitive Sciences, Massachusetts Institute of Technology, Cambridge, MA, USA. [3]Wolfson Sensory, Pain and Regeneration Centre, Institute of Psychiatry, Psychology and Neuroscience, King's College London, London, UK. [4]Department of Psychology, University of Oklahoma, Norman, OK, USA. [5]Department of Pediatrics, Rush University Medical Center, Chicago, IL, USA. [6]MRC Centre for Neurodevelopmental Disorders, King's College London, London, UK. [7]Tufts University, Boston, MA, USA. [8]Tanz Centre for Research in Neurodegenerative Diseases, University of Toronto, Toronto, ON, Canada. [9]Department of Pediatrics, Section on Developmental and Behavioral Pediatrics, University of Oklahoma Health Sciences Center, Oklahoma City, OK, USA. [10]Division of Child and Adolescent Psychiatry, Cincinnati Children's Hospital Medical Center, Cincinnati, OH, USA. [11]Department of Psychiatry and Behavioral Neuroscience, University of Cincinnati College of Medicine, Cincinnati, OH, USA. [12]Phelan-McDermid Syndrome Foundation, Osprey, FL, USA. [13]Division of Developmental Medicine, Boston Children's Hospital, Boston, MA, USA. [14]Harvard Medical School, Boston, MA, USA. ✉e-mail: saraks@mit.edu; mbear@mit.edu

circuit function is needed to better predict clinical trial outcomes[11–14].

Conserved electrophysiological phenotypes captured through field potential recordings of brain activity could be this bridge, as they are quantitative measures of pathophysiology and therapeutic target engagement that can be readily measured and consistently analyzed across species, a foundational requirement for translational biomarkers[14–17]. In children and adults with FXS, and in *Fmr1*[-/y] rodent models, resting-state electroencephalogram (rsEEG) recordings reveal elevated power in the gamma (30–60 Hz) frequency band (gamma phenotype)[18–29]. Adults with FXS also have slower alpha (8–13 Hz) oscillations[18,19,21,30] and, critically, alpha band disruptions demonstrate clinical correlations[19] and are more reliable across individuals and testing sessions than gamma power disruptions[31–33]. Although less well characterized, there are also reports of slower background activity (i.e., alpha oscillations) in children with FXS[34,35]. However, this alpha phenotype has yet to be characterized in genetically engineered rodent models of FXS.

A more reliable, lower frequency translational measure of FXS pathophysiology and treatment response would be practically important given the challenges in isolating higher frequency signals from human EEG, particularly in infants. Moreover, since alpha oscillations regulate sensory processing[36,37] and are the most prominent rsEEG signal in humans, their disruption also fundamentally contributes to the pathology of ID[38,39], ASD[40–42], schizophrenia[43–46], and other neuropsychiatric disorders[47,48]. However, as is the case for FXS, these disruptions have not yet been identified in preclinical models of any of these disorders. Thus, identifying the FXS alpha phenotype in mice could provide a framework for future study of the mechanistic bases of these alpha oscillation disruptions and perhaps give insight into the pathophysiology of other conditions if the shared phenotype is a broader marker of neural dysfunction.

In the current study, we characterized the cross-sectional developmental trajectory of rsEEG phenotypes in male humans with FXS using spectral analysis methods that forgo traditional frequency band delineations. This approach facilitated the study of oscillations across development, and, crucially, enabled us to identify and characterize a correlate of the alpha phenotype in primary visual cortex (V1) of male *Fmr1*[-/y] mice. Intracortical local field potential (LFP) recordings in layer (L) 4 of V1 of juvenile and adult mice revealed similar age-related changes in the alpha phenotype as for humans with FXS, a phenotype in the power of the alpha-like oscillations specific to juveniles in both species, and additional alterations in the power and temporal dynamics of the oscillations not detectable from the surface. Using mouse genetic tools, we found that knocking out FMRP in cortical excitatory neurons and glia was sufficient for all alpha-like oscillation disruptions, including impairing their coupling to higher frequency oscillations coordinated by gamma-aminobutyric acid (GABAergic) interneurons. We discovered that the power of alpha-like oscillations reflects differential activity of two genetically defined classes of cortical GABAergic interneurons, parvalbumin-positive (PV+) and somatostatin-positive (SOM+) cells. Moreover, we found that the power and center frequency of alpha-like oscillations (but not gamma oscillations) were more sensitive to enhancing GABAergic inhibition through administration of the GABA_B receptor agonist Arbaclofen in wild-type than *Fmr1*[-/y] mice. These results clarify the relationship between alpha oscillations and GABAergic inhibition, inform the cellular basis for the FXS alpha phenotype, and guide future therapeutic approaches for FXS. Furthermore, these findings bridge the preclinical and clinical worlds by showing for the first time how alpha oscillation disruptions can be studied in preclinical models and used to objectively measure pathophysiology and treatment response to better understand therapeutic outcomes.

## Results

### Oscillatory disruptions, including the alpha phenotype, are not identical in children and adults with FXS

To characterize oscillatory phenotypes cross-sectionally in children and adults, rsEEG was collected using 128−channel caps in male subjects with FXS and age-matched typically developing (TD) controls across two laboratories, one which recorded from children ages 2−7 and the other which recorded from adults ages 19−44 (Fig. 1a, k). We sampled from the same electrodes across the cortex for all subjects to generate an absolute power spectrum per group (FXS and TD) within each age range (Fig. 1b, l). Across all ages, we observed significantly elevated absolute power for FXS subjects below 8 Hz and above 23.5 Hz (non-parametric bootstrapping with 95% confidence interval, $n = 17$ per group in children, $n = 20$ per group in adults).

To further analyze these spectra, we improved on existing methodology to decompose a power spectrum into aperiodic (underlying 1/f slope) and periodic (oscillatory) components (Supplementary Fig. 1a, b)[49,50]. Isolating periodic spectra improved comparisons across ages over calculating absolute or relative power in different frequency bands, as the alpha rhythm sits in the theta band (3−9 Hz) in children less than 7 years old[50–52]. We found elevated aperiodic power in FXS subjects across age groups, accounting for the increased power in the absolute spectrum (Fig. 1c, m). Aperiodic offset (the power at 3 Hz, the edge of the fitting range) was significantly elevated in children and adults with FXS, while aperiodic slope was not significantly different between groups in either age bin (Fig. 1d, e, n, o). FXS aperiodic disruptions were consistent across age groups.

Removing the aperiodic component from the absolute spectra revealed that oscillatory (periodic) differences between FXS and TD subjects were not the same for the two age groups (Fig. 1f, p). We found two periodic peaks across all groups and ages: a narrow peak one (Pk1, 4−14 Hz) that was differentially altered in children and adults with FXS, and a broad, more variable peak two (Pk2, 15−40 Hz) that was strikingly aberrant only in children with FXS. Accordingly, the FXS gamma phenotype was due to elevated aperiodic power in adults, but due to elevated aperiodic and periodic (oscillatory) power in children.

We quantified the maximum power and center frequency of each peak for each subject (Supplementary Fig. 1c, h). The most pronounced EEG phenotype in children with FXS was an increase in the power and center frequency of Pk2 (Fig. 1g, Supplementary Fig. 1d, f). The center frequency of Pk2 in children with FXS younger than 60 months old clustered around the median (31.3 Hz), while the distributions for all TD and older FXS children were more variable (Fig. 1g, Supplementary Fig. 1e)[50]. There was no phenotype in periodic Pk2 in adults, although adult FXS subjects displayed a reduced correlation between the maximum power of Pk1 and Pk2 relative to TD subjects (Fig. 1q, Supplementary Fig. 1i-j).

As expected, the most pronounced phenotype in adults with FXS was a reduction in center frequency of Pk1 (median 8 Hz in FXS vs. 10.5 Hz in TD adults, Fig. 1r). Children with FXS had a modest reduction in center frequency (median 8.3 Hz in FXS vs 9 Hz in TD children; Fig. 1h), as well as a reduced maximum power of Pk1 relative to TD children (Fig. 1i). FXS children with the lowest Pk1 center frequency values also had the lowest Pk1 power (Fig. 1j), meaning alpha oscillations (periodic Pk1) were slower and weaker in FXS children. Unlike with Pk2, this Pk1 power phenotype persisted across all ages tested in FXS children (Supplementary Fig. 1f, g). However, Pk1 maximum power was not significantly different between FXS and TD adult subjects (Fig. 1s, Supplementary Fig. 1k), meaning alpha oscillations were much slower but not weaker in FXS adults. Therefore, the shift in Pk1 center frequency (the alpha phenotype) was the only oscillatory phenotype present across all ages, and was stronger in adults, while the Pk1 power phenotype was juvenile-specific.

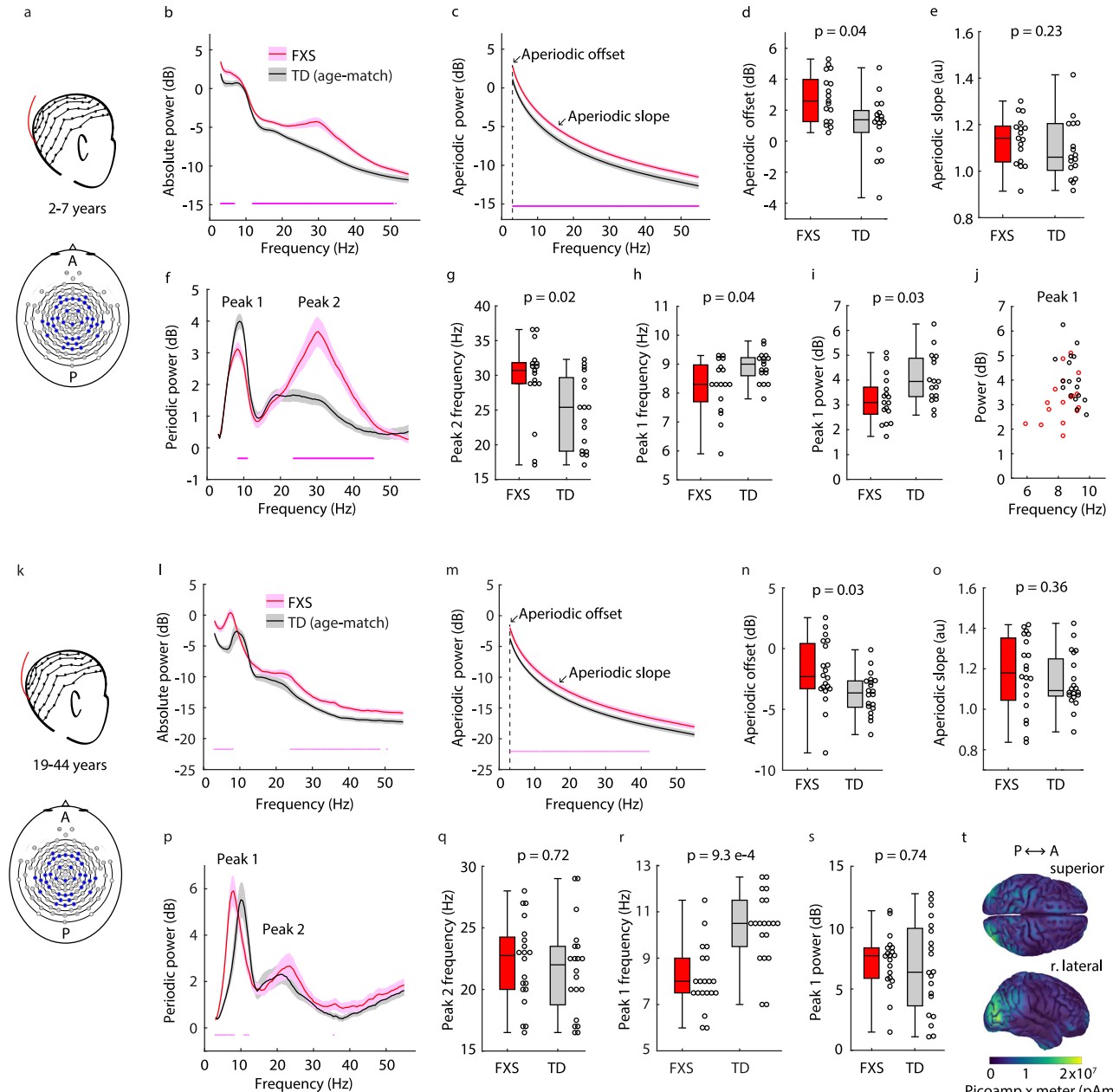

**Fig. 1 | Cross-sectional developmental trajectories of rsEEG phenotypes of FXS.**
**a** Experimental design. rsEEG data were collected from frontal, temporal, parietal, and occipital electrodes in FXS and TD subjects, ages 2–7 years (*n* = 17 per group). A/P = anterior/posterior. **b** Absolute power spectrum (mean +/− SEM). **c** Aperiodic fit (mean +/− SEM). **d** Boxplot and FDR-corrected *p*-value for aperiodic offset (power at 3 Hz, the edge of the fitting range). Uncorrected *p*-value = 0.019, *z*-statistic = 2.342, effect size = 0.474. **e** Boxplot and FDR-corrected *p*-value for aperiodic slope. Z-statistic = 1.206, effect size = 0.246. **f** Periodic power (mean +/− SEM) reveals two oscillatory peaks (Pk1, Pk2). FXS periodic power is significantly reduced at 8.3–11 Hz and elevated at 23.6–45.3 Hz. **g–i** Boxplots and FDR-corrected *p*-values for: **g** Center frequency of Pk2. Uncorrected *p*-value = 0.009, *z*-statistic = 2.621, effect size = 0.529; **h** Center frequency of Pk1. Z-statistic = −2.017, effect size = −0.405; and **i** Maximum power of Pk1. Uncorrected *p*-value = 0.023, *z*-statistic = −2.273, effect size = −0.46. **j** Pk1 maximum power as a function of center

frequency for FXS (red circles) and TD (black circles). **k–s** Like **a–i** but for adult FXS and TD subjects, ages 19–44 years (*n* = 20 per group). For **n**, uncorrected *p*-value = 0.014, *z*-statistic = 2.448, effect size = 0.455. For **o**, *z*-statistic = 0.906, effect size = 0.17. For **p**, FXS periodic power is significantly elevated 3–8 Hz and reduced 11–12 Hz. For **q**, uncorrected *p*-value = 0.54, *z*-statistic = 0.611, effect size = 0.115. For **r**, uncorrected *p*-value = 2.33e−4, *z*-statistic = −3.681, effect size = −0.678. For (**s**), *z*-statistic = 0.338, effect size = 0.065. **t** Source localization of mean absolute 4–14 Hz power across adult TD subjects. Superior and right (r.) lateral views. Dots at the bottom of plots in (**b**, **c**, **f**, **l**, **m**, **p**) indicate the points of significant difference between groups (non-parametric hierarchical bootstrap, 95% confidence interval). Boxplots show 25th, median, and 75th percentiles, with whiskers extending to minimum and maximum values. Uncorrected *p*-values and *z*-statistics calculated from two-sample, two-sided Wilcoxon Rank-Sum Tests and effect sizes calculated using Cliff's Delta. *P*-values adjusted with the Benjamini-Hochberg correction.

Although the alpha phenotype in FXS children and adults was detectable across all cortical regions (Supplementary Figs. 2, 3), we found that the source of absolute 4–14 Hz power (i.e., the range of periodic Pk1) localized to occipital cortex in adult TD subjects (Fig. 1t). Although enlarged head size made

source localization more difficult to estimate in FXS subjects, we also observed a posterior origin of 4–14 Hz power (Supplementary Fig. 1l). If a correlate of the alpha phenotype exists in rodent models, we expected it might be easiest to detect in visual cortex.

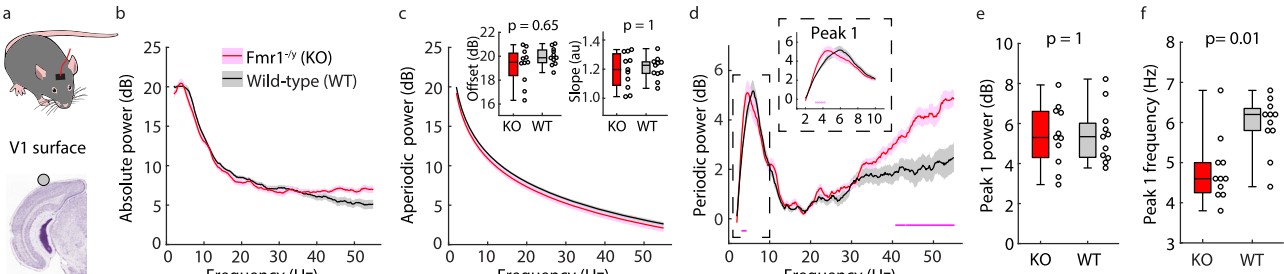

**Fig. 2 | A correlate of the alpha phenotype is present in V1 of adult *Fmr1⁻/y* mice.**
**a** Experimental design. EEG data were collected from an electrode on the cortical surface of V1 of freely-moving adult *Fmr1⁻/y* (KO) and littermate WT mice (p70-115, *n* = 11 per group). 100 seconds of resting-state data while the mice were awake but stationary were analyzed. Nissl from the Allen Reference Atlas – Mouse Brain[119], https://atlas.brain-map.org/. **b** Absolute power spectrum (mean +/− SEM). **c** Main: Aperiodic fit (mean +/− SEM). Inset: Boxplots and FDR-corrected *p*-values for aperiodic offset (power at 2 Hz, the edge of the fitting range) and aperiodic slope. Offset uncorrected *p*-value = 0.33, z-statistic = −0.985, effect size = −0.256; slope z-statistic = 0, effect size = −0.008. **d** Periodic power (mean +/− SEM). Dots at the bottom of the plot indicate the points of significant difference between groups

(non-parametric hierarchical bootstrap, 99% confidence interval). Power in KO mice is significantly elevated at 3.2–4.2 Hz and 41.2–55 Hz. The inset shows a zoom on periodic peak 1 (Pk1). **e–f** Boxplot and FDR-corrected *p*-values for: **e** the maximum power of Pk1. Z-statistic = 0, effect size = 0.008; and **f** the center frequency of Pk1. Uncorrected *p*-value = 0.007, z-statistic = −2.705, effect size = −0.686. Boxplots show 25th, median, and 75th percentiles, with whiskers extending to minimum and maximum values. Uncorrected *p*-values and z-statistics calculated from two-sample, two-sided Wilcoxon Rank-Sum Tests and effect sizes calculated using Cliff's Delta. *P*-values adjusted with the Benjamini-Hochberg correction. Source data are provided as a Source Data file.

## A correlate of the alpha phenotype is present in V1 of adult *Fmr1⁻/y* mice

Based on these human findings, we used our analyses to investigate if mice also have a periodic Pk1, and if so, whether its center frequency is reduced in V1 of *Fmr1⁻/y* mice. Adult *Fmr1⁻/y* and littermate wild-type (WT) mice (postnatal day (p)70–115, *n* = 11 per group) were implanted with screw-type electrodes in the skull over V1 and primary somatosensory cortex (S1) (Fig. 2a, Supplementary Fig. 4a). One week later, the awake, freely moving mice were sequentially placed in a video-monitored chamber while EEG from the cortical surface (electrocorticography data) were continuously recorded. Behavioral scoring using video recordings facilitated extraction of 100 seconds of resting-state (i.e., awake but stationary) data from each animal.

We observed significantly elevated absolute power from the S1 electrode in *Fmr1⁻/y* mice between 45.6–55 Hz (non-parametric bootstrapping with 99% confidence interval) and a similar trend in V1 (Fig. 2b, Supplementary Fig. 4b), replicating reports of elevated gamma power in *Fmr1⁻/y* mice[23–25,28,29]. We next analyzed the aperiodic and periodic components. Unlike in humans, there was no difference in aperiodic power or offset between the two groups (Fig. 2c, Supplementary Fig. 4c). Instead, both genotypes had a broad increase in periodic power at high frequencies which was significantly larger in *Fmr1⁻/y* mice (Fig. 2d, Supplementary Fig. 4d). There was a robust periodic Pk1 between 2–13 Hz in both genotypes and no significant difference between genotypes in the maximum power of Pk1 in either V1 or S1 (Fig. 2e, Supplementary Fig. 4e). Using the electrode over V1, we found a significant reduction in the center frequency of Pk1 in *Fmr1⁻/y* mice (median 4.6 Hz) relative to WT mice (median 6.2 Hz; Fig. 2f). Pk1 center frequency in S1 of *Fmr1⁻/y* mice had a broader interquartile range, so no significant difference between genotypes pulled out (Supplementary Fig. 4f) and the disruption was indeed easiest to detect in murine visual cortex. This shift in center frequency of Pk1 in V1 of adult *Fmr1⁻/y* mice was consistent with the shift in Pk1 seen in adult humans with FXS. We therefore identified a correlate of the alpha phenotype in mice.

## Intracortical recordings in V1 recapitulate the alpha phenotype in adult mice and reveal a power phenotype within a two-subpeak structure to periodic Pk1

Our discovery of the Pk1 phenotype in *Fmr1⁻/y* mice supports work positing a commonality between theta oscillations in V1 of mice and human alpha oscillations[53,54]. 3–6 Hz alpha-like oscillations in mouse V1

are present in all cortical layers but strongest in L4[53]. Therefore, we further characterized the alpha phenotype by implanting adult *Fmr1⁻/y* and littermate WT mice (p70–150, *n* = 22 per group) with LFP microelectrodes targeted to L4 of V1. After recovery, the mice were habituated to the recording setup (head-fixation in front of a monitor displaying a static, iso-luminant gray screen; Fig. 3a inset). Head-fixation permitted control of visual input and limited motion artifacts in high frequencies of the LFP. After two days of habituation, we characterized 150 seconds of data while the mice passively viewed the same, static gray screen on the monitor (i.e., resting-state data).

Only modest genotype differences pulled out of the absolute spectrum (significant difference 4.6–5.2 Hz, non-parametric bootstrapping with 99% confidence interval; Fig. 3a). Of note, the gray screen stimulus induced a narrowband periodic signal around 60 Hz in both genotypes[55]; this intracortical luminance signal was not present when the monitor was turned off and the animals were in the dark (Supplementary Fig. 5a). Consistent with the EEG recordings, we found no difference in the aperiodic fit between genotypes (Fig. 3b) even though we used an additional knee parameter to accommodate the large fitting range (Supplementary Fig. 5b–d). Removing the aperiodic component from the absolute spectra revealed the full broad shape of the periodic high-frequency signal in both genotypes (with the narrowband luminance signal sitting atop) and a significant elevation in this periodic power in *Fmr1⁻/y* mice between 21.2–127.6 Hz (Fig. 3c). Intriguingly, analyzing with a higher spectral resolution revealed that periodic Pk1 (2–10 Hz) was composed of two sub-peaks (Pk1a and Pk1b). There was a significant difference in periodic power between genotypes from 3.8–5.4 Hz, within Pk1a (Fig. 3c). Both the center frequency and maximum power of Pk1a (but not Pk1b) were reduced in adult *Fmr1⁻/y* mice (Fig. 3d–g). The alpha phenotype was contained within Pk1a and was accompanied by a power reduction that was not detectable from the surface.

## Differences in periodic Pk1 phenotypes between juvenile and adult *Fmr1⁻/y* mice are consistent with age-related differences observed in humans

After characterizing the alpha phenotype in adult mice with intracortical recordings, we next analyzed 150 seconds of L4 LFP resting-state data from head-fixed juvenile *Fmr1⁻/y* and WT littermate mice (p30–40, *n* = 67 per group) passively viewing the same gray screen to test if, like humans, oscillatory phenotypes differed across age groups. We observed a significant reduction in absolute power in juvenile *Fmr1⁻/y*

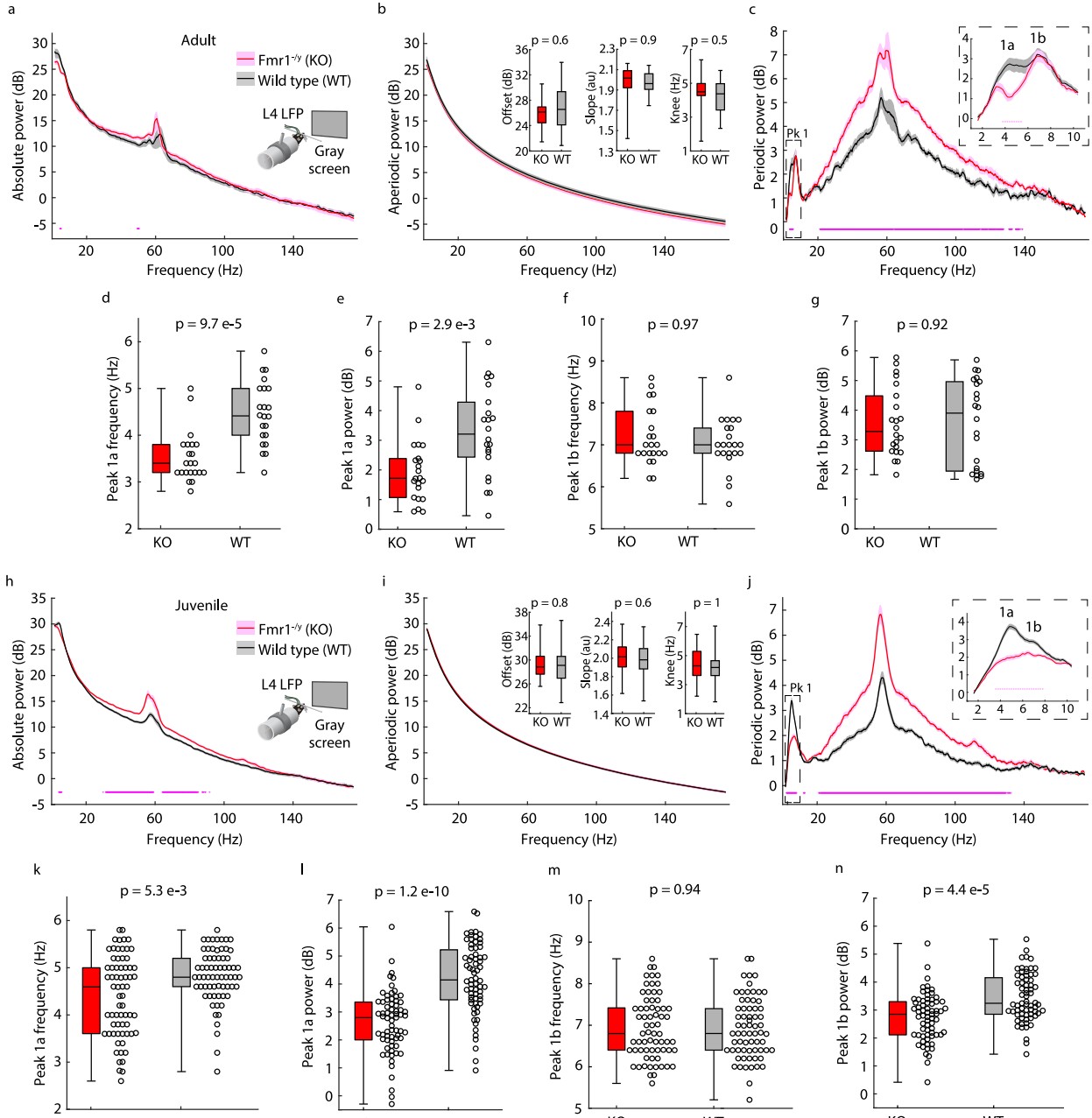

**Fig. 3 | Intracortical recordings in *Fmr1⁻/y* mice recapitulate human age-related changes in periodic Pk1 phenotypes. a** Inset: Experimental design. V1 L4 LFP data were collected in head-fixed adult *Fmr1⁻/y* (KO) and littermate WT mice (p70–150, *n* = 22 per group) passively viewing a static, iso-luminant gray screen. Main: Absolute power spectrum (mean +/− SEM) from 150 seconds of resting-state data. **b** Main: Aperiodic fit (mean +/− SEM). Inset: Boxplot and FDR-corrected *p*-values for aperiodic offset (uncorrected *p*-value = 0.379, z-statistic =−0.88, effect size = −0.157), slope (uncorrected *p*-value = 0.307, z-statistic = 1.02, effect size = 0.182), and knee frequency (z-statistic = 0.669, effect size = 0.12). **c** Main: Periodic power (mean +/− SEM). Inset: Higher spectral resolution reveals peak 1 (Pk1) has two sub-peaks (1a, 1b). **d–g** Boxplot and FDR-corrected *p*-values for: **d** Pk1a center frequency. Uncorrected *p*-value = 2.41e−5, z-statistic = −4.223, effect size = −0.742; **e** Pk1a maximum power. Uncorrected *p*-value = 0.002, z-statistic = −3.181, effect size = −0.562; **f** Pk1b center frequency. Uncorrected *p*-value = 0.731, z-statistic = 0.344, effect size = 0.062; and **g** Pk1b maximum power. Z-statistic = 0.106, effect

size = 0.021. **h–n** Like **a–g** but for juvenile KO and littermate WT mice (p30−40, *n* = 67 per group). For **i**, offset z-statistic = 0.196, effect size = 0.02; slope uncorrected *p*-value = 0.381, z-statistic = 0.877, effect size = 0.088; and knee frequency uncorrected *p*-value = 0.341, z-statistic = 0.952, effect size = 0.096. For **k**, uncorrected *p*-value = 0.004, z-statistic = −2.878, effect size = −0.288. For **l**, uncorrected *p*−value = 2.9e−11, z-statistic = −6.653, effect size = −0.666. For **m**, z-statistic = 0.076, effect size = 0.008. For **n**, uncorrected *p*-value = 2.18e−5, z-statistic −4.245, effect size = −0.425. Dots at the bottom of plots in (**a**, **c**, **h**, **j**) indicate the points of significant difference between groups (non-parametric hierarchical bootstrap, 99% confidence interval). Boxplots show 25th, median, and 75th percentiles, with whiskers extending to minimum and maximum values. Uncorrected *p*-values and z-statistics calculated from two-sample, two-sided Wilcoxon Rank-Sum Tests and effect sizes calculated using Cliff's Delta. *P*-values adjusted with the Benjamini-Hochberg correction. Source data are provided as a Source Data file.

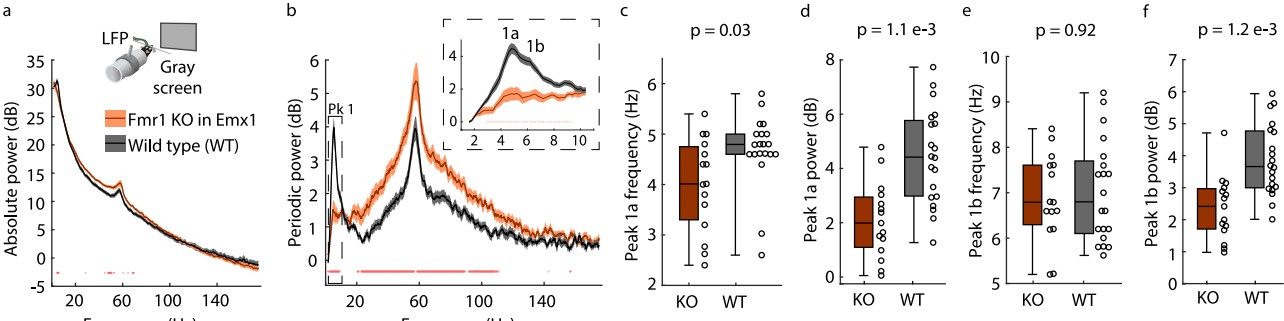

**Fig. 4 | Loss of FMRP in cortical excitatory neurons and glia is sufficient for the alpha phenotype in L4 of V1. a Inset:** Experimental design. V1 L4 LFP data were collected from juvenile (p30–40) mice with *Fmr1* specifically knocked out of cortical excitatory neurons and glia expressing the Emx1 promotor (Emx1-*Fmr1* KO mice; Cre+/Fmr1-, *n* = 15) and WT littermates (Cre/WT mice; Cre+/Fmr1+, *n* = 10 and WT/KO mice; Cre-/Fmr1-, *n* = 10). Mice were head-fixed in front of a monitor displaying a static, iso-luminant gray screen and habituated to head-fixation for two days before data collection. Main: Absolute power spectrum (mean +/− SEM). **b** Main: Periodic power spectrum (mean +/− SEM). Inset: Peak 1 (Pk1) at a higher spectral resolution. **c–f** Boxplot and FDR-corrected *p*-values for: **c** Pk1a center frequency. Uncorrected *p*-value = 0.0229, z-statistic = −2.275, effect size = −0.457; **d** Pk1a maximum power. Uncorrected *p*-value = 5.606e−4, z-statistic = −3.45, effect size = −0.693; **e** Pk1b center frequency. Z-statistic = −0.1, effect size = −0.023; and **f** Pk1b maximum power. Uncorrected *p*-value = 2.984e−4, z-statistic = −3.617, effect size = −0.727. Dots at the bottom of plots in (**a**, **b**) indicate the points of significant difference between groups (non-parametric hierarchical bootstrap, 99% confidence interval). Boxplots show 25th, median, and 75th percentiles, with whiskers extending to minimum and maximum values. Uncorrected p-values and z-statistics calculated from two-sample, two-sided Wilcoxon Rank-Sum Tests and effect sizes calculated using Cliff's Delta. *P*-values adjusted with the Benjamini-Hochberg correction. Source data are provided as a Source Data file.

mice relative to WT between 4.2–6 Hz and a significant elevation between 29–61 Hz and 63.6–85 Hz (Fig. 3h), and, as with adult mice, no difference in the aperiodic fit between genotypes (Fig. 3i). Subtracting out the aperiodic component revealed the same elevation in the broad high frequency periodic signal seen in adult *Fmr1-/y* mice (significant genotype difference between 21–132.8 Hz for juveniles; Fig. 3j). Additionally, the same two-subpeak structure was observed for periodic Pk1, but for juvenile *Fmr1-/y* mice, power was significantly reduced between 3.6–7.8 Hz and there was a significant genotype difference in the maximum power of both Pk1a and Pk1b (Fig. 3j, l, n). In adult mice, only Pk1a was different, so the power difference in Pk1b was specific to juvenile mice, just like the Pk1 power difference was specific to human children. Furthermore, the center frequency of Pk1a in juvenile *Fmr1-/y* mice was reduced relative to WT littermates (Fig. 3k), but the phenotype was much less strong than in adult *Fmr1-/y* mice. Excitingly, this recapitulated the difference in the alpha phenotype seen between human children and adults.

To ensure that our cross-sectional approach to studying developmental trajectories was not misleading, we ran a longitudinal study in a subset of mice (*n* = 7–8 per group). We analyzed 120 seconds of resting-state, head-fixed LFP data in the same cohort of *Fmr1-/y* and WT littermate mice viewing the static gray screen at postnatal 4 and 11 weeks and replicated the developmental changes in periodic Pk1 phenotypes. At ~p30, there was a significant difference in Pk1 between 2.4–6.2 Hz and 7.2–9 Hz (Pk1a and Pk1b), while at ~p80 the difference was between 3.8–5 Hz (Pk1a, with a clear shift in center frequency in *Fmr1-/y* mice; Supplementary Fig. 6a). Pk1b maximum power increased significantly over development in *Fmr1-/y* but not WT mice (Supplementary Fig. 6b). Pk1a center frequency decreased over development in both genotypes, but more significantly in *Fmr1-/y* than WT mice (Supplementary Fig. 6c), which matched our cross-sectional findings (Supplementary Fig. 6d). This validated our cross-sectional approaches and our reports of a juvenile-specific power phenotype and a stronger alpha phenotype in adults than juveniles in both species.

**In mice, the alpha phenotype is still present in the absence of luminance, but the juvenile-specific power phenotype is not**

In WT mice, alpha-like oscillations are strongly driven by a flash of light[53], so we investigated if genotype differences in the resting-state LFP persisted in the absence of luminance. In a cohort of *n* = 44 mice

per group (p30–150), we characterized 150 seconds of head-fixed LFP data while the monitor was turned off (i.e., resting-state data while the mice were in the dark). As before, we observed significantly elevated periodic power between 22–133.4 Hz and a significant reduction in Pk1a center frequency (i.e., the alpha phenotype) in *Fmr1-/y* mice (Supplementary Fig. 5e, f). There was also a modest reduction in Pk1a power, but not Pk1b power (Supplementary Fig. 5g). Even in juvenile mice (*n* = 38 per group) there was no significant difference in Pk1b power, unlike during the gray screen stimulus in the same mice (Supplementary Fig. 5h). Thus, the developmental differences in Pk1b power were specific to luminance conditions. Across all animals, luminance increased aperiodic power below 60 Hz and significantly increased aperiodic offset and slope (Supplementary Fig. 5i). In the juvenile mice, luminance enhanced the maximum power of Pk1a in WT, but not in *Fmr1-/y* mice, and it suppressed the maximum power of Pk1b in *Fmr1-/y*, but not WT mice (Supplementary Fig. 5j), thereby exacerbating genotype power differences. Consistent with the age-dependence of the Pk1b power phenotype, luminance did not have the same effect on Pk1b in adult *Fmr1-/y* mice (Supplementary Fig. 5k).

**Altered temporal dynamics of alpha-like oscillations in *Fmr1-/y* mice**

In humans, alpha oscillations occur in bursts and burst dynamics are altered in FXS[56]. To test if dynamics are altered in *Fmr1-/y* mice, we bandpass-filtered 100 seconds of continuous resting-state LFP data between 2–10 Hz (periodic Pk1) and used thresholding to identify microbursts in the analytical signal (Supplementary Fig. 7a). In both juvenile and adult *Fmr1-/y* mice, microbursts occurred more frequently but lasted for shorter durations across both 'gray screen' and 'monitor off' data (Supplementary Fig. 7b, d). Microbursts tended to occur during periods of behavioral quiescence, but there was no evidence of hyperactivity in *Fmr1-/y* mice (Supplementary Fig. 7c, e)[23,57], in contrast to FXS human subjects. Altered microburst dynamics were detectable in L4 but not at the cortical surface (Supplementary Fig. 7f). Similarly, we could not detect these effects in our rsEEG data from adult humans, largely due to immense variability in the TD data (Supplementary Fig. 7h). Intriguingly, L4 burst distributions in WT mice also had larger interquartile ranges than in *Fmr1-/y* mice (Supplementary Fig. 7b, d). Previous work identified elevated burst counts in FXS male subjects in inter-stimulus intervals of a task (not resting-state data)[56], consistent

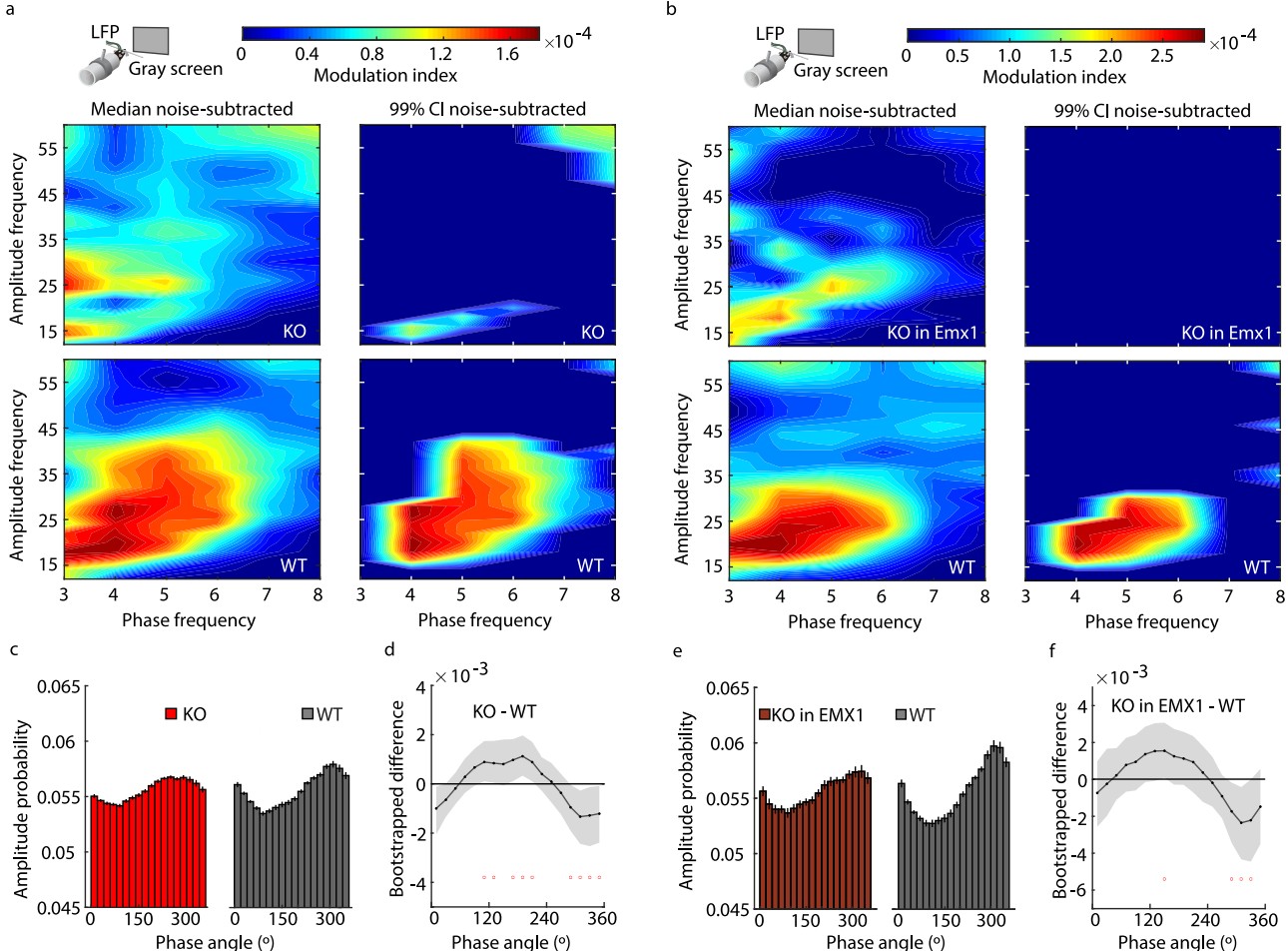

**Fig. 5 | Phase-amplitude coupling of alpha-like and 15-40 Hz oscillations is impaired in *Fmr1⁻/ʸ* mice. a** (left) Median bootstrapped noise-subtracted cross-frequency comodulogram for juvenile (p30–40) *Fmr1⁻/ʸ* (KO) mice (top) and littermate wild-type (WT) mice (bottom) viewing a static, iso-luminant gray screen, *n* = 67 per group. Warmer colors indicate stronger coupling (modulation index, MI) between the phase of alpha-like oscillations (periodic Pk1, 2–9 Hz) and the amplitude of higher frequency oscillations. (right) Boostrapped median comodulogram MI values that are significantly greater than noise (i.e., MI – noise > 0 with 99% confidence interval (CI), non-parametric hierarchical bootstrap). All other median MI values are set to 0. **b** Same as (**a**) but for juvenile Emx1-*Fmr1* KO (*n* = 15) and littermate WT mice (*n* = 29) viewing the static gray screen. **c** Probability distribution (mean +/ SEM) for gray-screen viewing juvenile KO mice (left, *n* = 67) and WT mice (right, *n* = 67) of 15–40 Hz amplitude values occurring in one of 18 bins of the periodic Pk1 (4–6 Hz) phase. A flatter distribution yields a smaller MI. **d** Bootstrapped difference (median +/− 99% CI) between the KO and WT mice amplitude probability distributions shown in (**c**). Dots at the bottom of the plot indicate the points of significant difference between groups, where the CI does not overlap with zero (non-parametric hierarchical bootstrap, 99% CI). **e**, **f** Same as **c**, **d** but for juvenile Emx1-*Fmr1* KO (*n* = 15) and littermate WT mice (*n* = 29). Source data are provided as a Source Data file.

with our results in *Fmr1⁻/ʸ* mice. For resting-state data, intracortical recordings revealed differences in temporal dynamics not detectable from the surface. Studying these altered temporal dynamics might help generate hypotheses of mechanistic disruptions in FXS, but from a translational perspective remains less useful than the alpha phenotype, which we detected across species and both inside and outside mouse cortex.

### Loss of FMRP in cortical excitatory neurons and glia in mice is sufficient for periodic Pk1 phenotypes

Thus far, we have characterized the alpha phenotype in both children and adults with FXS and established a correlate of the phenotype and its age-related changes in V1 of juvenile and adult *Fmr1⁻/ʸ* mice through surface and intracortical recordings, bolstering its potential translational utility for testing therapeutic outcomes. Importantly, we can interrogate the cellular basis of the phenotype in the mouse model in ways not possible in humans to inform therapeutic avenues. For example, it was previously shown that deletion of FMRP in excitatory neurons is sufficient for the gamma phenotype in *Fmr1⁻/ʸ* mice[58]. To test if it is also sufficient for the alpha phenotype, we employed a genetic

strategy to selectively knockout *Fmr1* from excitatory neurons and glia in the cortex (Emx1-*Fmr1* KO), sparing its expression in cortical inhibitory interneurons and subcortically[59]. We used juvenile mice (p30–40) and implanted electrodes in L4 of V1 to record head-fixed resting-state LFP while the monitor was off or displaying the gray screen. WT mice in this triple transgenic line (*n* = 20) showed no difference from WT mice used previously (*n* = 9) (Supplementary Fig. 8a–e). As shown in Fig. 4 and Supplementary Fig. 8f–m, all the resting-state periodic LFP phenotypes in both luminance and darkness replicated in Emx1-*Fmr1* KO mice (*n* = 15). Loss of FMRP in excitatory neurons and glia is sufficient for the alpha phenotype.

### Altered coupling between alpha-like and higher frequency oscillations in the absence of FMRP

Previous studies have shown a coupling between the phase of mouse alpha-like oscillations in L4 of V1 and the amplitude of higher frequency oscillations in the beta and gamma frequency bands (15–40 Hz)[53,60,61]. We have reproduced that finding in littermate WT mice and shown that this phase-amplitude coupling is impaired and phase-shifted in *Fmr1⁻/ʸ* and Emx1-*Fmr1* KO mice (Fig. 5). Since the

higher frequency oscillations are within the range of human Pk2, this echoes the reduced correlation between the power of periodic Pk1 and Pk2 in adult humans with FXS (Supplementary Fig. 1j). In mice, these higher frequency oscillations are thought to be coordinated by PV+ and SOM+ interneurons[61–65], suggesting a link between the activity of these inhibitory (GABAergic) interneurons and periodic Pk1.

### Alpha-like oscillations in mice are linked to the differential activity of genetically defined classes of GABAergic interneurons

We next investigated this hypothesized link between periodic Pk1 and the activity of PV+ and SOM+ interneurons. Alpha oscillations are broadly tied to cortical inhibition[37,66–68] and reflect activity in the thalamocortical loop[36,54,69], but whether they also reflect activity of specific classes of cortical inhibitory (GABAergic) interneurons is unclear. In mice, previous reports suggest that either PV+ or SOM+ interneurons coordinate cortical or visually-evoked theta oscillations, but these reports have been conflicting[60,61,70,71]. Here, we test the effect of temporarily inactivating PV+ and SOM+ neurons in WT mice on the resting-state LFP, particularly alpha-like oscillations.

We began by investigating the effect of inhibiting PV+ neurons in V1 of WT mice on the L4 LFP, particularly within periodic Pk1a and Pk1b. We re-analyzed previously published data wherein chemogenetic methods were used to temporarily inactivate PV+ cells in V1[72]. We analyzed 150 seconds of resting-state LFP data where head-fixed mice ($n = 16$) expressed the chemogenetic actuator hM4Di in PV+ neurons of V1 and passively viewed a static, iso-luminant gray screen before and after systemic injection of Clozapine N-Oxide (CNO) to trigger the hM4Di receptor. Inactivating PV+ cells increased absolute power between 1.6–110.4 Hz (non-parametric bootstrapping with 99% confidence interval; Fig. 6a). This effect resulted from an increase in aperiodic power between 1.6–106.2 Hz and an increase in periodic power between 9.6–115.8 Hz (Fig. 6b, f). The paradoxical increase in aperiodic and periodic gamma power is presumably through disinhibition of reciprocally connected excitatory cells[73–76]. Indeed, aperiodic slope, offset, and knee frequency values were all significantly affected by the manipulation, indicating a large effect on network excitability (Fig. 6c–e, Supplementary Fig. 9a, b). The manipulation also increased the frequency and shortened the duration of Pk1 microbursts without changing overall movement (Supplementary Fig. 9c–f), but this analysis was affected by the large changes in aperiodic power and was thus not further utilized in the manuscript. Removing aperiodic power crucially revealed that PV+ neuronal inactivation eliminated periodic Pk1b power (significant difference 6.8–8.2 Hz) but did not affect Pk1a power or center frequency (Fig. 6f–j).

Next, we used optogenetics to temporarily inactivate SOM+ activity in V1 while recording resting-state LFP from head-fixed mice passively viewing a static, iso-luminant gray screen (Fig. 6k). Optogenetic inactivation reduced absolute power between 4.6–6.2 Hz in mice expressing the inhibitory actuator Halorhodopsin (NpHR3) in SOM+ neurons of V1 ($n = 8$), relative to control mice ($n = 7$; Fig. 6k). The manipulation had no effect on the aperiodic spectrum (Fig. 6l–o), revealing a specific effect on periodic Pk1, with a significant reduction in periodic power in NpHR3-expressing mice between 4.4–6.4 Hz, specifically within Pk1a (Fig. 6p). Comparing Pk1a maximum power across genotypes revealed a strong trend supporting the bootstrapped spectrum results (Fig. 6r; uncorrected $p$-value = 0.02), with no apparent difference in Pk1b maximum power (Fig. 6t) nor the center frequency of either subpeak (Fig. 6q, s). The power of Pk1a and Pk1b are therefore separately affected by inhibition of SOM+ and PV+ cells, respectively.

### Periodic Pk1 is sensitive to activation of GABA$_B$ receptors by Arbaclofen

Since shutting off PV+ and SOM+ interneurons affected alpha-like oscillations in WT mice, the oscillations might also be sensitive to drugs which enhance GABAergic inhibition. Pharmacologically boosting inhibition is a well-trodden therapeutic avenue for FXS[77–79], wherein GABAergic activity is impaired[9,77,78]. Indeed, hypoactivity of SOM+ and/or PV+ interneurons has been documented for FXS[80–85], even when deletion of Fmr1 is restricted to excitatory neurons[58], so deficient activity of these classes of GABAergic interneurons might contribute to the electrophysiological differences we have observed in the LFP of Fmr1$^{-/y}$ and Emx1-Fmr1 KO mice. To test if boosting GABAergic inhibition might affect alpha-like oscillations in WT mice and even be therapeutic to periodic Pk1 phenotypes in Fmr1$^{-/y}$ mice, we used Arbaclofen, a GABA$_B$ agonist. Arbaclofen was previously shown to reverse several intracellular and behavioral phenotypes in Fmr1$^{-/y}$ mice[86,87] and has produced clinically meaningful improvements in several secondary measures in a phase-3 clinical trial with children with FXS[88]. We administered Arbaclofen intraperitoneally 60−90 minutes before characterizing 150 seconds of resting-state L4 LFP data in V1 while juvenile head-fixed mice (p30−40) passively viewed a static gray screen.

Arbaclofen was administered in WT mice at two doses, 0.5 mg/kg ($n = 10$) and 1 mg/kg ($n = 15$), and we analyzed the periodic and aperiodic spectra to probe for process engagement. Our analytical method more precisely read out dose response differences in the LFP than could be realized through analysis of the absolute spectra (Fig. 7). In the absolute spectra, there was a significant increase in low frequency power from 3.4−5 Hz for 0.5 mg/kg and from 1.6−16.2 Hz for 1 mg/kg relative to saline treatment (non-parametric bootstrapping with 99% confidence interval; Fig. 7a, k). 1 mg/kg of Arbaclofen significantly increased aperiodic power below 39 Hz and decreased it above 172 Hz (Fig. 7l), increased aperiodic slope and offset but not knee frequency (Fig. 7m–o), and decreased high frequency periodic power between 21–52.8 Hz and 56.2–76.6 Hz (Fig. 7p). At the 0.5 mg/kg dose, only increased low frequency aperiodic power and offset were observed (Fig. 7b–f). At both doses, Arbaclofen increased periodic power from 1.8–4.6 Hz (Fig. 7f, p) and increased the maximum power of Pk1a but not Pk1b (Fig. 7h, j, r, t). At 1 mg/kg, Arbaclofen also significantly reduced the center frequency of Pk1a (Fig. 7q). It is noteworthy that optogenetically stimulating SOM+ interneurons had the same effect on Pk1a center frequency (Supplementary Fig. 10c, d).

Significantly, these findings in WT mice demonstrate that Pk1a center frequency and power are sensitive to pharmacological intervention, bolstering the utility of alpha oscillation disruptions, including the alpha phenotype, as translatable measures of pathophysiology and therapeutic target engagement. Moreover, since both the 0.5 and 1 mg/kg doses increased Pk1a power, it is possible that Arbaclofen might improve the Pk1a power phenotype in Fmr1$^{-/y}$ mice.

### Periodic Pk1 is less sensitive to Arbaclofen in Fmr1$^{-/y}$ than WT mice

We next tested the same two doses (0.5 mg/kg, $n = 10$, and 1 mg/kg, $n = 14$) on the resting-state L4 LFP in head-fixed juvenile Fmr1$^{-/y}$ mice (p30−40) to investigate if Arbaclofen has a therapeutic effect on periodic Pk1a power. Relative to saline controls, we observed a significant increase in the absolute spectra below 9 Hz and a significant decrease between 72.4−74.6 Hz for the 1 mg/kg dose but not 0.5 mg/kg dose (Fig. 8a, f). As with WT mice, the 1 mg/kg dose significantly increased aperiodic power below 30.6 Hz and decreased it above 174 Hz (Fig. 8g), increased aperiodic slope and offset (Fig. 8g inset), and decreased high frequency periodic power between 21–125.6 Hz (Fig. 8h), while the 0.5 mg/kg dose only affected the aperiodic spectrum (Fig. 8b, c). The decrease in high-frequency periodic power recapitulated the therapeutic effect of Arbaclofen and racemic baclofen on the gamma phenotype in Fmr1$^{-/y}$ mice[25,89,90]. Unlike WT mice, Arbaclofen had no effect on Pk1a power at the lower dose (0.5 mg/kg; Fig. 8c–e). At 1 mg/kg, there was a significant periodic power difference from 2.6−4.4 Hz (Fig. 8h), so we compared Pk1a properties to saline-treated

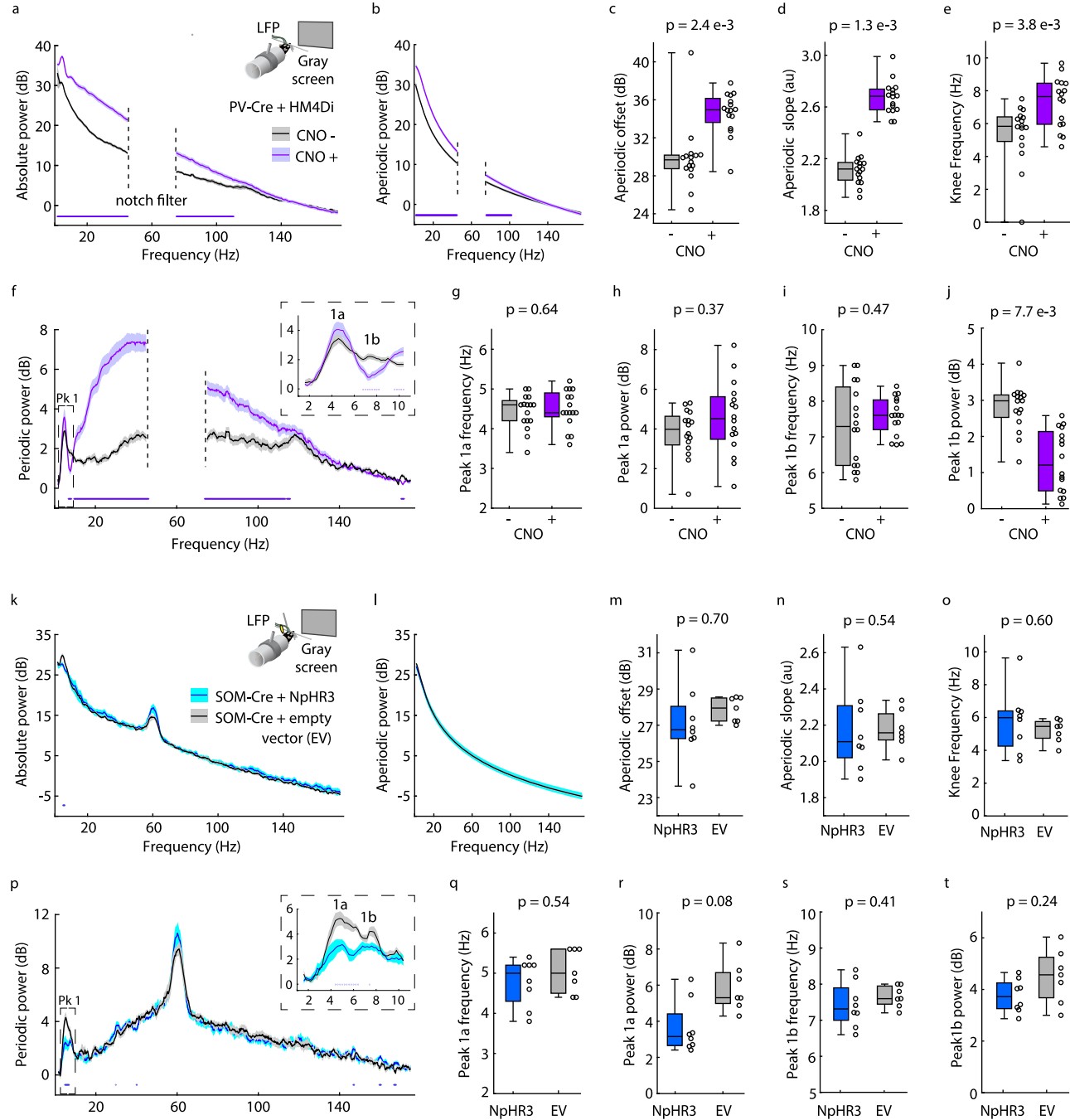

**Fig. 6 | Alpha-like oscillations are linked to differential activity of PV+ and SOM + GABAergic interneurons. a** Inset: Experimental design. V1 L4 LFP data were collected in head-fixed PV-Cre mice (p50–80, *n* = 16) with chemogenetic actuator hM4Di expressed in V1[72], viewing a static, iso-luminant gray screen. Main: Absolute power spectrum (mean +/− SEM) from 150 seconds of data before (black) and after (purple) systemic injection of CNO. Notch filter 45-75 Hz. **b** Aperiodic fit (mean +/− SEM). **c**–**e** Boxplot and FDR-corrected *p*-values for: **c** Aperiodic offset (uncorrected *p*-value = 1.6e−3, z-statistic = −3.15, effect size = −0.775); **d** Aperiodic slope (uncorrected *p*-value = 4.38e−4, z-statistic = −3.5216, effect size = −1); and **e** Aperiodic knee frequency (z-statistic = −2.896, effect size = −0.55). **f** Main: Periodic power (mean +/− SEM). Inset: Peak 1 (Pk1) with higher spectral resolution. **g**–**j** Boxplot and FDR-corrected *p*-values for: **g** Pk1a center frequency. Z-statistic = −0.487, effect size = −0.046; **h** Pk1a maximum power. Uncorrected *p*-value = 0.234, z-statistic = −1.189, effect size = −0.258; **i** Pk1b center frequency. Uncorrected *p*-value = 0.278, z-statistic = −1.086, effect size = −0.167; and **j** Pk1b maximum power. Uncorrected *p*-value = 1.9e−3, z-statistic = 3.103, effect size = 0.858. **k**–**t** Same as **a**–**j** but for SOM-Cre mice (p50-70) expressing either an inhibitory opsin virus (NpHR3, *n* = 8) or empty viral vector (EV, *n* = 7) in V1. Power spectra derived from 'laser on' periods. For **m**, uncorrected *p*-value = 0.232; **o** uncorrected *p*-value = 0.397; **q** uncorrected *p*-value = 0.408; **s** uncorrected *p*-value = 0.02; and **t** uncorrected *p*-value = 0.12. Dots at the bottom of plots in (**a**, **b**, **f**, **k**, **p**) indicate the points of significant difference between groups (non-parametric hierarchical bootstrap, 99% confidence interval). Boxplots show 25th, median, and 75th percentiles, with whiskers extending to minimum and maximum values. In (**c**–**e**, **g**–**j**), uncorrected *p*-values and z-statistics calculated from two-sample, two-sided Wilcoxon Signed-Rank Tests, and effect sizes calculated using Cliff's Delta. In (**m**–**o**, **q**–**t**), uncorrected *p*-values calculated from two-sample, two-sided Wilcoxon Rank-Sum Tests. Z-statistics and effect sizes not calculated for *n* < 10. *P*-values adjusted with the Benjamini-Hochberg correction. Source data are provided as a Source Data file.

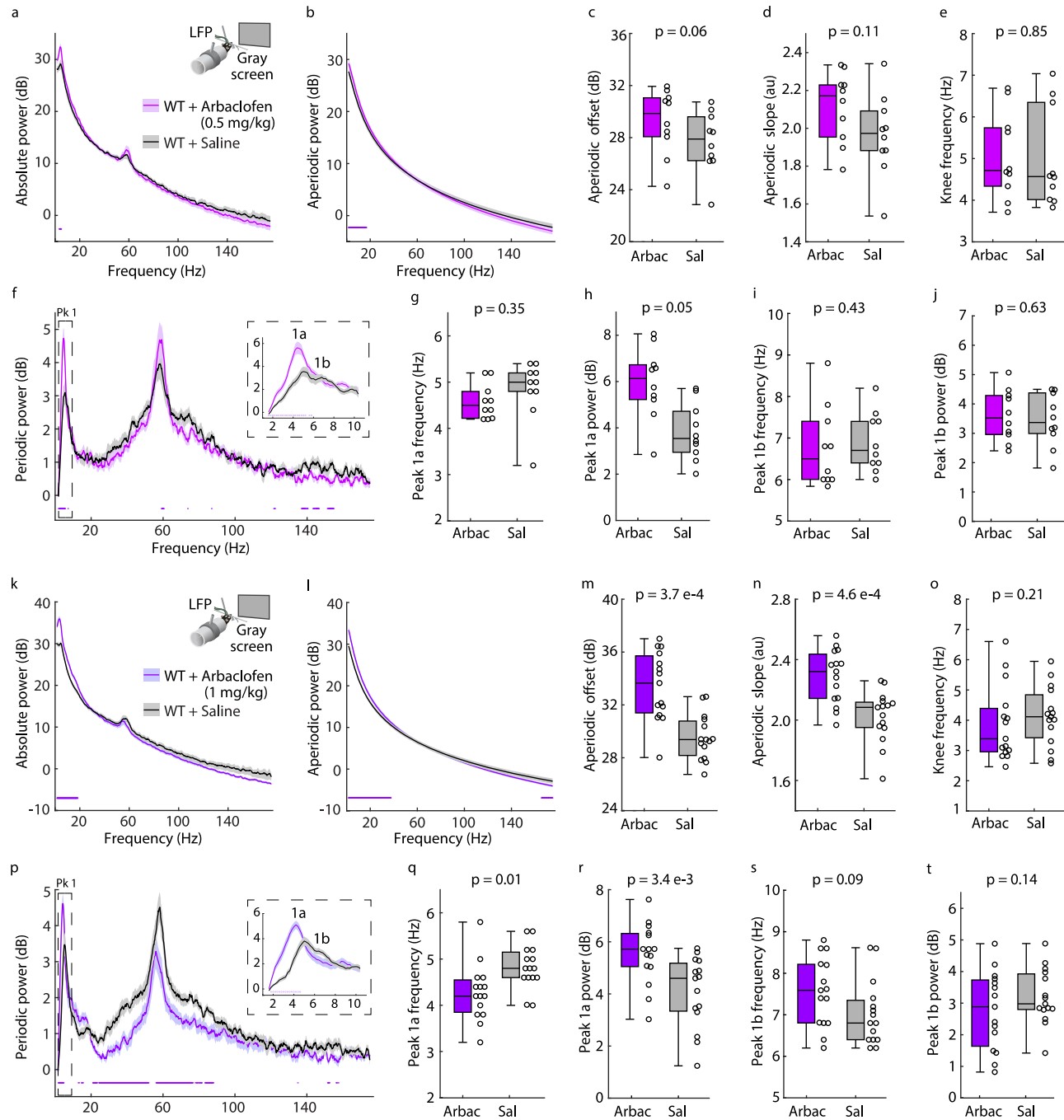

**Fig. 7 | Alpha-like oscillations are sensitive to the GABA_B agonist Arbaclofen.**
**a** Inset: Experimental design. V1 L4 LFP data were collected in head-fixed juvenile (p30–40) WT mice ($n = 10$, littermate to *Fmr1^-/y* in Fig. 8a–e) viewing a static, iso-luminant gray screen after systemic injections of saline (Sal, black) and 0.5 mg/kg Arbaclofen (Arbac, purple). Main: Absolute power spectrum (mean +/− SEM) from 150 seconds of data, 60–90 minutes after each injection. **b** Aperiodic fit (mean +/− SEM). **c–e** Boxplot and FDR-corrected *p*-values for: **c** Aperiodic offset (uncorrected *p*-value = 0.037, effect size = 0.378); **d** Aperiodic slope (uncorrected *p*-value = 0.037, effect size = 0.4); and **e** Aperiodic knee frequency (effect size = 0.156). **f** Main: Periodic power spectrum (mean +/− SEM). Inset: Peak 1 (Pk1) at higher spectral resolution. **g–j** Boxplot and FDR-corrected *p*-values for: **g** Pk1a center frequency. Uncorrected *p*-value = 0.176, effect size = −0.456; **h** Pk1a maximum power. Uncorrected *p*-value = 0.014, effect size = 0.689; **i** Pk1b center frequency. Uncorrected *p*-value = 0.322, effect size = −0.089; and **j** Pk1b maximum power.

Effect size = 0.022. **k–t** Like **a–j** but for juvenile WT mice ($n = 15$, littermate to *Fmr1^-/y* in Fig. 8f–j) treated with 1 mg/kg Arbaclofen (purple) or saline (black). For **m**, uncorrected *p*-value = 1.22e−4, effect size = 0.771. For **n**, uncorrected *p*-value = 3.05e−4, effect size = 0.695. For **o**, effect size = −0.171. For **q**, uncorrected *p*-value = 0.007, effect size = −0.6. For **r**, uncorrected *p*-value = 8.54e−4, effect size = 0.61. For **s**, uncorrected *p*-value = 0.07, effect size = 0.357. For **t**, effect size = −0.2. Dots at the bottom of plots in (**a**, **b**, **f**, **k**, **l**, **p**) indicate the points of significant difference between groups (non-parametric hierarchical bootstrap, 99% confidence interval). Boxplots show 25th, median, and 75th percentiles, with whiskers extending to minimum and maximum values. Uncorrected *p*-values calculated from two-sample, two-sided Wilcoxon Signed-Rank Tests. Z-statistics not calculated for $n < 16$. Effect sizes calculated using Cliff's Delta. *P*-values adjusted with the Benjamini-Hochberg correction. Source data are provided as a Source Data file.

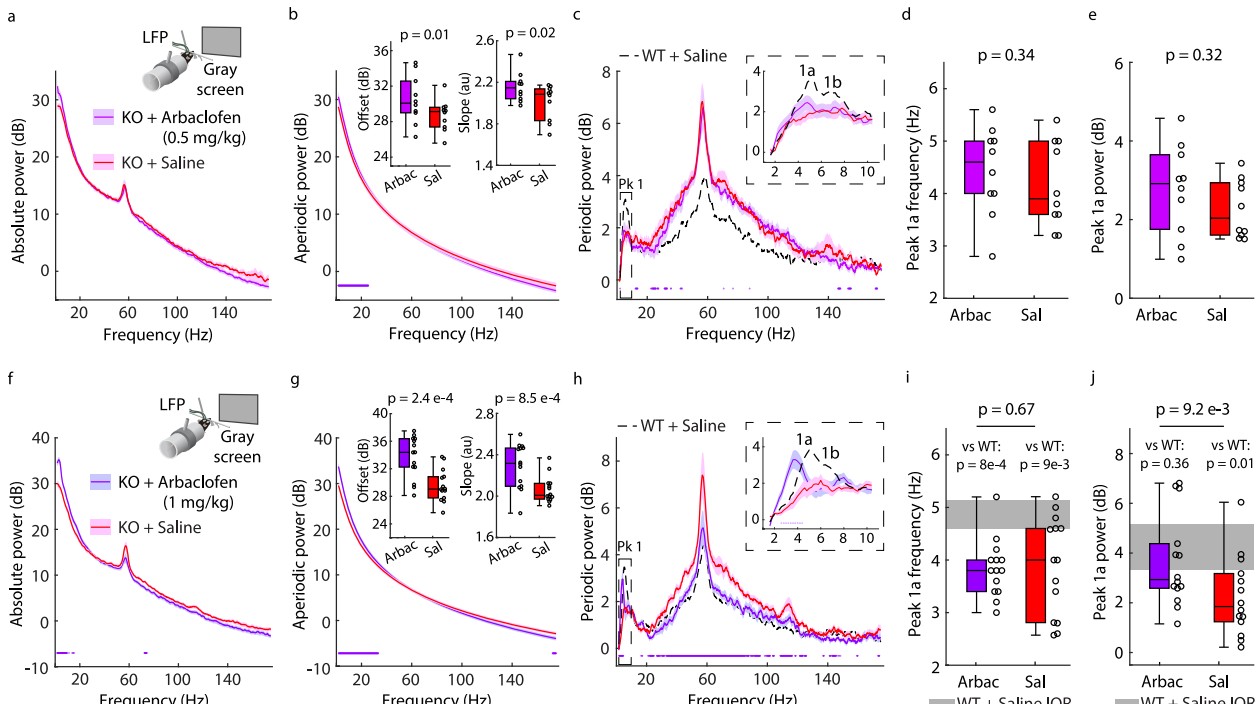

**Fig. 8 | Alpha-like oscillations are less sensitive to Arbaclofen in *Fmr1*⁻/y than WT mice. a** Inset: Like Fig. 7a, but for *Fmr1*⁻/y (KO) mice (*n* = 10, littermates to WT in Fig. 7a-j) after saline (Sal, red) and 0.5 mg/kg Arbaclofen (Arbac, purple) injections. Main: Absolute power spectrum (mean +/− SEM), 60–90 minutes after injections. **b** Main: Aperiodic fit (mean +/− SEM). Inset: Boxplot and FDR-corrected *p*-values for offset (effect size = 0.378) and slope (uncorrected *p*-value = 0.01, effect size = 0.422). **c** Main: Periodic power spectrum (mean +/− SEM). Dashed line is mean periodic power for saline-treated littermate WT mice (replotted from Fig. 7f). Inset: Pk1 at higher spectral resolution. **d–e** Boxplot and FDR-corrected *p*-values for: **d** Pk1a center frequency (effect size = 0.2); and **e** Pk1a maximum power (uncorrected *p*-value = 0.16, effect size = 0.267). **f–j** Like **a–e** but for KO mice (*n* = 14, littermates to WT in Fig. 7k-t) treated with 1 mg/kg Arbaclofen (purple) or saline (red). For **g**, offset uncorrected *p*-value = 1.22e−4, effect size = 0.703; slope effect size = 0.637. For **h**, dashed line replotted from Fig. 7p. For **i**, **j**, gray shading covers the interquartile range (IQR, 25th–75th percentiles) for saline-treated littermate WT (replotted from Fig. 7q, r). In **i**, between KO+Arbac and KO+saline, effect size =

−0.071. Between KO+Arbac and WT+saline, uncorrected *p*-value = 1.37e−4, z-statistic = −3.814, effect size = −0.833. Between KO+saline and WT+saline, uncorrected *p*-value = 0.006, z-statistic = −2.768, effect size = −0.605. In **j**, between KO+Arbac and KO+saline, uncorrected *p*-value = 0.003, effect size = 0.44. Between KO+Arbac and WT+saline, uncorrected *p*-value = 0.305, z-statistic = −1.206, effect size = −0.229. Between KO+saline and WT+saline, uncorrected *p*-value = 0.005, z-statistic = −2.815, effect size = −0.619. Dots at bottom of plots in (**b, c, f, g, h**) indicate points of significant difference between groups (non-parametric hierarchical bootstrap, 99% confidence interval). Boxplots show 25th, median, and 75th percentiles, with whiskers extending to minimum and maximum values. For **b, d, e, g** and within-genotype comparisons in (**i, j**), uncorrected *p*-values calculated from two-sample, two-sided Wilcoxon Signed-Rank Tests. Z-statistics not calculated for *n* < 16. For **i, j** cross-genotype comparisons, uncorrected *p*-values and z-statistics calculated from two-sample, two-sided Wilcoxon Rank-Sum Tests. Effect sizes calculated using Cliff's Delta. *P*-values adjusted with the Benjamini-Hochberg correction. Source data are provided as a Source Data file.

WT littermates (from Fig. 7q, r) as well as between treatment groups in *Fmr1*⁻/y mice. Between the latter, we observed a tighter interquartile range but no significant shift in Pk1a center frequency and a significant improvement in Pk1a maximum power with Arbaclofen treatment (Fig. 8h-j). Relative to saline-treated WT, the genotype difference in Pk1a power was no longer significant while the genotype difference in Pk1a frequency was enhanced in *Fmr1*⁻/y mice treated with Arbaclofen.

Since the minimum effective dose for Arbaclofen to shift Pk1a power was higher in *Fmr1*⁻/y mice, we also tested a 3 mg/kg dose in a separate, small cohort of head-fixed juvenile *Fmr1*⁻/y mice (*n* = 3, p30−40) passively viewing a static, iso-luminant gray screen. This dose increased aperiodic offset and slope relative to saline-treated WT littermates and essentially eradicated periodic power other than in Pk1a (Supplementary Fig. 11b-c). Our analytical methodology clarified the effective therapeutic dose range, as the negating effect of the 3 mg/kg dose on periodic power is not clear from the absolute spectrum (Supplementary Fig. 11a). This high dose also had a noticeable sedative effect on the mice, as movement over 100 seconds of the 150-second recording was essentially abolished for the mice treated with 3 mg/kg, unlike for *Fmr1*⁻/y mice treated with 1 mg/kg, where forepaw movement was comparable to saline-treated WT littermates (Supplementary Fig. 11d-e).

Finally, to assess if the changes observed in Pk1a were unique to the mechanism of Arbaclofen, we tested the effect of acutely administering CTEP (2-chloro-4-[2-[2,5-dimethyl-1-[4-(trifluoromethoxy)phenyl]imidazol-4-yl]ethynyl]pyridine), a negative allosteric modulator (NAM) of metabotropic glutamate receptor 5 (mGluR5). These NAMs showed immense preclinical promise but poor performance in clinical trials, possibly due to acquired treatment resistance[91–94]. 2 mg/kg of CTEP was administered intraperitoneally 90 minutes before resting-state L4 LFP data were collected while head-fixed adult *Fmr1*⁻/y mice and WT littermates (p70−90) passively viewed a static gray screen (Supplementary Fig. 12a,g). In both WT and *Fmr1*⁻/y mice, CTEP significantly elevated aperiodic offset relative to saline treatment and had a similar flattening effect on high-frequency periodic power as 3 mg/kg Arbaclofen (Supplementary Fig. 12b, c, h, i), revealing target engagement. However, this dose had no effect on Pk1a center frequency or power (Supplementary Fig. 12d, e, j, k), so the effects of Arbaclofen on periodic Pk1a are specific to its mechanism of action (boosting GABAergic inhibition).

## Discussion

Disruptions in alpha oscillations are characteristic of neurodevelopmental and neuropsychiatric disorders, including FXS wherein alpha oscillations are slowed. Although this FXS alpha phenotype is

measurable across the human cortex in both children and adults, we source-localized the oscillation to occipital cortex and discovered a correlate for this phenotype in V1 of $Fmr1^{-/y}$ mice. Since frequency band analysis of absolute spectra can be confounded by aperiodic signal when comparing the same periodic signal across different human ages (i.e., alpha oscillations fall in the theta band in children), we isolated the periodic component of resting-state EEG data. This revealed that the shift in center frequency of periodic Pk1 in FXS is conserved in V1 of $Fmr1^{-/y}$ mice and detectable through both surface EEG and L4 LFP recordings. This discovery supports work positing that theta band oscillations in mice are alpha-like[53,54], and justifies the use of mice to dissect the cause(s) of alpha-band oscillatory EEG phenotypes observed in humans and gain better understanding of how cortical network activity goes awry in disease states. To this end, we show that loss of FMRP in cortical excitatory neurons and glia is sufficient for the FXS alpha phenotype. We also show in mice that the center frequency (and power) of periodic Pk1 can be shifted pharmacologically, with differential sensitivity in $Fmr1^{-/y}$ and WT mice, which crucially validates the use of this transspecies phenotype to objectively quantify putative treatment responses in mice and, eventually, compare these with those in humans with FXS. This is an important advancement over the existing cross-species FXS gamma phenotype for at least two reasons: (1) the alpha phenotype is more reliable in human adults[31–33], and (2) we find that the gamma phenotype is from an aperiodic shift in human adults, from aperiodic and periodic phenotypes in human children, and from a periodic shift in mice, while the alpha phenotype is uniquely periodic across all three – providing a better foundation for comparing treatment responses.

A key variable we found to influence the alpha phenotype is age. Although the Pk1 frequency shift was present in both human children and adults with FXS, the difference was more exaggerated in adults. These cross-sectional findings suggest that the age-dependent increase in Pk1 center frequency observed in typical development[50–52] might stagnate in FXS, consistent with findings in ASD[42]. In future work, it will be important to confirm this conclusion with a longitudinal study conducted at a single site. Impressively, in L4 of V1 of $Fmr1^{-/y}$ mice viewing a static gray screen, we observed that the frequency shift within Pk1 was also exaggerated in adults relative to juveniles. Our L4 recordings clarified a two-subpeak structure of Pk1, with the frequency shift in $Fmr1^{-/y}$ mice in the first subpeak (Pk1a). Unlike humans, Pk1a center frequency decreased across development in mice, and more steeply in adult $Fmr1^{-/y}$ mice, revealing an important and limiting difference between the two species. Nevertheless, the mouse appears to recapitulate different developmental windows of the alpha phenotype.

Intriguingly, we found that in children with FXS, Pk1 was both shifted to a lower frequency and reduced in maximum power relative to age-matched controls. In contrast, the maximum power of Pk1 in FXS adults was not significantly different from adult controls. Thus, relative power differences in the alpha band in FXS adults (wherein the aperiodic shift is largely averaged out) come only from a slowing, not a weakening of alpha oscillations. Similarly, the maximum power of Pk1 measured with EEG over V1 of adult $Fmr1^{-/y}$ mice was not significantly different; however, the L4 LFP recordings revealed altered temporal dynamics and power differences that were not detectable at the surface. We note that intracortical recordings in $Fmr1$-KO rats have similarly shown reduced power of 3-9 Hz oscillations[27]. We observed a significant reduction in Pk1a power in adult $Fmr1^{-/y}$ mice and in both Pk1a and Pk1b power in juvenile $Fmr1^{-/y}$ mice. Pk1b power increased over development in $Fmr1^{-/y}$ mice, so the power phenotype in this subpeak was specific to juveniles, just like the Pk1 power phenotype was specific to FXS children. Since power and frequency alterations of Pk1 may have multiple mechanistic bases, efficacy of therapeutics could vary within different developmental windows[95]. The presence of a juvenile-specific power phenotype in addition to the alpha

phenotype in $Fmr1^{-/y}$ mice means the model may be useful to optimize the timing of treatments. Future resting-state recordings in younger mice (p21) might reveal further developmental differences[29,96].

Of note, we did not observe the power difference in Pk1b in juvenile mice when the animals were in the dark, as luminance suppressed Pk1b power in juvenile $Fmr1^{-/y}$ mice (but not in adults or in littermate WT mice). Luminance also enhanced Pk1a power in juvenile WT but not $Fmr1^{-/y}$ mice, further amplifying the genotype effect. In neurotypical humans, alpha oscillations are enhanced in the dark or when eyes are closed[97,98]. This difference could arise because mice are nocturnal while humans are diurnal. In the current study, FXS children were recorded in a dim room, but were watching different videos and room luminance was sometimes adjusted to increase compliance, a limitation of our current study. It will be of interest to test in future studies how luminance affects the alterations in Pk1 power in FXS children.

The utility of studying the mouse model is that it offers the opportunity to enter the brain to dissect the underlying basis of alpha-like oscillations to inform new avenues for therapy. Through our L4 LFP analysis in $Fmr1^{-/y}$ mice, we discovered a two-subpeak structure to the alpha-like oscillations, with the frequency shift in Pk1a mapping onto the alpha phenotype and the juvenile-specific power difference in Pk1b mapping onto the Pk1 power difference specific to FXS children. Our discovery of the two-subpeak structure of Pk1 permitted an important advancement in our understanding of how cortical GABAergic inhibition relates to alpha oscillations. Previously, theta (alpha-like) oscillations in WT mice have been analyzed as a single entity across this frequency range, giving rise to differing opinions as to which population of inhibitory interneurons coordinates these oscillations[60,61,70,71]. The two-subpeak structure has offered some clarity: inhibition of SOM+ and PV+ neurons independently affects Pk1a and Pk1b, respectively. A mechanistic distinction in subpeaks might translate to humans, given reports of differential functionality of the lower and upper portions of the alpha frequency band[19,99,100].

Our interest in the relationship between GABAergic interneurons and periodic Pk1 in mice stemmed from the observation that coupling of alpha-like oscillations to 15-40 Hz oscillations (regulated by SOM+ and PV+ interneurons[61–65]) is weaker in L4 of $Fmr1^{-/y}$ and Emx1-$Fmr1$ KO mice. Since GABAergic inhibition is deficient in FXS[9,77,78], our manipulations of SOM+ and PV+ interneurons suggest that the FXS gamma and Pk1 power phenotypes might reflect hypoactivity of SOM+ and PV+ interneurons, although this inference is correlative. Intriguingly, both classes of inhibitory interneurons are inhibited by vasoactive intestinal peptide-expressing neurons, which are documented as hyperactive in $Fmr1^{-/y}$ mice[101–104]. If altered network inhibition contributes to alpha oscillation disruptions, then there might be therapeutic potential in boosting inhibition to correct the disruptions.

Our manipulations of SOM+ and PV+ interneurons also suggest that alpha-like oscillations should be sensitive to pharmacologically boosting GABAergic inhibition. To investigate this, we tested the effects of administering the $GABA_B$ agonist Arbaclofen on alpha-like oscillations in WT mice at two different doses, 0.5 mg/kg and 1 mg/kg. Our analytic methodology allowed us to identify dose-dependent process engagement through changes in both the aperiodic and periodic components, including, critically, Pk1a power and frequency. Arbaclofen increased Pk1a power at both doses and significantly decreased its center frequency at the higher dose (with a clear trend at the lower dose). Intriguingly, optogenetically simulating SOM+ interneurons also decreased Pk1a center frequency. The effect of Arbaclofen on Pk1a was specific to its mechanism of action (boosting inhibition), as the mGluR5 NAM CTEP affected high-frequency periodic power and aperiodic power with no effect on Pk1a in either WT or $Fmr1^{-/y}$ mice.

While alpha-like oscillations were quite sensitive to Arbaclofen in WT mice, they were much less so in $Fmr1^{-/y}$ mice. We tested the same

two doses in *Fmr1*[-/y] mice and found no substantial shift in Pk1a center frequency at either dose (although there was a tightening of the interquartile range at the higher dose) and an increase in Pk1a power only at the higher dose. In contrast, the effect of the drug on the aperiodic spectrum and on reducing high-frequency periodic power at the higher dose (consistent with previous reports of improvement of the gamma phenotype by Arbaclofen and racemic baclofen[25,89,90] matched between WT and *Fmr1*[-/y] mice. This suggests that the differential sensitivity to Arbaclofen between genotypes was specific to alpha-like oscillations and demonstrates how measuring alpha oscillations (and their disruptions) clarifies our understanding of treatment response (especially in mouse models, where treatment comparisons between genotypes are more feasible). Future work should investigate this genotype difference in sensitivity of alpha-like oscillations to Arbaclofen, to test if this relates to the hypoactivity of SOM+ and/or PV+ interneurons reported in *Fmr1*[-/y] mice[80-85].

Relative to saline-treated littermate WT mice, *Fmr1*[-/y] mice treated with 1 mg/kg of Arbaclofen no longer displayed the gamma phenotype or a genotype difference in Pk1a maximum power, consistent with reports of a corrective effect of the drug on phenotypes in *Fmr1*[-/y] mice[25,86,87], but had an increased genotype difference in Pk1a center frequency. This can be compared to recent work in humans with FXS showing that the peak frequency of alpha oscillations could be increased by BPN14770, a phosphodiesterase (PDE)4D allosteric inhibitor[105]. Based on our findings, the improvement of the alpha phenotype by BPN14770 suggests that its therapeutic mechanism of action is likely not solely through altering inhibition. Since inhibition in FXS may be altered as a secondary consequence of changes in excitatory neuronal function[58,84,85,106], correction of the alpha phenotype might instead come from targeting excitatory neuron and corticothalamic activity, given the role of the thalamocortical loop in alpha oscillations[36,54,69]. Future work will hopefully elucidate how loss of FMRP affects excitatory neuron activity in vivo, clarifying whether it drives hyperexcitability[6,107] or hypoactivity[27,85] of principal cells, and how this altered activity affects alpha-like oscillations and relates to the therapeutic effect of the PDE4D inhibitor.

Until recently, human alpha oscillations were considered a signal of cognition that could not be studied in mice, and now we have discovered that the FXS alpha phenotype is conserved in alpha-like oscillations of *Fmr1*[-/y] mice and that the power and frequency of murine alpha-like oscillations can be shifted by therapeutics. This offers the possibility of a parallel measure of drug efficacy across species to improve screening of therapeutics preclinically and establish therapeutic dose ranges, which increases confidence that successful correction of the disruptions in the mouse model will translate to success in humans in clinical trials. Our findings may also fuel research into the underlying mechanisms of alpha oscillations and their disruptions to inform new therapeutic avenues. We hope our framework for studying alpha disruptions preclinically can be applied to research on other neuropsychiatric and neurodevelopmental disorders with pathological alpha oscillations to broadly advance translational research.

## Methods
### Ethics Approval
Our studies in human subjects complied with all relevant ethical regulations, as approval was obtained prior to starting each study from the Institutional Review Board (IRB) at Boston Children's Hospital or the Institutional Review Board at Cincinnati Children's Hospital and Medical Center, depending on the site where the study was conducted. Written informed consent was obtained from either the participant or from the participant's parent/guardian, with assent from the participant. For the pediatric study, written informed consent was obtained from all parents or guardians prior to their children's participation in the study.

Our studies in murine subjects also complied with all relevant ethical regulations, as all experimental techniques were approved by The Institutional Animal Care and Use Committees and Department of Comparative Medicine at MIT (Bear protocol: 2403000639) and conformed to the Guide for the Care and Use of Laboratory Animals published by the National Institutes of Health, or were approved by the Ethical Committee for Animal Use of King's College London (Home Office License: PF82DF031-5) and were in accordance with the United Kingdom Home Office Animals (Scientific Procedures) Act 1986, depending on the site where the data were collected.

### Human subjects
**Participants for Pediatric FXS study.** Resting-state EEG (rsEEG) data were collected from 17 males (27–78 months old) with full mutation of *FMR1* and 17 age-matched (27–80 months) typically-developing males. Participants were recruited as part of two studies (IRB-P00034676 and IRB-P00025493) conducted at Boston Children's Hospital/Harvard Medical School. An additional secondary analysis IRB was approved for combining and analyzing data from both studies. FXS participants all had documented full mutation of the *FMR1* gene, however methylation status was not known for many participants, and participants were not excluded for size mosaicism (mixture of full and premutation). Additional exclusion criteria included history of prematurity (<35 weeks gestational age), low birth weight (<2000gms), known birth trauma, known genetic disorders (other than FXS), unstable seizure disorder, current use of anticonvulsant medication, and uncorrected hearing or vision problems.

**Participants for Adult FXS study.** rsEEG data were collected from 20 males (19–43 years old; mean = 31.09, standard deviation = 7.67) with a full mutation of *FMR1* and 20 age-matched (20–44 years old; mean = 29.42, standard deviation = 7.12) typically-developed males. Participants were recruited as part of a study (IRB #2013-7327) conducted at Cincinnati Children's Hospital and Medical Center. Adult FXS participants all had full *FMR1* mutations (>200 CGG repeats), confirmed via past testing results made available in a participant's medical record or via Southern Blot and/or PCR conducted in collaboration with the Molecular Diagnostic Laboratory at Rush University. Participants were not excluded for size mosaicism (mixture of full and premutation). Additional exclusion criteria for adult participants included known syndromic conditions associated with intellectual disability (other than FXS; e.g., Down Syndrome), being on benzodiazepines or anticonvulsant medications, or being on any novel potential treatment (e.g., minocycline) known to affect EEG measures within 4 weeks of testing. Typically developed controls were excluded if they scored ≥ 8 on the Social Communication Questionnaire (SCQ ® Print Kit from Rutter, Bailey, & Lord, 2003, W-381 WPS Publishing, Torrence, CA), had a history of psychiatric or neurologic disorders, or had a first or second-degree relative with autism spectrum disorder or a serious psychiatric illness.

We note that our study focused on male subjects (both humans and mice). The human *FMR1* and mouse *Fmr1* genes are on the X-chromosome, FXS diagnoses are most prevalent in males, and the electrophysiological phenotypes are most prominent in the male subjects[19]. However, we recognize that there are sexually dimorphic phenotypes in FXS, including in alpha oscillations[18,56], and encourage future study of the alpha phenotype in human females with FXS and in female *Fmr1*[-/y] mice.

### Human rsEEG data collection and preprocessing
**Pediatric study.** rsEEG data were collected while the child either sat in their caregiver's lab or independently in a chair situated in a dimly lit, sound-attenuated, electrically shielded room. At times, room luminance levels were increased for the comfort of the child. Continuous EEG was recorded while the participant was shown a silent screen saver

of abstract, colorful moving images for 2–5 minutes depending on the child's compliance. In some cases, participants watched a silent video of their choosing to improve behavioral cooperation.

rsEEG data were collected using a 128–channel Hydrocel Geodesic Sensor Net (Version 1, EGI Inc, Eugene, OR) connected to a DC–coupled amplifier (Net Amps 300, EGI Inc, Eugene, OR). Data were sampled at 1000 Hz and referenced to a single vertex electrode (Cz). Raw Net-Station (NetStation version 4.5, EGI Inc, Eugene, OR) files were exported to MATLAB (version R2017a) for pre-processing using the Batch EEG Automated Processing Platform (BEAPP)[108] with integrated Harvard Automated Preprocessing Pipeline for EEG (HAPPE)[109]. Pre-processing has previously been described in detail for similar data[110]. Briefly, data were 1 Hz high-pass and 100 Hz low-pass filtered, down-sampled to 250 Hz, and then run through the HAPPE module for 60 Hz line noise removal, bad channel rejection and artifact removal using combined wavelet-enhanced independent component analysis (ICA) and Multiple Artifact Rejection Algorithm (MARA)[111]. Given the short length of EEG recording, 39 of the 128 channels were selected for ICA/MARA (Standard 10-20 electrodes: 22, 9, 33, 24, 11, 124, 122, 45, 36, 104, 108, 58, 52, 62, 92, 96, 70, 83; additional electrodes: 23, 28, 19, 4, 3, 117, 13, 112, 37, 55, 87, 41, 47, 46, 103, 98, 102, 75, 67, 77, 72, 71, 76). These electrodes were selected based on their spatial location, covering frontal, temporal, central, and posterior regions of interest (see Fig. 1). After artifact removal, channels removed during bad channel rejection were interpolated, data were re-referenced to an average reference, detrended using the signal mean, and segmented into 2-second segments. Any segments with retained artifact were rejected using HAPPE's amplitude and joint probability criteria. EEG signals were rejected for data quality if they had fewer than 10 segments (20 seconds total) or did not meet the following HAPPE data quality output parameters: percent good channels >80%, mean and median retained artifact probability <0.3, percent of independent components rejected <80%, and percent variance after artifact removal <32%.

**Adult study.** rsEEG data were collected while the subject sat independently in a chair situated in either a dimly lit, electrically shielded booth or a more brightly lit clinic room within the research center wherein care was taken to reduce electrical noise in the environment. At times, luminance levels in the rooms were adjusted for the comfort of the subject. Five minutes of continuous EEG were recorded while the participant was seated comfortably watching a silent video of their choosing on a battery-powered iPad, consistent with previous studies[19,20].

rsEEG was continuously recorded and digitized at 1000 Hz, filtered 0.01–200 Hz, referenced to Cz, and amplified 10,000x using a 128-channel saline Electrical Geodesics system (EGI, MagStim, Minnesota) with sensors placed approximately according to the International 10/10 system. Data were preprocessed using MATLAB (R2021b) and the Cincinnati Very High Throughput Pipeline (VHTP), which utilized EEGLAB 2021b[112].

rsEEG data used for the spectral power analysis were digitally filtered from 0.5 to 100 Hz with a 60 Hz notch (57–63 Hz; harmonics were removed up to the Nyquist frequency of the original sampling rate), channels with poor quality data were interpolated (no more than 5% of channels interpolated), segments of poor-quality data were manually rejected, and then remaining data were submitted to independent components analysis via EEGLAB for artifact removal. Artifacts (e.g., muscle tension, ocular-related events, heart rate, etc.) were removed as components and then data were average referenced and segmented into 2-second segments. During the creation of segments, data were submitted to an automatic amplitude rejection threshold where segments containing artifact exceeding +/−120 μV were automatically rejected.

For analysis of burst dynamics, data were also digitally filtered from 0.5 to 100 Hz with a 60 Hz notch (57–63 Hz; harmonics were

removed up to the Nyquist frequency of the original sampling rate), with no more than 5% of channels interpolated for bad data, but data did not undergo manual segment rejection. Instead, a section of continuous data was isolated starting from approximately 60 seconds into the recording through approximately 230 seconds into the recording, depending on the degree of artifact present. The continuous segment was then submitted to ICA for artifact correction, where removal of low-frequency artifacts (e.g., heart rate, ocular-related events, etc.) was prioritized due to the planned low-frequency filtering for burst analyses (i.e., high frequency noise was filtered out). Data were average referenced and then reduced to 90-second-long segments for analysis, as 90 seconds was approximately the longest segment of clean, continuous data available for analysis.

### Murine Subjects
*Fmr1*[−/y] mice were obtained from Jackson Laboratories, Maine, USA (stock # 003025) and backcrossed onto a C57BL/6 J background for at least six generations at Massachusetts Institute of Technology or King's College London. *Fmr1* cKO mice were also obtained from Jackson Labs (stock # 035184) and crossed with Emx1-Cre mice (Jackson stock # 005628). As previously reported, for hM4D(Gi) experiments, mice were Parvalbumin-Cre recombinase knock-in mice (B6;129P2-*Pvalb*[tm1(cre)Arbr]/J, PV-Cre) on a C57BL/6 background (Jackson stock # 017320)[71]. For optogenetic inhibition experiments, SOM-Cre mice were used (B6J.Cg-Ssttm2.1(cre)Zjh/MwarJ, Jackson stock # 028864). Experimental cohorts consisted of male littermates that were weaned at p21 and were p30–P150 at the time of experiments (n = 215 *Fmr1*[−/y] mice and littermate controls, p30–40 and p70–150, n = 44 Emx1-*Fmr1* KO mice and littermate controls, p30–40, n = 16 PV-Cre mice, p50–80, and n = 31 Som-Cre mice and littermate controls, p50–70). Mice were maintained on a 12-hour light-dark cycle (7 am – 7 pm) in a temperature- and humidity-controlled animal care facility (68–72° F and 30–70% humidity) with *ad lib* access to food and water and housed with 1–4 other littermates. All experiments were performed blind to genotype using age-matched littermate controls within the 12-hour light phase.

### Murine electrode implantation surgery
**L4 LFP electrodes.** Methods for LFP surgeries were followed as previously reported[65]. Mice (p30–150) were administered with preoperative analgesia (0.1 mg/kg Buprenex subcutaneously) and anesthetized with isoflurane (3% in oxygen at induction, 1.5% in oxygen through operation). The head was shaved and disinfected with povidone–iodine (10% w/v) and ethanol (70% v/v), the scalp was incised, and the skull surface was scored. A steel screw for head-fixation was bound to the front of the skull with cyanoacrylate glue, the skull was levelled, and a reference electrode (silver wire, A-M systems) was placed in the right frontal cortex. LFP tungsten recording electrodes (FHC), 75 μm in diameter at their widest point, were implanted in the binocular visual cortex (−3.5 mm bregma, +/− 3 mm midline, 450 μm depth). All electrodes were secured using cyanoacrylate and the skull was covered in dental cement. A nonsteroidal anti-inflammatory drug (meloxicam, 1.5 mg/kg) was delivered subcutaneously post-operatively for 3 consecutive days and mice were monitored daily for discomfort. Mice were given 48–96 hours to recover before being habituated to head-fixation.

**Cortical surface electrodes.** Using similar preoperative and surgical techniques (except use of carprofen as the pre-operative analgesia), *Fmr1*[−/y] and littermate WT mice (p70–115) were chronically implanted with screw-type electrodes (1.6 mm, Protech International) in the skull to measure EEG from the cortical surface (electrocorticography) in V1 and S1, and with the same type of LFP electrode as described above in V1. Screw electrodes were placed in burr holes over S1 (−2.06 mm bregma, 3.20 mm midline) and V1 (−3.50 mm bregma, −3 mm midline)

with a reference electrode and a ground over the left and right olfactory bulb. The coordinate for the V1 LFP electrode was the same as the skull screw (i.e., the same hemisphere) except the wire was inserted into the cortex tissue at a 10-degree angle and advanced in the dorsoventral axis to a depth of 0.52 mm from the cortical surface. Electrodes were connected to head-mounts and secured with dental cement and mice were given at least one week to recover before cortical surface EEG/depth LFP recordings.

**Murine resting-state EEG/LFP data collection and preprocessing**
Mice were habituated to the recording set-ups for at least two days. For habituation of head-fixed mice, the monitor was positioned 20 cm away from the animals and displayed a static, iso-luminant gray screen (80 lux luminance) generated using previously published code (https://github.com/jeffgavornik/VEPStimulusSuite) written in C ++ for interaction with a VSG2/2 card (Cambridge Research Systems) or MATLAB (MathWorks, R2012b) using the PsychToolbox extension (version 3.0.12, http://psychtoolbox.org). For resting-state LFP recordings in head-fixed mice, five minutes of data while the same monitor was turned off were collected in some of the mice, and three to five minutes of data while the monitor displayed the iso-luminant gray screen were collected in all mice. Data from head-fixed mice were collected either in a Faraday cage on a Plexon Recorder 64 system, with a PBX-211 2003 pre-amplifier (Plexon Inc, Dallas TX, USA) at MIT or on a Plexon OmniPlex® Neural Recording System, with Plexon's DigiAmp™ acquisition system (Plexon Inc, Dallas, TX, USA) at King's College London. At both institutions, LFP data were collected from the electrode placed in V1 (left or right hemisphere) and forepaw movement data were collected from a piezoelectric disk (C.B. Gitty) placed under the animals' forepaws; data in total were collected from three channels (the V1 channel, the ground in frontal cortex, and the piezo). LFP voltage data were sampled at 1 kHz. At the UK site, data were high-pass filtered at 0.5 Hz and low-pass filtered at 500 Hz. At the US site, data were band-passed through analog filtering via a low cut, 2-pole 3 Hz filter and a high-cut, 4-pole, 8 kHz filter, and frequencies down to 0.4 Hz were digitally recovered using the inverse transfer function for the Linkwitz-Riley analog filter obtained from Plexon. This and all other preprocessing were conducted in MATLAB (R2018b). Data from the V1 LFP channel were loaded into MATLAB, unfiltered, converted to microvolts, and detrended using the signal mean. A second-order IIR notch filter was applied with a bandwidth of 0.01 (for gray screen data from juvenile animals, collected at MIT), 0.05 (for gray screen data from adult animals, collected at King's College and MIT), or 0.025 (for black screen data from all animals, collected at MIT) to remove 50 or 60 Hz line noise while preserving underlying biological (i.e., luminance processing) signals around 60 Hz. Data from the Piezo channel were also loaded into MATLAB and rectified.

For cortical surface EEG recordings (electrocorticography) in freely-moving mice, the animals were tethered to four-channel EEG/EMG recording systems (Pinnacle Technology Inc.) and housed individually and sequentially in a sound-proof and light-controlled cabinet (standard lighting conditions, 200 lux luminance) equipped with a video camera. Voltage data were acquired continuously for a 10–12-hour period with a sampling rate of 1 kHz, amplified 100x, and band-passed filtered through analog filtering via a two-series, single pole simple RC filter at 1 Hz and an eighth-order elliptical filter at 100 Hz. Frequencies down to 0.6 Hz were recovered using the inverse-transfer function for the passive RC filter obtained from Pinnacle and implemented in MATLAB (R2018b). 100-second-long resting state time series from S1 and V1 electrodes in each mouse were prepared using the synchronous video to identify when mice were in a quiet wake (resting) vigilance state (i.e., sitting at the corner of the recording cage, moving the head but not the whole body, no running). These time-series data were loaded into MATLAB, unfiltered, converted to microvolts, and detrended using the signal mean. A second-order IIR notch filter with a

bandwidth of 0.15 was applied to remove 50 Hz line noise while preserving underlying biological signals.

**PV+ inactivation methods**
As described in Kaplan et al. (2016) for hM4D(Gi) experiments, in addition to implanting LFP electrodes, V1 of p30–60 PV-Cre mice was infected with an AAV9-hSyn-DIO-HA-hM4D(Gi)-IRES-mCitrine virus (UNC viral core – generated by Dr. Brian Roth's laboratory)[72]. Using a glass pipette and Nanoject system (Drummond Scientific, Broomall, PA, US), 81 nL of virus was delivered at each of 3 cortical depths: 600, 450, and 300 μm from the cortical surface, with 5 minutes between repositioning for each depth. Mice were allowed 3–4 weeks of recovery for virus expression to peak before experiments. Then, after two days of habituation, 5 minutes of resting-state LFP data with mice head-fixed in front of a static, iso-luminant gray screen were collected before and 30 minutes after Clozapine-N-oxide (CNO, Enzo Life Sciences) was diluted in saline and injected intraperitoneally at a dosage of 5 mg/kg. For these data, an aggressive 60 Hz notch filter was incorporated into the Plexon recording software, so no further digital notch filtering was applied during data preprocessing and frequencies between 45–75 Hz were not analyzed.

**SOM+ optogenetic inactivation and activation methods**
For optogenetic experiments, we infected V1 of p30–50 SOM-Cre mice with either pAAV-Ef1a-DIO eNpHR 3.0-EYFP (AAV5) virus (for inactivation), pAAV-EF1a-double floxed-hChR2(H134R)-EYFP-WPRE-HFHpA (AAV5) virus (for activation), or the empty viral vector pAAV-Ef1a-DIO EYFP (AAV5). Using the aforementioned Nanoject system, we delivered 100 nL of virus at 2 nL/s at 4 cortical depths: 600, 450, 300 and 150 μm from the cortical surface, with 6 minutes between re-positioning for each depth. In addition to implanting LFP electrodes and infusing the virus, ready-made optic fibers (200 μm girth) mounted in stainless steel optic cannulas (1.25 mm diameter, 2 mm fiber projection, Thor labs, CFMLC12L02, Newton, NJ, US) were then implanted lateral to the recording site and at a 22° angle to the recording electrode, 0.1 mm below surface in each hemisphere. 3 weeks later, mice were habituated for 2 days. For resting-state LFP data collection during SOM+ inactivation, while mice were head-fixed in front of a static gray screen, we delivered five 10-second long continuous pulses of green light (550 nm, 6–7 mW) into V1 using a digitally-controlled LED driver (PlexBright® Optogenetic Stimulation System). Each pulse was separated by 60 seconds, resulting in a recording session that lasted about 7 minutes. For SOM+ activation, we delivered three 100-second long continuous pulses of blue light (465 nm, 1.5 mW) with each pulse separated by 30 seconds, resulting in a recording session that also lasted about 7 minutes. For these data, notch filtering was not activated in the Plexon software, so a second-order IIR notch filter with a bandwidth of 0.2 was applied in MATLAB (R2018b) during data preprocessing to remove 50 Hz line noise.

**Drug preparation**
Following established protocols, Arbaclofen (R−4-amino-3-(4-chlorophenyl)butanoic acid) was dissolved in 0.9% saline (NaCl) and administered via an intraperitoneal injection at 0.5, 1, or 3 mg/kg in an injection volume of 0.1 mg/mL[87]. The vehicle was saline. Injections occurred 60–90 minutes before gray screen presentation and LFP data collection. Vehicle and Arbaclofen were tested on the same day, with the vehicle administered first and then Arbaclofen several hours later. Thus, the experimenter was blinded to genotype but not treatment.

CTEP (2-chloro-4-((2,5-dimethyl-1-(4-(trifluoromethoxy)phenyl) −1H-imidazol-4-yl)ethynyl)pyridine) solution was prepared according to established protocols[91]. Briefly, CTEP was synthesized at Roche and micro-suspended in vehicle (0.9% NaCl, 0.3% Tween-80) and administered via an intraperitoneal injection at 2 mg/kg in an injection volume of 0.2 mg/mL. The vehicle was a sterile solution of 10%

dimethyl sulfoxide (DMSO), 45% polyethylene glycol (PEG-400), and 45% artificial cerebrospinal fluid (aCSF). Injections occurred 90 minutes before gray screen presentation and LFP data collection. A crossover design was used and both *Fmr1*[-/y] and wildtype littermates in a pair received matching treatments on the same recording sessions, with 3–7 days between treatments. Thus, the experimenter was blinded to genotype and treatment.

**Cross-species resting-state EEG/LFP analysis**
**Spectral analysis.** Spectral analyses were performed in MATLAB (R2018b) using the Chronux toolbox (version 2.12)[113]. For all human subjects, the power spectral density (PSD) at each electrode, for each 2 second segment, was calculated with a multi-taper spectral analysis, using 3 orthogonal tapers and a time-bandwidth product (*TW*) of 2. PSDs had interpolated 0.1 Hz bin sizes for pediatric subjects and noninterpolated (i.e., equivalent to the frequency resolution) 0.5 Hz bin sizes for adult subjects. PSDs were averaged across segments and then averaged across electrodes. The following electrodes were averaged for each region of interest (ROI): frontal – 3, 4, 11, 19, 23, 24, 28, 117, 124; central – 13, 36, 37, 55, 87, 104, 112; temporal – 41, 45, 47, 46, 52, 58, 96, 98, 102, 103, 108; posterior – 67, 70, 71, 72, 75, 76, 77, 83. All listed electrodes from each ROI were averaged for the full brain absolute power spectrum.

For source modeling in human subjects, absolute power within the range of periodic pk1 (4–14 Hz) was averaged within group and submitted to Brainstorm for source localization in an approximation of previously published methods[19]. A head model was generated where EGI 128 channel locations were co-registered with the Montreal Neurological Institute (MNI) averaged ICBM152 common brain template (default anatomy), OpenM/EEG was used to compute a lead-field mesh with 15,002 vertices, and noise covariance was set to an identity matrix. Sources were then computed where a minimum norm estimate model was generated via current source density with dipole orientation constrained as normal to cortex.

For freely-moving murine subjects, the 100-second resting-state time series was segmented into 5 second segments, resulting in 20 segments per subject. Longer segments were needed for the mice as the periodic signals of interest were at lower frequencies. This yielded PSDs with 0.2 Hz bin sizes, equivalent to the frequency resolution. The appropriate time-bandwidth product (*TW*) was calculated as $TW = (FR*segment\ length)/2$[114] to achieve a consistent spectral resolution (*FR*) with human data (2 Hz). Multi-taper spectra were calculated with 9 tapers and a time-bandwidth product of 5, with no zero padding. For head-fixed murine subjects with L4 LFP electrodes, 150-seconds of resting-state and piezo data while the monitor was off or displaying a static gray screen were extracted for spectral analysis (except for SOM-Cre mice, where there were only 50 seconds of 'laser on' data, and for mice in the longitudinal experiment, where only 120 seconds of resting-state data were collected). 5-second segments were again used, but due to occasional contamination of the piezo signal into the LFP signal during certain movement bouts, a very small fraction of segments (usually 0–2 per mouse) were eliminated if they contained both a large movement artifact and a huge spike in low-frequency power (less than 1.5 Hz). As a result, 28–30 segments per animal per condition were usually analyzed (8–10 segments for SOM-Cre mice and 22–24 segments for mice in the longitudinal study). The multi-taper absolute spectra were calculated using 9 tapers and a time-bandwidth product of 5, as before. For additional analysis of 2–10 Hz data (periodic Pk1 subpeak parameters), multi-taper absolute spectra were recalculated with a higher spectral resolution (1 Hz) by using 4 tapers and a time-bandwidth product of 2.5.

**Modified one over F fitting analysis**
An effective way to model the 1/f signal of the absolute power spectrum across a variety of datasets is to use a Lorentzian function[49].

Published code (SpecParam, https://fooof-tools.github.io/fooof/) allows users to fit a Lorentzian to a spectrum, optimized for selection of a fitting range with no periodic power at the edges of the range[49]. Realistically, selecting such a fitting range is challenging with biological data, and indeed, the implementation of the Lorentzian fit through SpecParam does not work well for pediatric rsEEG data[50]. The algorithm cut through the absolute spectrum of the pediatric data in the middle of the fitting range, generating negative periodic power between 10–20 Hz, which is not theoretically possible. This issue was resolved previously by shifting the fit downwards (changing the y-intercept but not the slope) until there was no more negative periodic power[50]. In the present study, we encountered multiple issues of overestimated aperiodic power when using SpecParam to fit the 1/f component of both human EEG data of all ages and mouse LFP data, especially at the bottom edge of the fitting ranges and in the 10–20 Hz range (Supplementary Fig. 1b,5c). We ran SpecParam (version 1.1.1) in MATLAB R2018b using the MATLAB wrapper (https://github.com/fooof-tools/fooof_mat) and used the following settings: peak width = [0.5 18], number of peaks = 7, peak threshold = 1, and fixed mode for EEG data and knee mode for LFP data. Since our L4 LFP data in mice did not have a linear aperiodic component (Supplementary Fig. 5b, c), the fits could not be uniformly adjusted in the same way as before.

Therefore, we wrote custom code in MATLAB (R2018b) to fit Lorentzian functions to the 1/f signals effectively and consistently across all our datasets. Our implementation required the user to have confidence that only one, not both edges of their selected fitting range be without periodic power, which is more realistic given limitations of experimental datasets (such as artifacts, filters, etc.). The first step, as outlined previously, was to plot the average absolute power spectrum per group in log-log space[49]. This allowed us to identify a good fitting range and to decide if the 1/f fit would be linear or nonlinear (which affected the fitting procedure but maintained consistent application of math and methodology). Our fitting is performed in log-log space on the average absolute spectrum for each subject. Our approach is more data-driven than SpecParam, but this minimized negative periodic power in a consistent way across all datasets.

For data requiring a linear fit in log-log space, which were all data recorded outside of cortex (human and mouse EEG), the bottom edge of the fitting range was determined by the lowest frequency unaffected by high pass filtering (3 Hz for humans, 2 Hz for mice). The [x,y] coordinates for the first point of the fit were [edge of fitting range, average power of the lowest 0.6 Hz of the fitting range]. Next, the algorithm identified the frequency value above 8 Hz that yielded the largest slope magnitude for the fit, as shallower slopes would cut through the absolute spectrum and generate negative periodic power. The slope and intercept were calculated using this frequency value (x-coordinate) and the average power at that frequency and at the frequencies on either side of it (y-coordinate). The offset was calculated as the aperiodic power at the edge of the fitting range (2 or 3 Hz). The linear Lorentzian was then subtracted from the data, revealing the periodic spectrum. To analyze the periodic peaks for human data, we found the maximum values of the periodic spectra less than (Pk1) and greater than (Pk2) 15 Hz. The x-coordinate of these maximum values was the center frequency, and the y-coordinate was the maximum power of the peak. For mouse cortical surface EEG data, there was only one periodic peak, which was identified as the maximum value below 10 Hz through a locally-weighted smoothing linear regression filter.

For data with a non-liner 1/f signal in log-log space, which was all the L4 LFP electrode data, an additional knee parameter was needed in the Lorentzian function. Our general approach for fitting the knee was consistent with SpecParam, where we first fit a linear aperiodic component within the frequency range above the knee and then fit the knee to the low-frequency values below the inflection point[49]. Thus, we needed to fit our initial linear 1/f estimate based on the upper edge of the frequency range, which meant it was critical to select an upper

point with no periodic power. This required fitting out to 175 Hz, as there was still periodic power (especially in *Fmr1$^{-/y}$* mice) through about 160 Hz. The frequency value ($x_1$) with the lowest absolute power (ignoring the 45–75 Hz range if there was notch filtering) and the average power of it and its neighboring two frequency values ($y_1$) were used to form the first coordinate pair for the linear aperiodic estimate. Then, the algorithm identified the frequency value between 9 and 25 Hz (one of the problematic ranges for SpecParam) that yielded the smallest magnitude slope of the linear fit, as steeper slope values would cut through the absolute spectrum and generate negative power. The second coordinate pair for linear estimate was determined from this frequency value ($x_2$) and the average power at that frequency and at the frequencies on either side of it ($y_2$). The slope and intercept values of the Lorentzian were ascertained through this linear estimate. To determine the knee, we found the local minima power values at frequencies between 1.5 Hz (the edge of the fitting range) and 2.24 Hz (below the first periodic peak), and, assuming these values were less than the linear 1/f estimate at this frequency, used them to calculate multiple knee value estimates according to the Lorentzian function: $knee = 10^{b \cdot y} \cdot x^m$, where ($x,y$) are the (frequency, power) coordinates of the spectrum, $m$ is the slope, and $b$ is the intercept. The final knee parameter was calculated as the average of these estimates, and the offset was calculated as the aperiodic power at the edge of the fitting range (1.5 Hz). With the full equation for the non-linear 1/f, we then subtracted the fit from the absolute spectrum to analyze the periodic spectrum. To analyze the subpeaks of periodic Pk1 (2–10 Hz), we found the largest local minimum of the data in the middle of this range using a locally-weighted smoothing linear regression filter. Pk1a was either the maximum value of the periodic spectrum below the local minimum or the maximum value below 6 Hz, and Pk1b was the maximum value above the local minimum or 6 Hz. This automated selection was confirmed manually for each subpeak, with manual selection used in cases of poor automated detection.

## Phase-amplitude coupling analysis

Following previously published methods, modulation index (MI) for head-fixed murine LFP data was calculated from the probability distribution of the amplitude signal across 18 bins of phases[60,115]. Signals were band-passed using the *eegfilt* function in MATLAB (R2018b) designed by Scott Makeig (https://sccn.ucsd.edu/~arno/eeglab/auto/eegfilt.html, copyright 1997) and then the Hilbert transform was calculated to find the amplitude or phase values from the analytical signals. To generate comodulagrams, the phase for each 5-second epoch was analyzed between 2–9 Hz in step sizes of one and with a bandpass bandwidth of two, while the amplitude was analyzed between 10–60 Hz with a step size of two and a bandpass bandwidth of four, resulting in 6 low-frequency phase signals and 26 high-frequency amplitude signals. For each combination of low-frequency phase and high-frequency amplitude signals, the average amplitude across all phases within each of the 18 bins was normalized to the sum of all mean amplitudes across all bins, yielding the phase-amplitude distribution, *PA*. The MI was calculated as follows: $MI = \frac{\log(N) - (-\sum_{j=1}^{N} PA_j \log(PA_j))}{\log(N)}$, where $N$ = the number of phase bins (18). Noise comodulograms were generated by shuffling trials, correlating the low-frequency phase signal of one 5-second epoch with the high-frequency amplitude signal of a different 5-second epoch. MIs were noise-subtracted using a bootstrapping procedure described in the **Statistics** section below. For PA plots, the phase was calculated from the 4–6 Hz analytical signal and the amplitude was calculated from the 15–40 Hz analytical signal for each 5-second epoch.

## Burst analysis

**Murine subjects.** To analyze the temporal dynamics of periodic Pk1, the same bandpass filter and Hilbert transform methodology were used,

filtering 100 seconds of continuous head-fixed LFP data between 2–10 Hz and freely-moving EEG/LFP data between 3–8 Hz (the steeper aperiodic slope required a tighter range). If more than 100 seconds of continuous head-fixed LFP data were available, the starting point of the 100-second chunk was randomly generated. Head-fixed mice recorded on the piezo disk that did not have at least 100 seconds of continuous data free of large motion artifacts that spiked low frequency power (see **Spectral analysis** section) were excluded from this analysis. Following published methods, a burst threshold was set at the 90th percentile of the amplitude of the analytical signal for each animal[56]. Burst count was quantified as the number of threshold crossings, and burst duration was the length of time spent above the threshold in one crossing; a threshold crossing had to last for more than 3 milliseconds to count as a burst[56]. For animals with piezo data, movement bouts were identified where the piezoelectric signal exceeded a voltage of 2 mV (standard threshold across all animals). To compensate for voltage drops below 2 mV in the middle of movement bouts, a buffer threshold of 750 milliseconds was used. The piezo voltage signal was subsequently converted to binary movement state categorization (1 = moving, 0 = still) and aligned to the analytical signal to determine if bursts occurred during intervals of movement or quiescence. The sum of this binary signal, normalized to the length of the analytical signal (100 seconds) times the sampling frequency, was used to determine total % time spent moving. For both murine and human data, spectrograms were generated using the Chronux toolbox with 2 tapers, no zero-padding, 3-second moving windows, and 750-millisecond step sizes.

**Human subjects.** Burst counts and lengths were calculated from the same selection of occipital channels used for the spectral power analysis, but using the 90 seconds of continuous (non-segmented) data. Each individual channel was filtered according to a frequency range determined in a data-driven manner from the analysis of Pk1 in the periodic spectrum. Given the shift in center frequency of Pk1 for FXS subjects, data were filtered from 5–10 Hz for FXS and 7–12 Hz for TD. Filtered data were Hilbert-transformed, and the amplitude of the analytical signal was squared and log-transformed to express in units of power. As with the mice, the 90th percentile of the power for each channel was used as the threshold to calculate endogenous burst counts and burst lengths. Individual channel burst metrics were averaged across channels to provide the average burst dynamics over the occipital region of the head.

## Statistics

Statistics were conducted in MATLAB (R2022b). Initial statistical tests using the one-sample, two-sided Kolmogorov-Smirnov test (MATLAB function *kstest*) confirmed non-normal distributions of the various datasets. Thus, statistical tests were conducted using methods that did not assume normality, including the two-sample, two-sided Wilcoxon Rank-Sum test (also known as the Mann-Whitney U-test, MATLAB function *ranksum*), the two-sample, two-sided Wilcoxon Signed-Rank text (for paired comparisons pre/post drug within genotype, MATLAB function *signrank*), and non-parametric hierarchical bootstrapping[65,116]. The Wilcoxon tests were used to test for group differences for metrics that could be well-summarized by a single (or average) value per animal/subject (i.e., aperiodic parameters, periodic peak parameters, microburst count), while bootstrapping was used for more complicated datasets where a single-value summary failed to capture the variability in the data (i.e., power spectra, comodulograms, phase-amplitude distributions).

The output of the MATLAB *ranksum* and *signrank* functions yielded a rank-sum or sign-rank statistic, a z-statistic for larger sample sizes (greater than 10 for *ranksum*, and greater than 15 for *signrank*), and a *p*-value. For all Wilcoxon tests, the *p*-values were corrected using the False Discovery Rate (FDR) Benjamini-Hochberg correction when multiple comparisons were made per plot. These non-parametric

Wilcoxon tests do not have degrees of freedom. Effect sizes were determined for sample sizes of 10 or more using Cliff's Delta (MATLAB function *meanEffectsize*, with either paired or unpaired comparisons based on the type of Wilcoxon test used), yielding a value between −1 and 1 (where values >= 0.3 or <= −0.3 indicated a medium to large effect size) and confidence intervals (CIs).

CIs for power spectra were assessed through non-parametric hierarchical bootstrapping. Code for bootstrapping was written following published methods[65,116]. For bootstrapping absolute power spectra in human adults and for all sub-strains of mice, a random subset of subjects within each group or genotype equal to the total sample size was selected with replacement, and for each subject, a random subset of epochs equal to the total number of epochs for that subject was selected with replacement. The average power spectrum was tabulated for each genotype from these random subsets of subjects and epochs, and the difference between genotypes (FX or KO − TD or WT) was stored as the first bootstrapped difference spectrum. The process was repeated 1000 times, and for each frequency in the spectrum, the values were rank ordered and the 5th and 995th values (for 99% CIs, used for murine data) or 25th and 975th values (for 95% CIs, used for human data) were stored along with the median. For each frequency, if the lowest (5th or 25th) value of the CI of the bootstrapped difference was greater than 0, or if the largest (995th or 975th) value of the CI was less than 0, the CI did not include zero and we report a statistically significant difference between genotypes with a small dot below the corresponding frequency on the power spectra plot. For bootstrapping absolute power spectra in human children and aperiodic and periodic spectra for all species, there was only one interpolated or fitted spectrum per subject, so only one level of hierarchical bootstrapping was performed (a random subset of subjects within each genotype equal to the total sample size was selected with replacement). For all power spectra, to utilize littermate pair comparisons, the random selection of subjects (and epochs, if relevant) was matched between the distributions of TD and FXS human subjects and between WT and *Fmr1*[−/y] mice when their sample sizes (and epoch counts, if relevant) were equal. This matching was always used for within-subject matched-pair comparisons of periodic and aperiodic spectra before and after a drug.

For phase-amplitude coupling statistical comparisons in WT and *Fmr1*[−/y] mice, the generation of bootstrapped 99% CIs for PA plots across the 18 phase bins followed the same procedure as described above for bootstrapping murine absolute power spectra across the frequency bins. The procedure for generating noise-subtracted comodulograms differed slightly. As before, a random subset of subjects within each genotype equal to the total sample size was selected with replacement, and for each subject, a random subset of epochs equal to the total number of epochs for that subject was selected with replacement to generate each of 1000 average comodulograms per genotype. Additionally, a second random subset of epochs equal to the total number of epochs for that subject was selected with replacement (with each epoch value in the second list never matching the corresponding epoch value in the first list) to generate each of 1000 average noise comodulograms per genotype. The difference between the 1000 comodulograms and noise comodulograms was tabulated for each genotype, and the resulting noise-subtracted comodulograms values were rank ordered, with the median, 5th, and 995th values stored (for 99% CIs). Median noise-subtracted comodulograms were plotted (Fig. 5a, b left). MI values in the noise-subtracted comodulograms where the 99% CI included zero (i.e., the MI was not significantly greater than noise) were set to 0 in the median noise-subtracted comodulograms and the plots were replotted (Fig. 5a, b right).

Finally, for measurement of the linear correlations in human EEG data, Pearson's $r^2$ (which also does not assume normality) was determined by fitting a line to the data (MATLAB function *polyfit*) and then dividing the sum of the squared residuals by the total sum of squares

(i.e., the variance times the sample size minus one) and subtracting that value from one. The effect size (Pearson's r) was determined by taking the square root of $r^2$, yielding a value between 0 and 1, where values <= 0.1 were considered a small effect size.

### Reporting summary

Further information on research design is available in the Nature Portfolio Reporting Summary linked to this article.

### Data availability

Preprocessed resting-state data from mice used to produce the figures in this manuscript are available on Figshare under the accession code (https://doi.org/10.6084/m9.figshare.29940884)[117]. Resting-state data are only a portion of the raw Plexon and Pinnacle murine data files, which are available upon request from the authors. Please contact the corresponding authors with such requests; the expected timeframe for response is one week. For data from human subjects, consents obtained from participants prohibit the sharing of identifiable or de-identified individual data without data use agreements in place. The current study includes data collected from several independent projects, including some in which consent for open sharing was not collected from the subjects. Thus, our ability to share data from human subjects depends on a variety of factors: which specific data are requested; whether and to what extent the participants included in the requested data have consented to the sharing and future use of their data; whether deidentified, limited, or identified data are requested; and the purpose for which the data are requested. Prior to sharing data, we may need to confirm whether Cincinnati Children's Hospital and Medical Center or Boston Children's Hospital and the institution of the individual requesting the data have existing agreements or sub-contracts with terms of data use and sharing that define the collaboration (e.g., data use agreements, data access agreements, subawards, reliance agreements). Please contact the corresponding authors with data requests to determine availability of the specific data of interest. If data use or data access agreements are needed, approval between institutions can take at least 2-3 months. Once approved, data is shared for the length of the agreement (usually 1 year, with opportunity for renewal). Source data are provided with this paper.

### Code availability

Analysis code is available on GitHub under the accession code: (https://doi.org/10.5281/zenodo.18201252)[118].

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

## Acknowledgements

The authors thank R. Komorowski, M. Fai-Fong, and A. Heynen for their contributions to early experiments in *Fmr1^{-/y}* mice and D. Stoppel for his expertise in and management of various *Fmr1* KO mouse lines. The authors also thank L. DeStefano for feedback on the project, J. Gavornik for creating the visual stimulus delivery code, and A. Palisano for administrative support. This work was supported by the following grants: NIH 1K23DC017983-01A1 (C.L.W.), NIH 1T32MH112510 (C.L.W.), U54HD082008 (C.A.E.), U54HD104461 (C.A.E), NIH K23 HD101416 (L.M.S), NIH F31 HD113221-01A1 (J.E.N.), NIH R01EY023037 (M.F.B.), NIH R21NS123499 (M.F.B.), NSF GRFP (S.S.K-S.), FRAXA Foundation (S.S.K-S. and C.L.W), Autism Science Foundation (C.L.W.), The Pierce Family Fragile X Foundation (C.L.W.), Thrasher Research Fund (C.L.W.), Harvard Catalyst Medical Research Investigator Training Award (C.L.W.), Simons Foundation SFARI 575135 (M.F.B.), the Picower Institute Innovation Fund (M.F.B) Wellcome (207727/Z/17/Z)(S.F.C) and the Biotechnology and Biological Sciences Research Council (BBSRC) (BB/S008276/1)(S.F.C).

## Author contributions

S.S.K-S. and M.F.B. conceived the project, secured funding with P.S.B.F., and designed the experiments with input from P.S.B.F. and M.J.H. C.L.W. secured funding and collected and preprocessed all pediatric EEG data with assistance from M.K., and helped conceive cross-species analyses. L.M.S. and C.A.E. secured funding and collected all adult EEG data, which J.E.N. preprocessed in multiple ways and analyzed for burst dynamics and source localization with input from L.E.E. S.F.C. secured funding and C.G. collected all freely-moving mouse EEG data with input from S.F.C. F.A.C. collected all optogenetics LFP data with input from S.F.C. P.S.B.F. and M.L. collected some head-fixed mouse data, including the longitudinal and CTEP studies. S.S.K-S. collected the rest of the head-fixed resting-state LFP data, conceived and wrote the MATLAB code for all cross-species spectral analyses and analyzed all mouse and human data (after it was preprocessed). S.S.K-S. interpreted the data and wrote the manuscript with review and editing from M.F.B, C.L.W., C.A.E., L.E.E., J.E.N., C.G., P.S.B.F, L.M.S., and S.F.C.

## Competing interests

M.F.B. is a co-founder of Allos Pharma, developing Arbaclofen for the treatment of Fragile X syndrome. The remaining authors declare no competing interests.
