## [Transparent Peer Review file · Nature Communications]

A human electrophysiological signature of Fragile X pathophysiology is shared in V1 of Fmr1-/- mice

Corresponding Author: Professor Mark F. Bear

Version 0:

Reviewer comments:

Reviewer #1

(Remarks to the Author)

This manuscript by Kornfeld-Sylla and colleagues reports neurophysiological results obtained from human FXS subjects and the Fmr1-/- mouse model. The authors compare EEG recordings in human subjects and mice, aiming to find common altered patterns to be used as biomarkers for the disease. The authors reason that this would allow for circuit dissection in the Fmr1-/- mouse model. The authors find some EEG and LFP alterations in mouse V1, and they correlate them with human EEG alterations. They go on and attempt to define the role of PV and SOM interneurons and excitatory pyramidal neurons in the emergence of LFP alterations in FXS mice. Finally, the authors analyze alterations in VEPs and visual plasticity in control and Fmr1-/- mice.

The topic is of high interest, and the manuscript contains high-quality data. However I feel that in the present form, the paper includes a series of orthogonal observations and falls short in providing a cohesive narrative and a clear solid message. The authors provide data from EEG from mice and humans, LFP analysis in 'resting state' and in response to visual stimulations in L4 of V1, analysis of visual evoked potentials and their plasticity only in mice. This is intermingled with an ambiguous and inconclusive comparison of different cell types that raises more questions than answers. In addition, even the individual sections of the paper often lack a clear conclusion and, when present, the interpretation requires some additional evidence to be fully convincing. The overall impression I had is that the end of the paper does not relate with the initial case that was built in the introduction.

Major issues:

1. If the authors wish to provide an overarching comparison between human FXS subjects and the mouse model of this condition, they should compare most spontaneous and visually evoked alterations in both organisms, when this is feasible.
2. The experiments on cell-type dissection are confusing. Silencing PV and SOM cells can lead to non-controllable network alterations that preclude the interpretation of the results. Moreover, the authors do not perform this analysis on wt and Fmr1-/- mice, so their conclusions are correlative at best. Importantly the silencing methods (pharmaco- vs. optogenetic silencing) present profound temporal controls, adding more confusing method-dependent alterations. I wonder whether one can compare pharmacogenetic silencing of PV cells, optogenetic silencing of SOM cells and genetic removal of FRMP in glutamatergic neurons and glial cells.
3. The authors should be cautious in comparing burst activity in humans and mice, as it seems that enhanced bursts occur in FXS patients only during tasks, whereas here the authors analyze bursts in mice only during resting states. Are these microbursts the same? The authors do not record burst activity during resting state in their human samples, raising a potential problem in comparing the two conditions in the two organisms.
4. The experiments of Fig. 8 are difficult to follow. Why are only KO mice shown in panels B and F? KO data is usually compared to their wt littermates, otherwise it becomes anecdotal. In panel C, why are VEPs lacking the P1 peak? What is the interpretation of the results illustrated in panel K? Overall, the SRP results are not even discussed. These results seem preliminary, and they do not relate with the alterations of the alpha phenotype described in Figs. 1-5, which represents the only associated element with the human disease.

(Remarks on code availability)

Reviewer #2

(Remarks to the Author)

(Remarks on code availability)

Reviewer #3

(Remarks to the Author)

This manuscript reports a translational electrophysiological biomarker linking alpha-frequency slowing in humans with homologous Pk1a deceleration in Fmr1 KO mice. The multimodal convergence—human EEG, mouse VEPs, and intracortical LFPs—is compelling and methodologically sophisticated. The study also attempts to assign mechanistic specificity to cortical excitatory neurons using Emx1-Cre conditional knockouts.

However, the central causal claim—that this biomarker is caused by loss of FMRP in excitatory neurons (as seen in the title)—goes beyond what the current data can justify. The Emx1-Cre model demonstrates sufficiency but not necessity, as no selective rescue experiment is performed. Developmental confounds and indirect network effects remain plausible explanations. Without demonstrating that restoring FMRP in excitatory neurons abolishes the biomarker, the causal inference is incomplete and should be reflected within the manuscript and title.

Behavioral state discrepancies (e.g., arousal, visual input) between humans and mice are not explicitly controlled or covaried and are known to be strongly associated with alpha states. This should be more clearly documented and raised as limitation.

Cross-species translation is thoughtfully approached but weakened by anatomical and behavioral mismatch. Human alpha is recorded at the scalp without source localization, whereas mouse data come from layer 4 V1. Human EEG reflects a blend of signals from various cortical areas, not just V1 in contrast to the murine recordings. To a degree, the strong translational claim of functional homology between V1 and human occipital cortex is hinged on this argument. A source-reconstruction analysis (e.g., beamforming or CSD) is needed to confirm anatomical homology. At minimum, if source analysis is not performed than interpretation should be tempered on circuit level homology and speak more broadly to contributions from parietal, frontal, and non-visual regions.

Statistical methodology is rigorous but under-documented in places. Multiple spectral comparisons per figure require formal correction (e.g., FDR). Even if bootstrapping is used, it is not typically a reason to avoid multiple comparison correction since that is determined by the number of tests performed. Additional reporting on experimental blinding, randomization, and replication strategies—especially for opto- and chemogenetic conditions.

The biomarker's specificity is unproven, as similar alpha-frequency slowing has been documented in other neurodevelopmental and psychiatric disorders, including autism spectrum disorder (ASD), Down syndrome, and first-episode psychosis. This raises the possibility that alpha slowing is a non-specific marker of global neural dysfunction rather than a signature unique to Fragile X Syndrome. Without comparative data or test-retest reliability in FXS, its translational utility remains uncertain, claims of a biomarker should be tempered and discussed in the limitations.

Despite these critiques, your manuscript represents an ambitious and highly promising contribution. With careful revision addressing these points, your work can set a new standard for rigor in translational biomarker research. I look forward to seeing a revised version that realizes the full impact and potential of your findings.

(Remarks on code availability)

Datasets are available on publication, human data is on request

Reviewer #4

(Remarks to the Author)

This study identifies an electrophysiological biomarker of Fragile X Syndrome (FXS)—a shift in alpha oscillations—that is present in both human patients and a mouse model (Fmr1^{-/-} mice). Specifically, the researchers found that children and adults with FXS exhibit a consistent slowing of alpha rhythms in resting-state EEG, and this same pattern is mirrored by a corresponding shift in low-frequency oscillations in the visual cortex (V1) of Fmr1 knockout mice. The authors show that although both PV⁺ and SOM⁺ interneurons modulate different aspects of the affected oscillatory activity in mice (although not the frequency shift), it originates from a loss of the FMRP protein in cortical excitatory neurons, not inhibitory neurons. The study further links these oscillatory disruptions to altered sensory responses. Overall, this work provides a useful cross-species EEG analysis and some mechanistic insight into the pathophysiology of Fmr1^{-/-} mice.

Major comments:

1. The developmental trajectory of oscillatory changes differs between mice and humans – i.e., the alpha frequency decreases in adulthood in mice, whereas in humans it stagnates. This weakens the model's predictive power for developmental timing in translational research. Could the authors please address this? Also, could the authors please comment on the developmental trajectory of this alpha frequency phenotype in mice in their data, from juvenile to adulthood? Are the developmental differences only observed under luminescence conditions?
2. The alpha frequency changes in humans were consistent across different neocortical regions, whereas in the mouse model, they were observed in V1 but not S1. Do the authors have an explanation for this? How would this affect their interpretation as a translational biomarker? Unless the authors have a common function related to alpha frequency oscillations in V1, it will be difficult to reconcile these findings in a translational manner. Can the alpha phenotype be reversed or normalized by therapeutics? To be useful in clinical trials, the biomarker must respond to treatment. Does pharmacological treatment or re-expression of FMRP, or targeted V1 circuit manipulation, reverse the phenotype? And could this correct, for instance, the N1 component of the VEP? I think it would be important to demonstrate the link between alpha and VEP or another physiological readout to strengthen the relevance of alpha oscillations in V1 as a biomarker.
3. What is the exact mechanism driving the alpha frequency shift in FXS? While PV⁺ and SOM⁺ interneurons were shown to modulate different subcomponents of the alpha-like oscillations, neither accounted for the frequency shift – the core 'alpha phenotype.' This suggests the proposed mechanism is incomplete and may involve other mechanisms or circuits, such as the thalamocortical loop. In light of this, the interneuron modulation experiments seem insufficient to address the mechanism underlying the alpha frequency shift.
4. Although alpha oscillations are proposed as a translational biomarker, spectral power and frequency measures are influenced by many biological and technical variables, such as brain region, recording setup, and luminance, amongst others. It may be difficult to generalize the biomarker for clinical use or across labs without standardization. Could the authors please clarify whether, and if so, how they standardized the recordings across labs? Do they have data from either lab to support the developmental trajectory of their alpha wave findings?
5. The authors go into some detail to describe the neurobiological underpinnings of alpha frequency wave alterations in mice by focusing on layer 4. Some critical findings (e.g., burst dynamics, two-subpeak structure) were only observable in intracortical recordings and not surface EEG. This limits how well human EEG studies can capture the same level of detail as mouse intracortical recordings. Can they please address this and explain the usefulness of investigating the burst dynamics and two-peak structure?
6. The center frequency in Fmr1^{-/-} mice is ~3-4 Hz within V1. Can the authors please discuss how this can be considered alpha-like instead of theta?

Minor:

- is this particular alpha oscillation phenotype (frequency shift) also reported in other developmental disorders? Is it unique, or could it serve as a broader biomarker for neural circuit dysfunction?
- can the authors discuss in more detail how the particular alpha wave alterations might contribute to the sensory symptoms in FXS?

(Remarks on code availability)

Version 1:

Reviewer comments:

Reviewer #1

(Remarks to the Author)

This manuscript is a much-revised version of the previous submission. The authors removed some results (VEPs) and introduced new results on the effects produced by GABA_B agonist Arbaclofen. Although improved, I found it still quite dislocated, disorganized and I struggle to grasp a cohesive conclusion.

The authors find some modest effects in alpha-band oscillations in mice (albeit EEG effects are present only in V1 but not in S1). Based on the robust alpha phenotype in humans, they claim that alterations of alpha-oscillations represent a general phenotype of FRX.

The rest of the paper is still very confusing. I am not sure what the main goal of the paper is (finding a biomarker? Studying the mechanisms of alpha activity dysfunctions?).

Cell-type-specific manipulations: Genetic removal of frmp from glutamatergic neurons and glial cells of the forebrain results in effects on alpha activity. However, results from PV and SOM interneurons do not bear any causality with these

experiments on EMX-cre mice nor, in general, with FRX. Silencing PV cells might result in massive (non-specific?) downstream effects on pyramidal neurons. Also, silencing SOM cells with NpHR3 does not produce any significant results (Fig. 6q-t), although the authors claim a reduction of peak 1a and peak1b based on Fig. 6p (inset). In this context, the results of SOM activation with Chr2 are confusing. Overall, what do the authors conclude from these experiments on different cell types? How can they compare genetic removal of *frmp* from pyramidal neurons with broad pharmacogenetic silencing of PV cells, optogenetic silencing of SOM cells and optogenetic activation only of SOM cells? I am profoundly confused by the approach used by the authors.

Role of glial cells in LFP generation (EMX1 line): The conclusion that “loss of FMRP in excitatory neurons is sufficient for the alpha phenotype” underplays the established role of glia in shaping local field potentials. Acknowledging this limitation would strengthen interpretation.

Arbaclofen results: this effect might indeed be interesting, but I have a few perplexities. First, what is the rationale for using GABA_BR agonists in the context of fast inhibition provided by PV and SOM cells? To my knowledge, GABA_BRs are activated in pyramidal neurons mainly by neurogliaform interneurons. Importantly, these pharmacological results, as illustrated, are very difficult to understand. I struggle to judge whether there is any effect. Perhaps a better way to analyze/illustrate these results would be to plot alpha-oscillation effects for WT (saline + drug) and *Frmp*^{+/y} mice (saline + drug) with an overall ANOVA analysis to test for differences, based on treatment and/or genotype. Results illustrated in different graphs and figures are highly misleading (and not statistically sound).

Definition and use of “resting-state”: The manuscript refers to both a gray-screen condition and a screen-off condition as “resting-state.” Given that your own prior work (Hayden et al., 2021) demonstrated increased high-frequency power during gray-screen presentation, these conditions cannot be considered equivalent. The authors should clarify this terminology, as “resting-state” is a central concept in the manuscript.

The manuscript reports peak center frequencies with sub-Hz precision (e.g., 4.6 Hz), yet the underlying spectral resolution in the mouse datasets is 1 Hz at best. This suggests that the extra decimal precision is generated by the FOOOF/SpecParam model fits rather than directly measured from the PSD. While model fitting is appropriate for estimating peak positions, the manuscript should clarify that these values are interpolated estimates based on a fitted function, not actual measured frequencies. Explicitly stating the true frequency resolution (Δf) alongside the modeled estimates would prevent readers from overinterpreting the apparent precision.

In many occasions, results are moved from main figures to supplementary figures, although they are described as full independent datasets (see the analyses on microbursts, for example). This makes reading the paper quite a painful task. In my view, supplementary data should be non-essential extended information of data illustrated in the main figures (as is the case for the human data).

(Remarks on code availability)
none

Reviewer #2

(Remarks to the Author)

(Remarks on code availability)

I was able to inspect the MATLAB scripts provided for burst detection and 1/f (“aperiodic/periodic”) spectral decomposition in mouse EEG/LFP. The code is reasonably organized and references external helper functions (e.g., `improved_EEG_fit.m`, `improved_LFP_fit.m`, `st_fit_*`, `plot_spectrum_two_groups.m`, `fg_bootstrap_two_groups.m`) and third-party toolboxes (Chronux, EEGLAB’s `eegfilt`). However, reproducibility is only partial as currently written: several key dependencies, data files, and parameter conventions are insufficiently documented or hard-coded, which impedes exact replication by an independent lab.

On a more technical aspect:

- 1) Using `eegfilt` without explicit padding/filtfilt can introduce edge artifacts; I recommend zero-phase IIR/FIR with documented band edges and edge trimming.
- 2) The code reports Wilcoxon p-values and a Z-to-r effect size; however, multiple comparisons are not corrected across many tests (intercept, exponent, knee, peaks, etc.). Bootstrap CI arguments might be reversed depending on the helper’s expected order. These issues can alter significance calls and thus reproducibility.
- 3) Some analyses (e.g., bootstraps) can be made deterministic with `rng(seed)`. This is not set, which undermines computational reproducibility, as another researcher running the same script would not get the exact same numerical output.
- 4) Regarding effect size, A more standard/robust approach would be to calculate the matched-pairs rank biserial correlation coefficient or Cliff’s delta.

Reviewer #3

(Remarks to the Author)

We appreciate the authors' thorough and timely response to our reviewer queries. I am satisfied with the breadth of the responses and the modifications. I think this will be a fantastic contribution to the field.

(Remarks on code availability)

Reviewer #4

(Remarks to the Author)

The revised manuscript addresses my prior concerns comprehensively. The additional analyses/clarifications improve the paper's rigor and readability. I have no substantive remaining issues.

(Remarks on code availability)

Version 2:

Reviewer comments:

Reviewer #1

(Remarks to the Author)

The authors responded to my previous criticisms with an almost identical manuscript, and a rebuttal letter that expresses understandable frustration but provides quite unconvincing arguments. All the major issues that I raised remain, despite the authors claim factual errors from my side.

Overall, I am persuaded that this manuscript does not make the level of Nature Communications. I still think that the paper does not provide a clear conclusion nor a specific 'alpha' electrophysiological signature of FXS pathophysiology that spans species. In addition, I think that there are serious concerns about signal processing, thereby hindering the interpretation of the results.

Here's a tentative response to the authors' rebuttal.

Narrative and conclusions:

The authors claim that they find an alpha phenotype in humans and mice. However, the EEG/LFP is strongly affected in all frequency ranges. So, in my opinion, in both humans and mice there is not an alpha phenotype, but a general EEG/LFP phenotype. Just seeing widespread changes that include alpha frequencies is not generally sufficient to declare an alpha-specific phenotype. This suggests a broader electrophysiological pattern involving multiple frequency bands. This broader spectral change may reflect a state of cortical hyperexcitability or altered network dynamics rather than a purely alpha-specific phenotype. Therefore, it is more accurate to describe such findings as an altered spectral phenotype or a disrupted oscillatory profile rather than solely an alpha phenotype. The authors cite published literature justifying their alpha phenotype case. However, in these papers (e.g. Sohal et al., 2009; Cardin et al., 2009; Chen et al., 2017; Veit et al., 2017; Huang 2020) conclusions are drawn because specific manipulations induced effects on restricted frequency bands.

The authors then ask: Can the FXS "alpha" phenotype be reproduced by loss of FMRP restricted to cortical excitatory neurons and glia? Again, the evidence points towards a general effect, and not really an 'alpha' phenotype. A broad spectral effect resulting from FRMP deletion in pyramidal neurons and glial cells is unsurprising, as glutamatergic neurons and glial cells make the vast majority of the cell population constituting the cortical network from which they record.

They then look whether brute-force silencing of PV or SOM interneurons alters the LFP of the mouse V1. Silencing PV cells does indeed alter the LFP of V1. This observation is, however, orthogonal to the main topic of the paper. If the authors do not mean to provide a mechanism involving PV-cell role in FXS, the relationship between PV-cell activity and alpha oscillations is merely descriptive and correlative. But again, the main problem, in my opinion, is that silencing PV cells does not affect the b-peak of the alpha-frequency band of the LFP, but it affects many frequencies.

Regarding SOM cells, I am not convinced that there is a statistically significant effect following optogenetic silencing. In their response, the authors state: "Although the p-value for the by-animal comparison of peak 1a maximum power was 0.08 following multiple comparisons corrections (Fig. 6r), the genotype difference is still strongly apparent in the separation of the datasets and uncorrected p-value of 0.02." I find it confusing. Why did they use multiple comparisons to analyze Peak 1a power in two conditions? Why do they claim significant changes from uncorrected p-values? When do they use corrected-vs. uncorrected p-values?

Regardless of whether silencing SOM cells does indeed change the LFP power spectrum, and similarly to the case of PV cells, I really don't see how this experiment fits with a pathophysiological investigation of FXS.

Supplementary information, in the present form, is not extended data to results illustrated in the main figures. Instead, they are stand-alone findings that make the paper confusing. The authors' response is quite underwhelming, as this organization does very little to "connect[ing] worlds that are often not in sufficient dialogue with each other". I strongly encourage the authors to de-clutter the paper, focusing on a simple straightforward message.

Regarding the criticisms of signal processing and statistics:

Spectral resolution. In their response they state: “we used a different definition of frequency resolution based on the tapers (smoothing) used in the multitaper method, which sets the minimum resolvable distance between spectral peak”. Therefore, spectral resolution is determined by the taper bandwidth, and not by the actual (numerical) frequency resolution. The Fourier transform has bins spacing at $1/T = 0.2$ Hz. However, the taper window (TW) = 5 so the actual resolution (W) is 1 Hz. Hence, peak separation must follow W and not $1/T$. Whereas they can obtain a numerical sub-Hz resolution, this would be biologically misleading as sub-Hz values are extrapolated after multi-tapering. This makes sub-Hz precision claims biologically meaningless and thus decimals should not be compared. The separation in peaks 1a and 1b seems very aleatory (I don't see clear peaks in Fig. 3c, inset, black trace; Fig. 3j inset (both black and red trace); Fig. 4b, inset (both black and brown traces); Fig. 7f,p insets (both black and magenta traces); Fig. 8c, inset (both magenta and purple traces).

(Remarks on code availability)

Reviewer #2

(Remarks to the Author)

(Remarks on code availability)

Reviewer #3

(Remarks to the Author)

I appreciate the authors thoughtful responses. Though my initial queries were satisfactory, I was confused on this recent revision (though this issue may have existed in previous versions).

The methods clearly state that a 60 Hz notch filter was used throughout for murine preprocessing. However, across nearly every figure in the main manuscript and supplementary figures there prominent 60 Hz features. In addition, it appears as though statistics were performed showing significance bars under 60 Hz peaks.

I tried to come up with an alternative reason why this might be the case, but if the notch was performed properly it would significantly affect multiple analyses. I would appreciate if the authors could comment on this curiosity, as typically a line noise filter greatly attenuate these frequencies.

(Remarks on code availability)

Reviewer #4

(Remarks to the Author)

Overall, the authors have addressed Reviewer 1's comments appropriately. I have one additional remark regarding a specific point (see below).

Reviewer #1 comment:

Also, silencing SOM cells with NpHR3 does not produce any significant results (Fig. 6q-t), although the authors claim a reduction of peak 1a and peak1b based on Fig. 6p (inset). In this context, the results of SOM activation with ChR2 are confusing. Overall, what do the authors conclude from these experiments on different cell types?

Authors' response:

The reviewer is mistaken—the bootstrapped difference in periodic power was significantly reduced from 4.4-6.4 Hz, corresponding only to peak 1a, in NpHR3-expressing mice (see Fig. 6p and manuscript line 365). Although the p-value for the by-animal comparison of peak 1a maximum power was 0.08 following multiple comparisons corrections (Fig. 6r), the genotype difference is still strongly apparent in the separation of the datasets and uncorrected p-value of 0.02.

My comment:

The presentation of this result is somewhat confusing. In the main text, the authors claim a significant reduction in the periodic power of Pk1a, yet the figure reports a p-value of 0.08. The legend adds that the uncorrected p-value is 0.02. However, the figure citation appears incorrect: the text refers to panel 's' when it should cite panel 'r.'

(Remarks on code availability)

Reviewer #1 (Remarks to the Author):

This manuscript by Kornfeld-Sylla and colleagues reports neurophysiological results obtained from human FXS subjects and the Fmr1-/- mouse model. The authors compare EEG recordings in human subjects and mice, aiming to find common altered patterns to be used as biomarkers for the disease. The authors reason that this would allow for circuit dissection in the Fmr1-/- mouse model. The authors find some EEG and LFP alterations in mouse V1, and they correlate them with human EEG alterations. They go on and attempt to define the role of PV and SOM interneurons and excitatory pyramidal neurons in the emergence of LFP alterations in FXS mice. Finally, the authors analyze alterations in VEPs and visual plasticity in control and Fmr1-/- mice.

The topic is of high interest, and the manuscript contains high-quality data. However I feel that in the present form, the paper includes a series of orthogonal observations and falls short in providing a cohesive narrative and a clear solid message. The authors provide data from EEG from mice and humans, LFP analysis in 'resting state' and in response to visual stimulations in L4 of V1, analysis of visual evoked potentials and their plasticity only in mice. This is intermingled with an ambiguous and inconclusive comparison of different cell types that raises more questions than answers. In addition, even the individual sections of the paper often lack a clear conclusion and, when present, the interpretation requires some additional evidence to be fully convincing. The overall impression I had is that the end of the paper does not relate with the initial case that was built in the introduction.

We thank the reviewer for the positive assessment of the data quality and for the helpful comments. We understand that the narrative was difficult to follow and have worked to streamline the story and provide a cohesive thread between sections with clear conclusions for each section. We believe the manuscript is much stronger now, thanks to the feedback of this reviewer.

Some of our major changes to the structure of the manuscript include:

1. We have removed data related to visual stimuli and plasticity, to center the paper around measurements that can be compared between humans and mice. Thus, we just focus on resting-state LFP recordings in mice and tie this to EEG data in mice and humans.
2. We have clarified that the manipulations of PV+ and SOM+ cells are not meant to explain the emergence of LFP alterations in FXS mice, but rather to test the hypothesis that the activity of inhibitory interneurons is linked to alpha-like oscillations (in mice, in general, see text lines 319-66). Importantly, these results

validate the two subpeak structure of periodic Pk1 (this is a novel finding in our work) and set us up to show that pharmacologically boosting inhibition affects these alpha-like oscillations.

3. To this end, we have also added experiments using Arbaclofen, a GABA_B agonist, to address reviewer 4's interest in studying the effects of pharmacological treatment on the phenotypes (see text lines 368-434 and new figures 7 and 8 and supplementary fig. 11). We first examined the effect of Arbaclofen on WT mice, showing that it increases Pk1a maximum power and decreases Pk1a center frequency. In KO mice, the drug had the same general effect, although their alpha-like oscillations were less sensitive to Arbaclofen, while the sensitivity of their gamma oscillations to the drug was consistent between the two genotypes (see text lines 542-71). This reveals how measuring alpha oscillation disruptions refines our understanding of treatment response and relates to the initial goal of the paper to establish a translatable measure of pathophysiology and treatment response.

Major issues:

1. If the authors wish to provide an overarching comparison between human FXS subjects and the mouse model of this condition, they should compare most spontaneous and visually evoked alterations in both organisms, when this is feasible.

We thank the reviewer for this insight and agree wholeheartedly. The human and mouse datasets were not designed around each other but rather were collected separately and then brought together afterwards for the paper and analyzed uniformly. We see that the VEP data in mice are confusing because they are not coordinated with any human data, and thus we have removed this data to focus on the cross-species comparisons of spontaneous/resting-state data. We recognize that we do not have parallel human data for our Arbaclofen results but feel this is reasonable given the observed results for Pk1a center frequency in mice and the complexity of collecting such a dataset in humans.

2. The experiments on cell-type dissection are confusing. Silencing PV and SOM cells can lead to non-controllable network alterations that preclude the interpretation of the results. Moreover, the authors do not perform this analysis on wt and Fmr1-/- mice, so their conclusions are correlative at best. Importantly the silencing methods (pharmaco- vs. optogenetic silencing) present profound temporal controls, adding more confusing method-dependent alterations. I wonder whether one can compare pharmacogenetic

silencing of PV cells, optogenetic silencing of SOM cells and genetic removal of FMRP in glutamatergic neurons and glial cells.

The reviewer raises a valid concern about relating these interneuron manipulations in WT mice to our experiments in Fmr1-KO and EMX1-Cre Fmr1-KO mice. First, we have changed the way we present the data (see text lines 301-448). We now begin with the removal of FMRP in glutamatergic neurons and glial cells, to show that this is sufficient for the “alpha” phenotype, as it is for the “gamma” phenotype. We then replicate that the alpha-like oscillations are coupled to higher frequency (beta and gamma, 15-40 Hz) oscillations in WT mice and show that this coupling is impaired in KO and EMX1-KO mice. Since these higher frequency oscillations are known to be coordinated by PV+ and SOM+ interneurons, this suggests there might be a link between these interneurons and the alpha-like oscillations in all mice. And, if such a link does exist, then perhaps pharmacologically boosting inhibition might affect alpha-like oscillations in WT mice, and potentially improve FXS periodic Pk1 phenotypes, since inhibition is known to be deficient in FMR1 KO mice (and in mice where FMRP is only knocked out of excitatory neurons). Thus, we test if there is a link between these inhibitory interneurons and alpha-like oscillations in WT mice and then test the effect of boosting inhibition in both WT and KO mice with Arbaclofen. By setting the narrative up this way, we believe we are doing a better job of relating the interneuron manipulations to the alpha-like oscillations in general, rather than specifically comparing these manipulations to FXS. Any written inferences suggesting, for example, that the effect of a given interneuron manipulation looks similar to an FXS LFP phenotype, are correlative – not a direct comparison - and noted as such with clarifying language in the discussion (see text lines 531-40).

Regarding the methods of these manipulations - although such experiments might cause broad network alterations (which we acknowledge in the text lines 345-54), it is well established in the literature to use these methods to relate interneurons to different oscillations (see Cardin et al, 2009; Sohal et al, 2009; Chen et al, 2017; and Viet et al, 2017). Since we have clarified that our goal is to relate the interneurons to the oscillation, rather than to relate these various manipulations to the mechanistic basis of FXS phenotypes, we believe we are now well constrained within the precedents set by previous literature and can conclude that the alpha-like oscillations are differentially sensitive to manipulation of different classes of interneurons. Importantly, both the PV+ and SOM+ manipulations used viral instead of purely transgenic methods, limiting the manipulations to V1 and thus controlling for some non-specific cortical network effects (see methods lines 781-810).

Regarding the difference in pharMO- and opto-genetic approaches, we initially tried pharmacological inhibition of SOM+ interneurons but found this method was not successful, which is why we switched to an optogenetic approach. The timescales are of course different, but we have carefully selected time-series data where the effect of the manipulation will be maximized (i.e., during the laser on periods for the optogenetic manipulation, and 30 minutes after injection of CNO for the chemogenetic manipulation). We do not feel this temporal/methodological difference precludes demonstrating a relationship between the different classes of interneurons and periodic Pk1.

3. The authors should be cautious in comparing burst activity in humans and mice, as it seems that enhanced bursts occur in FXS patients only during tasks, whereas here the authors analyze bursts in mice only during resting states. Are these micro-bursts the same? The authors do not record burst activity during resting state in their human samples, raising a potential problem in comparing the two conditions in the two organisms.

In fact, we did study burst activity in our resting-state data from humans in our original submission and plotted the results in a supplemental figure, but we recognize that presenting the mouse and human data split up between main and supplemental figures made the data more challenging to locate and parse. Thus, so everything is together, we have moved the mouse burst analysis to the supplement as well and everything can now be seen together in Supplementary Fig. 7 (see also text lines 281-99).

We note did not observe a significant difference in burst dynamics between human TD and FXS subjects for our resting-state data due to large variability in the TD dataset. At this time, it is unclear whether this differing result from what was previously reported in the inter-stimulus period between tasks (Norris et al, 2022) is due to greater internal noise during rsEEG, differing analytical approaches and time-frames measured, the general engagement in periodic sensory stimuli vs. rest, or the task itself. It is intriguing that altered resting-state burst dynamics in Fmr1 KO mice were only detectable in layer 4 and not at the cortical surface. However, from a translational perspective these altered temporal dynamics are not as useful as the “alpha” phenotype, which we detected across species and both inside and outside mouse cortex, which is why we have combined all burst analysis into a supplementary figure.

4. The experiments of Fig. 8 are difficult to follow. Why are only KO mice shown in panels B and F? KO data is usually compared to their wt littermates, otherwise it becomes anecdotal. In panel C, why are VEPs lacking the P1 peak? What is the interpretation of the results illustrated in panel K? Overall, the SRP results are not even discussed. These results

seem preliminary, and they do not relate with the alterations of the alpha phenotype described in Figs. 1-5, which represents the only associated element with the human disease.

We have removed all mouse VEP data from this paper, including the SRP experiments.

Reviewer #2 (Remarks to the Author):

Reviewer #3 (Remarks to the Author):

This manuscript reports a translational electrophysiological biomarker linking alpha-frequency slowing in humans with homologous Pk1a deceleration in Fmr1 KO mice. The multimodal convergence—human EEG, mouse VEPs, and intracortical LFPs—is compelling and methodologically sophisticated. The study also attempts to assign mechanistic specificity to cortical excitatory neurons using Emx1-Cre conditional knockouts.

We thank the reviewer for the positive assessment of our methods and approach.

However, the central causal claim—that this biomarker is caused by loss of FMRP in excitatory neurons (as seen in the title)—goes beyond what the current data can justify. The Emx1-Cre model demonstrates sufficiency but not necessity, as no selective rescue experiment is performed. Developmental confounds and indirect network effects remain plausible explanations. Without demonstrating that restoring FMRP in excitatory neurons abolishes the biomarker, the causal inference is incomplete and should be reflected within the manuscript and title.

We thank the reviewer for this insight and have changed all language related to the Emx1-Cre results from describing causality to describing sufficiency. We have removed this clause from the title entirely, which was a bit too long anyway!

Behavioral state discrepancies (e.g., arousal, visual input) between humans and mice are

not explicitly controlled or covaried and are known to be strongly associated with alpha states. This should be more clearly documented and raised as limitation.

The reviewer raises an important point that while visual input was controlled for mouse LFP recordings and movement was measured with a piezo disk, revealing no significant difference in total movement between KO and WT groups (see Supplementary Fig. 7), this was not the case for human EEG recordings. We have now explicitly noted in the text that FXS children and adults move more and Fmr1 KO mice do not (see lines 287-89). As a point of consistency, for both mouse LFP and human EEG recordings, epochs containing large motion artifacts were excluded (while for mouse EEG recordings, epochs with no movement were selected through video-monitoring). For humans we recognize that there are differences between the experimental groups: for example, FXS subjects, especially the children, sometimes required different room lighting or visual input (i.e., a cartoon instead of the standard screen saver on the screen) to help them sit still. This has been more explicitly stated in the discussion (lines 511-514) and methods (lines 637-42 and 668-73) and raised as a limitation.

Cross-species translation is thoughtfully approached but weakened by anatomical and behavioral mismatch. Human alpha is recorded at the scalp without source localization, whereas mouse data come from layer 4 V1. Human EEG reflects a blend of signals from various cortical areas, not just V1 in contrast to the murine recordings. To a degree, the strong translational claim of functional homology between V1 and human occipital cortex is hinged on this argument. A source-reconstruction analysis (e.g., beamforming or CSD) is needed to confirm anatomical homology. At minimum, if source analysis is not performed than interpretation should be tempered on circuit level homology and speak more broadly to contributions from parietal, frontal, and non-visual regions.

We agree whole-heartedly with this comment and have performed source localization to show that 4-14 Hz absolute power (the range of periodic Pk1, i.e., alpha oscillations) is localized to occipital cortex in TD. We have included this analysis in Figure 1t. Despite the enlarged head size of FXS patients, which presents technical differences in interpreting source localization results, the power still localized to the back of the head, though slightly more superior and anterior. We have included this analysis in Supplementary Fig. 1l). See also text lines 166-172.

Statistical methodology is rigorous but under-documented in places. Multiple spectral comparisons per figure require formal correction (e.g., FDR). Even if bootstrapping is used, it is not typically a reason to avoid multiple comparison correction since that is determined

by the number of tests performed. Additional reporting on experimental blinding, randomization, and replication strategies—especially for opto- and chemogenetic conditions.

We thank the reviewer for the helpful feedback regarding statistics and have added the FDR multiple comparison correction for all Mann-Whitney tests from aperiodic and periodic spectra. We now report FDR-corrected p-values in the figures and uncorrected p-values in the figure legends. We have updated our statistical methods section to reflect our corrections of multiple comparisons (see text lines 990-92).

We have also added additional details to the Nature Reporting Summary about experimental blinding, randomization and replication strategies for experiments with drug treatments (including the new Arbaclofen experiments and the opto/chemogenetic experiments). Further details on chemogenetic methods can be found in Kaplan et al, 2016, as we reanalyzed data from this manuscript.

The biomarker's specificity is unproven, as similar alpha-frequency slowing has been documented in other neurodevelopmental and psychiatric disorders, including autism spectrum disorder (ASD), Down syndrome, and first-episode psychosis. This raises the possibility that alpha slowing is a non-specific marker of global neural dysfunction rather than a signature unique to Fragile X Syndrome. Without comparative data or test-retest reliability in FXS, its translational utility remains uncertain, claims of a biomarker should be tempered and discussed in the limitations.

We have removed the term “biomarker” when describing our results and have used “signature,” “phenotype,” or “translational measure of pathophysiology and treatment response” to offer a clearer, more limited description. By this definition, the measure could be shared by other disorders (see text lines 82-89), so long as it is not observed in the control/typically-developing groups, as it is not intended to be a diagnostic measure (since EEG is not needed to diagnose FXS). Instead, the measure offers a way to parse patients and monitor treatment responses across a variety of neuropsychiatric conditions.

Despite these critiques, your manuscript represents an ambitious and highly promising contribution. With careful revision addressing these points, your work can set a new standard for rigor in translational biomarker research. I look forward to seeing a revised version that realizes the full impact and potential of your findings.

We thank the reviewer for the positive assessment of the research and really appreciate the helpful feedback.

Reviewer #3 (Remarks on code availability):

Datasets are available on publication, human data is on request

Reviewer #4 (Remarks to the Author):

This study identifies an electrophysiological biomarker of Fragile X Syndrome (FXS)—a shift in alpha oscillations—that is present in both human patients and a mouse model (Fmr1-/- mice). Specifically, the researchers found that children and adults with FXS exhibit a consistent slowing of alpha rhythms in resting-state EEG, and this same pattern is mirrored by a corresponding shift in low-frequency oscillations in the visual cortex (V1) of Fmr1 knockout mice. The authors show that although both PV+ and SOM+ interneurons modulate different aspects of the affected oscillatory activity in mice (although not the frequency shift), it originates from a loss of the FMRP protein in cortical excitatory neurons, not inhibitory neurons. The study further links these oscillatory disruptions to altered sensory responses. Overall, this work provides a useful cross-species EEG analysis and some mechanistic insight into the pathophysiology of Fmr1-/- mice.

We thank the reviewer for the positive assessment of the cross-species approach and mechanistic studies.

Major comments:

1. The developmental trajectory of oscillatory changes differs between mice and humans – i.e., the alpha frequency decreases in adulthood in mice, whereas in humans it stagnates. This weakens the model's predictive power for developmental timing in translational research. Could the authors please address this? Also, could the authors please comment on the developmental trajectory of this alpha frequency phenotype in mice in their data, from juvenile to adulthood? Are the developmental differences only observed under luminescence conditions?

In both our human and mouse data, the developmental trajectory is cross-sectional and not longitudinal. To be able to confidently comment on the developmental trajectory in mice, as reasonably requested by this reviewer, we included in new Supplemental Fig. 6 a new cohort of mice which were recorded at both 4 and 11 weeks of age (i.e., a longitudinal study). This confirmed what was seen in the cross-sectional data, which was a difference in Pk1b power only in juvenile KO mice and a larger difference in Pk1a center frequency

between adult KO and WT mice than between juvenile mice. This validates cross-sectional approaches, generally.

This longitudinal data also confirms that both WT and KO mice *decrease* their peak 1a center frequency across development, with a steeper decrease in KO relative to WT (see text lines 249-60). As the reviewer noted, this differs from humans where periodic Pk1 center frequency *increases* across development, with stagnation in FXS. We have more clearly stated this difference in the discussion (line 483-86) and raised this as a limitation and important cross-species difference. The longitudinal data also reveals that Pk1b power increases across development in KO but not WT mice, confirming a juvenile-specific power difference, just like the Pk1 power difference is specific to human children. The developmental windows (i.e., greater power difference in juvenile, greater frequency shift in adults) are conserved across species but the reason for which these differences are reached is not conserved.

Finally, we do indeed only see the developmental difference in periodic power in luminance conditions, and we have clarified this in the text (lines 262-79 and 506-08) and Supplementary Fig. 5. As shown in Supplementary Fig. 5h, there is no Pk1b difference in the ‘monitor off’ condition in juvenile animals, but there is a Pk1b difference in the ‘gray screen’ condition in these same animals. This power difference emerges in the light because there is an inhibitory effect on Pk1b in juvenile KO (but not WT) mice (Supplementary Fig. 5j). There is not an inhibitory effect of light on Pk1b in adult KO mice, consistent with the developmental difference in luminance only (Supplementary Fig. 5k).

2. The alpha frequency changes in humans were consistent across different neocortical regions, whereas in the mouse model, they were observed in V1 but not S1. Do the authors have an explanation for this? How would this affect their interpretation as a translational biomarker? Unless the authors have a common function related to alpha frequency oscillations in V1, it will be difficult to reconcile these findings in a translational manner.

This is an excellent observation which relates to a comment from reviewer number 3. We have performed source localization analysis to show that the locus of 4-14 Hz absolute power (i.e., the range of periodic peak 1, or alpha oscillations) is occipital cortex (Fig. 1t). So, although the oscillations are strongly resonant in humans and measurable across cortex, they localize to human occipital cortex. Thus, the strongest prediction is for a conserved phenotype in occipital regions of the mouse, motivating our study of EEG there (see text lines 166-172). Given the human occipital source, we believe the presence of the phenotype in mouse V1 is sufficient for interpreting it as a translational measure of pathophysiology and treatment response. To emphasize that this is an occipital-centered

hypothesis, we have moved our mouse S1 EEG data to Supplementary Fig. 4. We note that the interquartile range of the S1 center frequency data is much broader, and this variability might contribute to why a phenotype is not observed here in mouse. We have added this observation to the text (lines 192-96). Perhaps with more mice, or more electrodes (i.e. more power), it could be observed in this other cortical region of KO mice, but it is most strongly conserved in V1.

Can the alpha phenotype be reversed or normalized by therapeutics? To be useful in clinical trials, the biomarker must respond to treatment. Does pharmacological treatment or re-expression of FMRP, or targeted V1 circuit manipulation, reverse the phenotype? And could this correct, for instance, the N1 component of the VEP? I think it would be important to demonstrate the link between alpha and VEP or another physiological readout to strengthen the relevance of alpha oscillations in V1 as a biomarker.

The reviewer makes a critical insight here: a biomarker must not only measure pathophysiology but also therapeutic target engagement. Thus, since we have explored the connection between the activity of inhibitory interneurons and the alpha-like oscillations in mice, we examined the effect of boosting inhibition with the GABA_B agonist Arbaclofen on the alpha-like oscillations in WT and Fmr1 KO mice (new figures 7 and 8, see text lines 368-434). Administering Arbaclofen in WT mice showed that both the power and frequency of Pk1a can be shifted by the drug (Figure 7). Intriguingly, alpha-like oscillations in KO mice were less sensitive to Arbaclofen and so responded to treatment similarly but more modestly (Figure 8). In contrast, the sensitivity of gamma oscillations to the drug was similar between the two genotypes, showing that the differential sensitivity between KO and WT mice was specific to the alpha-like oscillations and revealing how measuring alpha oscillation disruptions clarifies our understanding of treatment response. Please see text lines 542-71.

We also looked at the effect of a second class of therapeutic, an mGluR5 negative allosteric modulator called CTEP (Supplementary Fig. 12). The effect of Arbaclofen on Pk1a is specific to its mechanism of action (boosting inhibition), as administering 2 mg/kg of CTEP had no effect on Pk1a in either WT or KO mice (see text lines 436-48). We also note in the discussion that although Arbaclofen increased the genotype difference in Pk1a center frequency, a recent study found the PDE4D inhibitor BPN14770 had a therapeutic effect on the “alpha” phenotype in FXS human adults (see text lines 568-73).

Finally, to address the requests of reviewer one, we have removed all visually-evoked data from the paper and focused only on resting-state data where we can relate the data across

species. Thus, we no longer have VEP N1 data (although, we do find that Arbaclofen improves the N1 phenotype, not shown). We can, however, explore the link between the alpha oscillation disruptions and well-established “gamma” phenotype: high doses of both Arbaclofen and CTEP decrease periodic gamma power, but only Arbaclofen affects alpha-like oscillations. Moreover, only alpha-like oscillations have differential sensitivity between WT and KO genotypes to Arbaclofen, unlike gamma oscillations. This shows a distinction between the “alpha” and “gamma” phenotypes and demonstrates the relevance of using alpha oscillations as an additional measure of therapeutic response.

3. What is the exact mechanism driving the alpha frequency shift in FXS? While PV+ and SOM+ interneurons were shown to modulate different subcomponents of the alpha-like oscillations, neither accounted for the frequency shift – the core ‘alpha phenotype.’ This suggests the proposed mechanism is incomplete and may involve other mechanisms or circuits, such as the thalamocortical loop. In light of this, the interneuron modulation experiments seem insufficient to address the mechanism underlying the alpha frequency shift.

The reviewer makes an excellent point which relates to our interpretation of our results with Arbaclofen (lines 568-82). Arbaclofen decreased Pk1a center frequency in WT (as did optogenetically stimulating SOM+ interneurons, see Supplementary Fig. 10) and increased the genotype difference in Pk1a center frequency relative to saline-treated littermate WT mice when administered to KO mice. This suggests that the previously-reported therapeutic effect of the PDE4D inhibitor on the “alpha” phenotype, for example, is likely not solely through altering inhibition.

We are no longer attempting to explain the mechanism of the phenotype or proposing an “inhibition mechanism”. We have re-written our motivation of the inhibitory interneuron manipulations to test for a relationship between inhibition and alpha-like oscillations in general, rather than to specifically identify the mechanistic basis of the phenotype (see text lines 319-38). This set the stage for testing if boosting inhibition affects alpha-like oscillations in WT mice, and if so, in a therapeutic direction for *Fmr1* KO mice (see text lines 368-82). Overall, the aim of our paper was to establish a translatable low frequency oscillatory phenotype, which, thanks to the reviewers’ comments, we can now say measures both pathophysiology and target engagement. Our goal was thus not to solve the mechanism but to establish the phenotype, establish a link between inhibition and alpha-like oscillations, and then to use the link to motivate testing for pharmacological response in periodic Pk1 after administration of Arbaclofen. We look forward to future work which

will explicitly set out to solve the mechanism of the “alpha” phenotype and focus on the role of the thalamocortical loop in the phenotype!

4. Although alpha oscillations are proposed as a translational biomarker, spectral power and frequency measures are influenced by many biological and technical variables, such as brain region, recording setup, and luminance, amongst others. It may be difficult to generalize the biomarker for clinical use or across labs without standardization. Could the authors please clarify whether, and if so, how they standardized the recordings across labs? Do they have data from either lab to support the developmental trajectory of their alpha wave findings?

In our study, the two different human age groups were collected in different labs, as logistically such a developmental study is quite challenging within the same lab. The primary form of standardization was the uniform method used to analyze the data across the same electrodes; however, the recordings themselves were roughly standardized in terms of brain region and EEG net, with the same number of recording electrodes (128 sensors referenced to Cz) across roughly the same regions (or, as similar as possible with head size changing across development), see text lines 116-21 and methods. Luminance was not controlled for human data and we have now specifically raised this as a limitation in the discussion (lines 511-14) and clarified in the methods (lines 637-42 and 668-73). We noted that the child and adult datasets were collected at different sites (see text lines 116-19) and have now proposed a longitudinal study all at the same site as a future direction (see text lines 475-80).

5. The authors go into some detail to describe the neurobiological underpinnings of alpha frequency wave alterations in mice by focusing on layer 4. Some critical findings (e.g., burst dynamics, two-subpeak structure) were only observable in intracortical recordings and not surface EEG. This limits how well human EEG studies can capture the same level of detail as mouse intracortical recordings. Can they please address this and explain the usefulness of investigating the burst dynamics and two-peak structure?

Our logic for looking in layer 4 is that the alpha-like oscillations have been shown to be strongest in layer (L) 4 of WT mouse V1 (see Senzai et al, 2019), so we felt this type of measurement might be informative. Our finding of the two sub-peak structure in L4 was unexpected, as even in mouse literature this has never been shown before, but analyzing the structure was necessary to identify the intracortical “alpha” phenotype. A strength of our manuscript is that it ties together both EEG and intracortical recordings, so we understand the similarities (i.e, the core phenotype) and differences (i.e., the two sub-peak

structure) at these different levels. Studying the same phenotype at different levels offers different utilities: a clear translational utility for the EEG and more mechanistic insight in L4 LFP. For example, our hypothesis of inhibitory interneuron activity relating to alpha-like oscillations was built around the phase-amplitude coupling result, and that coupling is unique to L4 (see again Senzai et al, 2019). From a basic science perspective, the significance and usefulness of our observation of the two sub-peak structure was highlighted by the interneuron manipulations, as this showed each subpeak was related to a different class of interneuron, which clarified conflicting reports about which type of interneuron coordinated these oscillations (see text lines 521-27). From a translational perspective, the two subpeak structure gives insight into the mechanistic basis of alpha-like oscillations in general, which importantly becomes hypothesis-generating in terms of developing new therapeutic avenues (since we now show that these oscillations can be shifted by treatment). What is measured at the surface is a mix of signals (i.e., two subpeaks), which makes it harder to interrogate mechanistically what is going on, as noted by the reviewer; however, importantly, the core “alpha” phenotype and juvenile-specific power phenotype measured at the surface are still detectable through intracortical recordings, allowing for continuity as well as increased granularity in layer 4. Additionally, we note that human alpha oscillations themselves are also often analyzed in lower and upper halves (see text line 527-29), perhaps echoing the two-subpeak structure after all! Finally, in this updated version of the paper, the investigation of burst dynamics is less central and is no longer presented in main figures. We add a discussion in the text of the more limited translational utility of burst dynamics for resting-state data compared to the “alpha” phenotype, which could be measured inside and outside the cortex and across species and treatment conditions (see lines 296-99 and 351-54).

6. The center frequency in *Fmr1*-/- mice is ~3-4 Hz within V1. Can the authors please discuss how this can be considered alpha-like instead of theta?

We have added text in the discussion (lines 455-62) to address this key point. Calculating absolute power in traditional frequency bands can make it difficult to study oscillatory signals across different species, or even across different ages within the same species. For example, alpha oscillations for human children technically fall in the theta frequency range, but the field still considers them “alpha” oscillations (see text lines 127-29). The same logic has been broadly applied to mice. This is why we isolate periodic power in our manuscript, analyze everything as periodic Pk1 (which is the alpha band oscillation in human adults), and show the consistency in the disruption in this oscillation in FXS across species, which is of course technically in the theta frequency range in mice and human children. Because it is the same periodic Pk1 with the same disruptions in *Fmr1* KO mice,

we interpret it as related to the human adult alpha oscillation (so, alpha-like). We believe this adds credence to the already-existing precedent in the murine literature to refer to these theta oscillations in mouse V1 as alpha-like (see Nestvogel et al, 2021 and Senzai et al, 2019), and we cite these papers multiple times in the text (see lines 203-05 in addition to lines 455-62 mentioned above).

Minor:

- is this particular alpha oscillation phenotype (frequency shift) also reported in other developmental disorders? Is it unique, or could it serve as a broader biomarker for neural circuit dysfunction?

The shift in peak alpha frequency is also observed in other developmental disorders such as intellectual disability and idiopathic ASD. Thus, it likely can serve as a broader measure of neural circuit dysfunction (i.e., the pathophysiology of other neuropsychiatric and neurodevelopmental disorders) and we have included this insight in the text (lines 86-89).

- can the authors discuss in more detail how the particular alpha wave alterations might contribute to the sensory symptoms in FXS?

Pedapati et al, 2022 found that reduced relative alpha power in FXS correlated with poor performance on the Woodcock-Johnson III auditory attention task (which included auditory discrimination tasks), abnormal speech, hyperactivity, and impaired social-communication functioning. Regarding other clinical correlations, they found that higher anxiety and obsessive behaviors measured with the Anxiety, Depression, and Mood Scale (ADAMS) correlated with both reduced relative alpha power and reduced peak alpha frequency. We have added a specific mention of these clinical correlations in the text (line 74).

Reviewers' comments:

As reviewers 3 and 4 were satisfied with our revisions, we begin by acknowledging their feedback.

Reviewer #3 (Remarks to the Author):

We appreciate the authors' thorough and timely response to our reviewer queries. I am satisfied with the breadth of the responses and the modifications. I think this will be a fantastic contribution to the field.

We are grateful for the reviewers' helpful feedback, positive assessment of our work, and the recognition that this study identifies an electrophysiological signature of FXS pathophysiology that shows great promise for bridging the divide between mismatching preclinical and clinical measures of treatment response.

Reviewer #4 (Remarks to the Author):

The revised manuscript addresses my prior concerns comprehensively. The additional analyses/clarifications improve the paper's rigor and readability. I have no substantive remaining issues.

We thank the reviewer for acknowledging the comprehensiveness, rigor, and readability of our revised manuscript. Addressing the reviewer's thoughtful comments greatly strengthened the paper.

Reviewer #1 (Remarks to the Author):

This manuscript is a much-revised version of the previous submission. The authors removed some results (VEPs) and introduced new results on the effects produced by GABA_BR agonist Arbaclofen. Although improved, I found it still quite dislocated, disorganized and I struggle to grasp a cohesive conclusion.

In light of the other reviewers' comments, we are surprised to learn that this reviewer found it difficult to follow the trajectory and conclusions of the revised manuscript. Our overarching goal was to establish that disrupted resting-state alpha oscillations in FXS are an electrophysiological signature of pathophysiology that translate across species and can be used as a quantitative measure of treatment response.

Our findings, organized in four main sections, address the following questions:

- 1. Is there a difference in how alpha oscillations are disrupted in FXS children and adults, and is there a parallel "alpha" phenotype in *Fmr1* KO mice?** In the first part of the manuscript, we characterize resting-state alpha oscillation disruptions in children and adults with FXS (Fig. 1, lines 115-65), showing that the slowing of alpha oscillations ("alpha" phenotype) is more pronounced in FXS adults than children, while in FXS children these oscillations are also reduced in power – a phenotype not present in adults. We next localize the source of the oscillations to human occipital cortex (Fig. 1, lines 167-73) and identify a parallel "alpha" phenotype in adult *Fmr1* KO mice using surface recordings over V1 (Fig. 2, lines 175-200). Finally, using LFP recordings in L4 of V1, we show parallel developmental

changes to the disruptions in alpha-like oscillations in *Fmr1* KO mice as seen in humans with FXS (Fig. 3, lines 202-61). More specifically, we show for the first time that murine alpha-like oscillations recorded in L4 contain two subpeaks (a and b), with the “alpha” phenotype contained in the first subpeak and a juvenile-specific power phenotype contained in the second subpeak. *We conclude that alpha oscillation disruptions are a translatable measure of FXS pathophysiology with consistent developmental changes across species (lines 453-61 and 477-506).*

- 2. Can the FXS “alpha” phenotype be reproduced by loss of FMRP restricted to cortical excitatory neurons and glia?** After establishing the “alpha” phenotype in the *Fmr1* KO mouse model, in the next part of the manuscript we took advantage of the genetic tools available in mice to interrogate the cellular basis for the phenotype (Fig. 4, lines 302-319). *We conclude that loss of FMRP in the excitatory neurons and glia of the cortex is sufficient for the disruptions in alpha-like oscillations, as it is for the “gamma” phenotype (lines 318-19 and 461-66).*
- 3. In WT mice, how are “alpha” oscillations influenced by the activity of cortical GABAergic inhibitory interneurons?** The novel identification of the two-subpeak structure of alpha-like oscillations in mice highlights how little is understood about this rhythmic activity. Therefore, in the next part of the manuscript, we investigated in WT mice the consequences of manipulating GABAergic inhibition on alpha-like oscillations in V1. *We conclude (1) that the ‘a’ and ‘b’ subpeaks of periodic peak 1 (alpha-like oscillations) reflect differential activity of two classes of GABAergic interneurons (SOM+ and PV+; Fig. 6, lines 342-68) and (2) that the power and center frequency of alpha-like oscillations are sensitive to systemic modulation of GABAergic inhibition by the GABA_B receptor agonist Arbaclofen (Fig. 7, lines 386-405 and 544-54).*
- 4. Can “alpha” oscillations be used to assess CNS target engagement and therapeutic responses to systemic drug treatments under investigation for FXS?** Based on the sensitivity of alpha-like oscillations to manipulations of GABAergic signaling, in the final part of the manuscript we investigate the effect of Arbaclofen in *Fmr1* KO mice. We show that treatment improves periodic peak 1a (alpha-like) and high-frequency (gamma) power in KO mice but exacerbates the genotype difference in peak 1a center frequency (Fig. 8, lines 407-425). In addition, we find that a higher dose of Arbaclofen was required to modulate alpha-like oscillations in KO than WT mice, whereas gamma oscillations were equally sensitive across the genotypes (please see lines 466-75 and 556-68). In contrast, another putative therapeutic for FXS, the mGluR5 NAM CTEP, had no effect on alpha-like oscillations, Supp. Fig. 12, lines 438-50). *We conclude that measuring changes in alpha oscillations holds great promise for objective cross-species quantification of treatment responses (lines 466-75 and 586-96).*

Overall, we feel that we achieved our main goal of establishing the “alpha” phenotype as a measure of FXS pathophysiology that spans species and can be used to assess treatment response. Thanks to the guidance from our reviewers, we were able to weave together the applied and basic science experiments into an organized and cohesive storyline, using results in applied experiments to motivate basic science experiments and using basic science results to inform therapeutic approaches. In this newest revision, we have made a few additional minor changes to strengthen the logical link between the basic science experiments targeting GABAergic inhibition (lines 49, 51, 104, 107, 330, 371-81, 450, 524, 533, 545) and to further clarify our Arbaclofen results in WT and KO mice (lines 394-96, 415-16, 422, and 559). Our manuscript spans basic science to clinical research and has an abundance of data from multiple sources, organized to address distinct questions. Communicating these results coherently and thoughtfully is important: since disruptions in alpha-like oscillations are characteristic of many neuropsychiatric conditions, we hope our framework for studying alpha disruptions preclinically can be applied to broadly advance translational research.

The authors find some modest effects in alpha-band oscillations in mice (albeit EEG effects are present only in V1 but not in S1). Based on the robust alpha phenotype in humans, they claim that alterations of alpha-oscillations represent a general phenotype of FRX.

We feel that this summary does not do justice to what is accomplished in the first part of the paper. We show parallel differences in the “alpha” phenotype at different developmental stages in both species, with effects in mice being more than just “modest” (see Fig. 3 for highly significant FDR-corrected p-values). We also discovered a two-subpeak structure to the alpha-like oscillations in mice when recording in L4, with differential phenotypes in the subpeaks in *Fmr1* KO mice based on age (Fig. 3). We do not feel that the “alpha” phenotype being only in V1 in mice is a weakness as the reviewer implies, given that—thanks to the previous feedback from other reviewers—we have now shown a link between occipital cortex and the source of alpha oscillations in human subjects (Fig. 1t).

The rest of the paper is still very confusing. I am not sure what the main goal of the paper is (finding a biomarker? Studying the mechanisms of alpha activity dysfunctions?).

As stated, the main goal of the paper is to establish an electrophysiological signature of FXS pathophysiology that spans species and can report treatment response. In the first part of the paper, we established the signature as a trans-species measurement of FXS pathophysiology. In the second part of the paper, we use the “alpha” phenotype to assess treatment response to administration of Arbaclofen as it would be used clinically (i.e., systemically). These two parts were bridged by basic science experiments in WT mice that substantiated results from the first part and informed our therapeutic approach in the second part of the manuscript.

To clarify and reiterate, we are not studying mechanisms of alpha activity dysfunctions. Indeed, we stated the following in our previous rebuttal: *“We are no longer attempting to explain the mechanism of the phenotype or proposing an “inhibition mechanism”. We have re-written our motivation of the inhibitory interneuron manipulations to test for a relationship between inhibition*

and alpha-like oscillations in general, rather than to specifically identify the mechanistic basis of the phenotype (see text lines 319-38). This set the stage for testing if boosting inhibition affects alpha-like oscillations in WT mice, and if so, in a therapeutic direction for Fmr1 KO mice (see text lines 368-82). Overall, the aim of our paper was to establish a translatable low frequency oscillatory phenotype, which, thanks to the reviewers' comments, we can now say measures both pathophysiology and target engagement. Our goal was thus not to solve the mechanism but to establish the phenotype, establish a link between inhibition and alpha-like oscillations, and then to use the link to motivate testing for pharmacological response in periodic Pk1 after administration of Arbaclofen. We look forward to future work which will explicitly set out to solve the mechanism of the "alpha" phenotype and focus on the role of the thalamocortical loop in the phenotype!"

Cell-type-specific manipulations: Genetic removal of frmp from glutamatergic neurons and glial cells of the forebrain results in effects on alpha activity. However, results from PV and SOM interneurons do not bear any causality with these experiments on EMX-cre mice nor, in general, with FRX.

The reviewer's suggestion that we infer a causal relationship between the data in Fmr1-EMX-Cre and the cell-specific manipulation data is factually incorrect. We are not equating either of these manipulations to *Fmr1* KO or EMX1-Fmr1-KO mice. In our original point-by-point reviewer response and previously updated manuscript, we made it very clear that this data was not linked to FXS, stating that we were *"relating the interneuron manipulations to the alpha-like oscillations in general, rather than specifically comparing these manipulations to FXS"* and that *"any written inferences suggesting, for example, that the effect of a given interneuron manipulation looks similar to an FXS LFP phenotype, are correlative – not a direct comparison - and noted as such with clarifying language in the discussion (see text lines 531-40)."*

Silencing PV cells might result in massive (non-specific?) downstream effects on pyramidal neurons.

After the reviewer made this same statement in the first review, we wrote this in our previous reviewer response:

"Regarding the methods of these manipulations - although such experiments might cause broad network alterations (which we acknowledge in the text lines 345-54), it is well established in the literature to use these methods to relate interneurons to different oscillations (see Cardin et al, 2009; Sohal et al, 2009; Chen et al, 2017; and Viet et al, 2017). Since we have clarified that our goal is to relate the interneurons to the oscillation, rather than to relate these various manipulations to the mechanistic basis of FXS phenotypes, we believe we are now well constrained within the precedents set by previous literature and can conclude that the alpha-like oscillations are differentially sensitive to manipulation of different classes of interneurons. Importantly, both the PV+ and SOM+ manipulations used viral instead of purely transgenic methods, limiting the manipulations to V1 and thus controlling for some non-specific cortical network effects (see methods lines 781-810)."

Indeed, the list of studies that have employed opto- or chemo-genetic silencing of PV+ cells and studied resulting neural spiking or oscillatory activity in V1 or other cortical areas is very substantial (here are a few more citations: Stark et al, 2013; Struber et al, 2022; and Huang et al, 2024). Since the reviewer restated the concern without referring to our response, we see no reason why our use of this method is unacceptable given the vast precedent in literature and our acknowledgement of the broad network effects in the text.

Also, silencing SOM cells with NpHR3 does not produce any significant results (Fig. 6q-t), although the authors claim a reduction of peak 1a and peak1b based on Fig. 6p (inset). In this context, the results of SOM activation with ChR2 are confusing. Overall, what do the authors conclude from these experiments on different cell types?

The reviewer is mistaken—the bootstrapped difference in periodic power was significantly reduced from 4.4-6.4 Hz, corresponding only to peak 1a, in NpHR3-expressing mice (see Fig. 6p and manuscript line 365). Although the p-value for the by-animal comparison of peak 1a maximum power was 0.08 following multiple comparisons corrections (Fig. 6r), the genotype difference is still strongly apparent in the separation of the datasets and uncorrected p-value of 0.02. The factual error here reveals why the reviewer did not understand the conclusion from these experiments. The results of our interneuron manipulations revealed a separable mechanistic basis for subpeaks 1a and 1b of peak 1 (the alpha-like oscillation). We are the first to report these sub-peaks, as theta oscillations in mice have always been uniformly analyzed over the 3-8 Hz frequency range. Since these subpeaks reveal different FXS phenotypes across development, separably mapping subpeak 1a and 1b onto SOM and PV+ interneurons, respectively, critically substantiated our observation of the two subpeak structure in the FXS data. We dedicate an entire paragraph in the conclusion of the manuscript to the significance of this peak 1a and 1b distinction (lines 518-31). We delineated and reiterated many times in the introduction, results, conclusions, and reviewer responses that the most significant conclusion from the interneuron manipulations is the distinction between peak 1a (SOM+) and 1b (PV+), so we feel that the conclusion is sufficiently clear.

How can they compare genetic removal of *fmrp* from pyramidal neurons with broad pharmacogenetic silencing of PV cells, optogenetic silencing of SOM cells and optogenetic activation only of SOM cells? I am profoundly confused by the approach used by the authors.

This sentence is repeated from the first review, but as stated we did not have a link in our revised manuscript between the genetic removal of FMRP and the inhibitory interneuron experiments. As clearly outlined in the manuscript and the previous rebuttal, we show a coupling between the phase of alpha-like oscillations and the amplitude of higher frequency (beta/gamma band, 15-40 Hz) oscillations coordinated by these classes of inhibitory interneurons (and this phase-amplitude coupling is weaker in *Fmr1* KO and EMX1-*Fmr1*-KO mice), which motivated our manipulations (Figure 5, lines 321-30) – but made clear that these interneuron experiments do not inform the mechanisms for alpha disruptions in FXS mice models. Instead, the experiments allowed us to study the mechanisms of resting-state alpha-like oscillations in WT mice, which to date have not been well-studied, meaning we were addressing a basic science question motivated by the FXS

results. In addition to revealing the separable mechanistic basis for subpeaks 1a and 1b, the results of our interneuron manipulations suggested a role for the GABAergic system in these oscillations, motivating our Arbaclofen experiments in FXS mice.

Indeed, we stated this in our previous rebuttal to reviewer 1 : *“We have clarified that the manipulations of PV+ and SOM+ cells are not meant to explain the emergence of LFP alterations in FXS mice, but rather to test the hypothesis that the activity of inhibitory interneurons is linked to alpha-like oscillations (in mice, in general, see text lines 319-66). Importantly, these results validate the two subpeak structure of periodic Pk1 (this is a novel finding in our work) and set us up to show that pharmacologically boosting inhibition affects these alpha-like oscillations.”*

Role of glial cells in LFP generation (EMX1 line): The conclusion that “loss of FMRP in excitatory neurons is sufficient for the alpha phenotype” underplays the established role of glia in shaping local field potentials. Acknowledging this limitation would strengthen interpretation.

We agree with the reviewer that it is important to acknowledge the possible contribution of glia to the reported results in EMX1-Fmr1-KO mice, but they are factually incorrect that we failed to do that. We mentioned the contribution of glia in the summary of results in the intro (lines 100-103), in the section heading in the results (lines 302-303), in the conclusions (lines 464-66), and throughout the figure legend for Fig. 4. For utmost thoroughness, we have now made sure every sentence related to this result mentions glia (see lines 46 and 318).

Arbaclofen results: this effect might indeed be interesting, but I have a few perplexities. First, what is the rationale for using GABA_BR agonists in the context of fast inhibition provided by PV and SOM cells? To my knowledge, GABA_BRs are activated in pyramidal neurons mainly by neurogliaform interneurons.

To clarify, the central objective of these experiments was to determine if the “alpha” phenotype shared by mice and humans could reveal therapeutic target engagement and treatment response. Although a large and growing number of treatments have been proposed for FXS, we chose Arbaclofen as a test case because (1) it has shown promise in multiple mouse and human studies, and (2) it is a GABA_B receptor agonist that, based our studies of GABAergic SOM+ and PV+ interneuron involvement in alpha-like oscillations in WT mice, we expected to influence alpha oscillations (see lines 371-82). Additionally, because Arbaclofen and racemic baclofen were previously shown to affect the gamma phenotype in humans and animal models, we knew we would have a control within our electrophysiological measures to confirm target engagement (lines 416-18). We do not claim an exact mechanistic connection between Arbaclofen and the inhibitory interneurons studied in Fig. 6, nor do we believe that one is necessary to tie together the findings in the manuscript. Ultimately, our analytical method revealed multiple metrics of target engagement, including in the alpha-like oscillation parameters that were unique to the mechanism of action of Arbaclofen on the GABAergic system (as the mGluR5 NAM CTEP did not affect alpha-like oscillations, see Supp. Fig. 12 and text lines 438-50).

Importantly, these pharmacological results, as illustrated, are very difficult to understand. I struggle to judge whether there is any effect. Perhaps a better way to analyze/illustrate these results would be to plot alpha-oscillation effects for WT (saline + drug) and Frmp+/y mice (saline + drug) with an overall ANOVA analysis to test for differences, based on treatment and/or genotype. Results illustrated in different graphs and figures are highly misleading (and not statistically sound).

The reviewer asserts that we do not statistically test the effects of Arbaclofen across genotype, but this is factually incorrect because we do compare alpha-like oscillation properties across genotype and treatment (WT+saline, FX+saline, and FX+drug; Fig. 8i-j and text lines 419-25) with appropriate nonparametric statistical tests and FDR multiple comparisons corrections (the ANOVA the reviewer suggests assumes normality and so would not be statistically sound). The effect of boosting GABAergic inhibition using Arbaclofen in WT mice is a separate basic science question we address in Fig. 7, building on our basic science experiments discussed in Fig. 6. As summarized in lines 544-68 of the discussion, we found that boosting GABAergic signaling increased peak 1a power in WT mice at both doses but only had this effect in FXS mice at the higher dose (in contrast, the drug's effect on gamma periodic power was consistent across genotypes). That the differential sensitivity of KO and WT mice to Arbaclofen was specific to the alpha-like oscillations reveals how measuring this electrophysiological signature clarifies our understanding of treatment response.

At the higher dose, Arbaclofen also reduced peak 1a center frequency in WT mice and increased the genotype difference in KO mice, revealing treatment response but not a normalizing effect on the "alpha" phenotype (in contrast to the PDE4D inhibitor BPN14770, which was found in a recent study to have a therapeutic effect on the "alpha" phenotype in FXS human adults - see text lines 570-75). Moreover, the effect of Arbaclofen on Pk1a is specific to its mechanism of action (boosting GABAergic inhibition), as administering 2 mg/kg of CTEP had no effect on Pk1a in either WT or KO mice (see text lines 438-50). We were quite clear about these drug effects in the results, conclusions, and reviewer responses (to both reviewers 1 and 4), and other reviewers seemed to understand them without issue (including reviewer 4, who requested the experiments in the first place). However, in our current manuscript revision, inspired by feedback from reviewer 2 to further clarify these results, we re-ran the Arbaclofen data using a matched-pairs statistical test design for within-genotype comparisons and updated the p-values accordingly (Fig. 7-8), which strengthened some of our reported effects.

We also clearly summarized these drug results previously in our rebuttal to reviewer 1: *"We first examined the effect of Arbaclofen on WT mice, showing that it increases Pk1a maximum power and decreases Pk1a center frequency. In KO mice, the drug had the same general effect, although their alpha-like oscillations were less sensitive to Arbaclofen, while the sensitivity of their gamma oscillations to the drug was consistent between the two genotypes (see text lines 542-71). This reveals how measuring alpha oscillation disruptions refines our understanding of treatment response and relates to the initial goal of the paper to establish a translatable measure of pathophysiology and treatment response."*

Definition and use of “resting-state”: The manuscript refers to both a gray-screen condition and a screen-off condition as “resting-state.” Given that your own prior work (Hayden et al., 2021) demonstrated increased high-frequency power during gray-screen presentation, these conditions cannot be considered equivalent. The authors should clarify this terminology, as “resting-state” is a central concept in the manuscript.

In the vast literature of EEG studies with human subjects, resting-state measurements are often collected under both “lights on” and “lights off” conditions, as the varying luminance affects oscillatory power. Therefore, the reviewer is factually incorrect to assert that “resting-state” terminology must change under conditions of varying luminance. Since the varying luminance does not affect the designation as resting-state in humans, why should it for mice? Treating mouse research differently goes against the whole premise of the paper and is unfounded.

The referee cites previous work from the Bear Lab to show differences between black and gray screen, a finding which we already addressed in the manuscript (lines 215-17) as being a result of luminance processing. Of course there is a difference in periodic and aperiodic electrical activity in primary visual cortex when there is luminance, and we show these differences in Supplementary Fig 5, but as stated, that has no bearing on the concept of “resting-state.”

The manuscript reports peak center frequencies with sub-Hz precision (e.g., 4.6 Hz), yet the underlying spectral resolution in the mouse datasets is 1 Hz at best. This suggests that the extra decimal precision is generated by the FOOOF/SpecParam model fits rather than directly measured from the PSD. While model fitting is appropriate for estimating peak positions, the manuscript should clarify that these values are interpolated estimates based on a fitted function, not actual measured frequencies. Explicitly stating the true frequency resolution (Δf) alongside the modeled estimates would prevent readers from overinterpreting the apparent precision.

The reviewer confuses the standard signal processing definition of frequency resolution with the definition we presented in our methods section (lines 858-60) for our multi-taper spectral analysis. By convention, frequency bin size (resolution) through the Fourier Transform is calculated based on the sampling frequency and the sample length. By this definition, we have 0.2 Hz frequency bins (resolution) for mice and thus are not making up or artificially creating any decimal values through our analysis, as the referee falsely asserts. As we clearly described in our Methods section, we used a different definition of frequency resolution based on the tapers (smoothing) used in the multi-taper method, which sets the minimum resolvable distance between spectral peaks. Importantly, this *does not* change the underlying 0.2 Hz frequency bins in our power spectra, which is why we have sub-Hz data values. However, for utmost clarity, we have added sentences to our methods explicitly stating the PSD frequency bin sizes for each group/species (mice, pediatric human subjects, and adult human subjects – see methods lines 838-39 and 857-59) and changed our terminology from “frequency resolution” to “spectral resolution” (see methods lines 859 and 869).

Moreover, we do not use the FOOOF/SpecParam model as the reviewer erroneously assumes. We explained this very clearly in our “Modified one over f fitting analysis” section in the methods (lines 872-944). We do not fit gaussians to our data to model the periodic signals, as is done with FOOOF. We wrote novel code to fit Lorentzian functions to our data to model the aperiodic signal, then subtracted that out from original PSD signal and found the maximum value(s) in range of the periodic peaks from the resulting spectrum. In other words, the peak frequency values are measured directly from the PSD.

In many occasions, results are moved from main figures to supplementary figures, although they are described as full independent datasets (see the analyses on microbursts, for example). This makes reading the paper quite a painful task. In my view, supplementary data should be non-essential extended information of data illustrated in the main figures (as is the case for the human data).

This manuscript is the union of clinical research and basic science, attempting to connect worlds that are often not in sufficient dialogue with each other, thereby stunting scientific and clinical progress. We bridge this divide with uniform analytical methods. Of course, because there is so much data from so many different groups, we naturally need supplementary material. As we clearly explained in the reviewer responses, we prioritized the analyses that yielded translational phenotypes in the main figures, so we do not feel that it is unreasonable (or painful for the reader) to place the burst analysis in a supplement.

Reviewer #1 (Remarks on code availability):

none

Reviewer #2 (Remarks to the Author):

Reviewer #2 (Remarks on code availability):

I was able to inspect the MATLAB scripts provided for burst detection and $1/f$ (“aperiodic/periodic”) spectral decomposition in mouse EEG/LFP. The code is reasonably organized and references external helper functions (e.g., `improved_EEG_fit.m`, `improved_LFP_fit.m`, `st_fit_*`, `plot_spectrum_two_groups.m`, `fg_bootstrap_two_groups.m`) and third-party toolboxes (Chronux, EEGLAB’s `eegfilt`). However, reproducibility is only partial as currently written: several key dependencies, data files, and parameter conventions are insufficiently documented or hard-coded, which impedes exact replication by an independent lab.

We are very grateful for this new feedback from reviewer 2, as we are new to the open code submission process and benefited greatly from the review. We are pleased that there were no technical issues raised with the novel aperiodic and periodic analysis portion of our code. We

apologize for any difficulty with replication due to our statistical analysis code or issues with our data files. We have updated the GitHub in the following ways to address the concerns:

- 1) We have updated our burst analysis code to be able to handle data cases with edge effects, which we fortunately do not see in our data.
- 2) We have included multiple comparisons corrections for non-parametric tests, differentiated between matched-pairs and independent samples for certain tests in the manuscript (text lines 988-92 and 997-1001) and code, and updated our method of calculating effect size (see methods section in text, lines 1001-05).
- 3) We have fixed the dependencies and parameter conventions related to bootstrapping outlined in the technical points below, including generating deterministic results.
- 4) We have included on the Github two complete sample data files for each type of code (one for EEG, one for LFP – in epochs for 1/f analysis and continuous data for burst analysis). We previously had multiple, partial sample data files uploaded due to file size limitations on GitHub, but we can see that this could be confusing to the user. All complete data files, including the large (n = 67 animal .mat files) are on Figshare, and the filenames in the code and data files uploaded in Github now match the filenames in Figshare.
- 5) We have more thoroughly commented our code to improve readability and reproducibility.

On a more technical aspect:

1) Using `eegfilt` without explicit padding/`filtfilt` can introduce edge artifacts; I recommend zero-phase IIR/FIR with documented band edges and edge trimming.

We appreciate the reviewer's concern about edge artifacts. We do not use `eegfilt.m` under causal mode, which means that the function uses a zero-phase least squares FIR filter implemented through "`filtfilt`". Due to this, the long continuous data segments we use (100 sec), and the lack of artifacts within our detrended signals, our data from mice fortunately do not have any sizeable edge effects. We have added in an option into the code for the user to see a time-frequency analysis to better visualize that there are no edge effects, along with a supporting MATLAB function "`BURST_ANALYSIS_W_SPECGM.m`." However, we understand that future users may want to analyze their own more noisy data that could include edge effects, so we have also added in options for the user to implement zero-padding (before filtering) or edge trimming (removing a "buffer zone" of 2 sec on either end after filtering) to the `run_burst_analysis.m` code. Since edge effects were minimal for our data, using zero-padding or edge-trimming did not significantly change the results with the data from the manuscript. As a result, we did not change our supplementary figures, but just added the option to use zero padding or edge trimming to our code for the benefit of future users with other datasets. We have also more clearly documented our band edges in the comments in the code and in the methods section of the manuscript (lines 956-60).

2) The code reports Wilcoxon p-values and a Z-to-r effect size; however, multiple comparisons are not corrected across many tests (intercept, exponent, knee, peaks, etc.).

Although we did use an FDR correction for multiple comparisons, as requested by reviewer 3, discussed in our methods (lines 999-1000), and demonstrated in the updated p-values in our figures in our revised manuscript relative to our initial submission, we made these computations by hand as they were relatively straightforward (simple rank ordering and multiplication). We sincerely apologize that we did not include the calculations for these corrections in our code, and they are now included, calling on a new helper function called “FDR_correct.m.” Consistent with the corrections made by hand and described in our methods previously, we corrected for each comparison made per plot (i.e., separate corrections for tests from the aperiodic plot and tests from the periodic plot). Performing these calculations in MATLAB produced the same results as before but allowed for a bit more precision with the p-values. Overall, we also generally switched to the convention of reporting p-values out to two decimal places in figures to improve readability.

Bootstrap CI arguments might be reversed depending on the helper’s expected order. These issues can alter significance calls and thus reproducibility.

Since we are always plotting the points of significant difference between groups (i.e. non-zero points on the difference plot, marked with the red scatter plot points, not the specific difference plot itself), the sign of the difference plot values (and therefore the order of input arguments, i.e., x-y vs y-x) shouldn’t matter. However, to facilitate exact replication, we have now made it clear in the beginning of the run_improved_EEG_fit.m or run_improved_LFP_fit.m how to call the different datasheets to precisely follow our approach. We have also made it clear exactly which files in FigShare correspond to each figure, and these filenames match the datasheets listed in MATLAB.

3)Some analyses (e.g., bootstraps) can be made deterministic with rng(seed). This is not set, which undermines computational reproducibility, as another researcher running the same script would not get the exact same numerical output.

We have updated the bootstrapping code so it can run either in either a deterministic or random mode, so the user can decide if they would prefer to have a reproducible bootstrap test or use a different random sample each time to assess consistency of the bootstrapping methodology. The user will answer yes (1) or no (0) to the prompted question in run_improved_*.mat to set the mode as deterministic or random. If the user answers yes to make the mode deterministic, they can decide if they would like to make their own seeds or use pre-made ones in a subsequent prompted question; the pre-made seeds allow the user to get the same bootstrapped results across different MATLAB sessions. Please note that if the user wants to use pre-made seeds, they MUST have the seeds1.mat, seeds2.mat, and seeds3.mat files in their current folder. Please also note that these seed .mat files were too large to upload on GitHub, and must be accessed through FigShare. Alternatively, the user could generate and save their own seeds and save them to the current folder when prompted, and use their own seeds across sessions for reproducibility. We have included new helper functions “makeseed.m” and “fg_bootstrap_two_groups_seed.m” to support the deterministic mode, and uploaded seed matrices which work with any of the manuscript datasets to the FigShare. In summary, the user can now either use the pre-set seed matrices (assuming the

required .mat files are in the current folder), can make and save their own seed matrices and get their own reproducible results across runs or sessions, or can get random results each time.

4) Regarding effect size, A more standard/robust approach would be to calculate the matched-pairs rank biserial correlation coefficient or Cliff's delta.

We thank the reviewer for the helpful suggestion, and we have implemented the use of Cliff's delta by calling the MATLAB function *meanEffectSize*. This required us to rerun the code in a newer version of MATLAB (R2022b), and we have updated this requirement in the GitHub Readme and the manuscript's methods (see text line 986). We have also updated all effect size values in the figure legends.

We selected Cliff's delta because it allows for the flexibility of calculating effect size for both paired and independent groups. We have updated the code so that the user can decide between paired or unpaired statistical tests. The standard in the field is to treat groups with different genotypes or genetic backgrounds as independent, even if litter-matched (mice) or age-matched (humans), but to treat comparisons of the same animal on/off drug as paired (i.e., Janz et al, 2025, Wilkinson et al, 2021, etc.). So, if the user is making a comparison within genotype (i.e., Fig. 6 PV+ data, Fig. 7 and 8, and Supp. Fig. 12), the user will respond yes to the prompted question about paired datasets in *run_improved_LFP_fit.m* and Wilcoxon signed rank tests and "paired" mode for effect size calculation will be used instead of Wilcoxon rank sum tests and "unpaired" effect size calculations. In the corresponding figures and legends, p-values have been updated to reflect the use of the signed rank test instead of the rank sum test for paired comparisons, and a description of these updated statistical methods has been added to the text (lines 988-92 and lines 997-99).

We begin with our responses to Reviewers 2/3 and to Reviewer 4, who kindly arbitrated our dispute with Reviewer 1. This is followed by a response to additional comments from Reviewer 1.

Reviewer #2 (Remarks to the Author):

Reviewer #3 (Remarks to the Author):

I appreciate the authors thoughtful responses. Though my initial queries were satisfactory, I was confused on this recent revision (though this issue may have existed in previous versions). The methods clearly state that a 60 Hz notch filter was used throughout for murine preprocessing. However, across nearly every figure in the main manuscript and supplementary figures there prominent 60 Hz features. In addition, it appears as though statistics were performed showing significance bars under 60 Hz peaks. I tried to come up with an alternative reason why this might be the case, but if the notch was performed properly it would significantly affect multiple analyses. I would appreciate if the authors could comment on this curiosity, as typically a line noise filter greatly attenuate these frequencies.

We thank the reviewer for the helpful feedback and acknowledgement of our revisions. Regarding line noise, the reviewer is correct that many notch filters completely attenuate the signal and thus make analysis of signals around 60 Hz irrelevant. Such is the case for the line noise filter in our data in Fig. 6a-h, wherein a strong 60 Hz notch filter was activated in the recording software, so we did not analyze data between 45-75 Hz. We have clarified this in the methods of the manuscript text (see lines 807-09). However, the downside of this aggressive filtering approach for head-fixed mice viewing a gray screen is that there is a narrowband signal around 60 Hz related to luminance processing (see text lines 217-19 – these are the 60 Hz features observed by the reviewer), so using such an aggressive notch filter eliminates this biological signal. Our goal was to apply a less aggressive notch filter that would sufficiently limit line noise while still preserving any underlying biological signals to statistically compare these signals across species. We applied a second-order IIR notch filter (using *filtfilt*) and carefully selected the most modest bandwidth (at the -3dB point) possible for each dataset to remove line noise without ablating the biological signals (i.e., the narrowband luminance-processing feature) around 60 Hz, which is why we still have signal to statistically compare after filtering. To illustrate, the power spectrum for an example head-fixed animal before and after our notch filtering are shown below.

Reviewer #4 (Remarks to the Author):

Overall, the authors have addressed Reviewer 1's comments appropriately. I have one additional remark regarding a specific point (see below).

Reviewer #1 comment:

Also, silencing SOM cells with NpHR3 does not produce any significant results (Fig. 6q-t), although the authors claim a reduction of peak 1a and peak1b based on Fig. 6p (inset). In this context, the results of SOM activation with Chr2 are confusing. Overall, what do the authors conclude from these experiments on different cell types?

Authors' response:

The reviewer is mistaken—the bootstrapped difference in periodic power was significantly reduced from 4.4-6.4 Hz, corresponding only to peak 1a, in NpHR3-expressing mice (see Fig. 6p and manuscript line 365). Although the p-value for the by-animal comparison of peak 1a maximum power was 0.08 following multiple comparisons corrections (Fig. 6r), the genotype difference is still strongly apparent in the separation of the datasets and uncorrected p-value of 0.02.

My comment:

The presentation of this result is somewhat confusing. In the main text, the authors claim a significant reduction in the periodic power of Pk1a, yet the figure reports a p-value of 0.08. The legend adds that the uncorrected p-value is 0.02. However, the figure citation appears incorrect: the text refers to panel 's' when it should cite panel 'r'."

We thank the reviewer for catching this unclear presentation of results and we have clarified our discussion of this result in the main text (see lines 363-71). We specify that the significant reduction in Pk1a power in NpHR3-expressing mice is observed upon analysis of the bootstrapped periodic power spectrum (Fig. 6p), elucidating the significant difference

in power observed in the bootstrapped absolute power spectrum (Fig. 6k). We also add the following sentence to further clarify: “Comparing Pk1a maximum power across genotypes revealed a strong trend supporting the bootstrapped spectrum results (Fig. 6r; uncorrected p-value = 0.02), with no apparent difference in Pk1b maximum power (Fig. 6t) nor the center frequency of either subpeak (Fig. 6q,s).”

Reviewer #1 (Remarks to the Author):

The authors responded to my previous criticisms with an almost identical manuscript, and a rebuttal letter that expresses understandable frustration but provides quite unconvincing arguments. All the major issues that I raised remain, despite the authors claim factual errors from my side.

Overall, I am persuaded that this manuscript does not make the level of Nature Communications. I still think that the paper does not provide a clear conclusion nor a specific ‘alpha’ electrophysiological signature of FXS pathophysiology that spans species. In addition, I think that there are serious concerns about signal processing, thereby hindering the interpretation of the results.

Here’s a tentative response to the authors’ rebuttal.

Narrative and conclusions:

The authors claim that they find an alpha phenotype in humans and mice. However, the EEG/LFP is strongly affected in all frequency ranges. So, in my opinion, in both humans and mice there is not an alpha phenotype, but a general EEG/LFP phenotype. Just seeing widespread changes that include alpha frequencies is not generally sufficient to declare an alpha-specific phenotype. This suggests a broader electrophysiological pattern involving multiple frequency bands. This broader spectral change may reflect a state of cortical hyperexcitability or altered network dynamics rather than a purely alpha-specific phenotype. Therefore, it is more accurate to describe such findings as an altered spectral phenotype or a disrupted oscillatory profile rather than solely an alpha phenotype.

The reviewer claims that there is no “alpha” phenotype in FXS in either species because there are changes in multiple frequency bands of the absolute power spectrum. However, the “alpha” phenotype — that is, the slowing of alpha oscillations in FXS — is already extremely well documented in humans (see manuscript lines 74-78). While there are indeed changes in power across large swathes of the absolute power spectrum in FXS, these broadband changes in *power* are captured in the aperiodic spectrum in humans and the high frequency periodic spectrum in mice (i.e., the “gamma” phenotype), and the shift

in *center frequency* of alpha oscillations is an isolated phenotype unique to this narrowband periodic signal. Our $1/f$ modelling allowed us to separate out the broadband changes from the narrowband periodic phenotype, and this approach revealed a similar slowing in a low-frequency narrow-band oscillatory feature (alpha-like oscillation) in *Fmr1* KO mice. Thus, the “alpha” phenotype spans species. In human children with FXS, alpha oscillations are disrupted both in center frequency and power, but this power phenotype is clearly a distinct phenotype from the underlying broadband increase in aperiodic spectral power because there is a *decrease* in periodic peak 1 power in FXS. Similarly, in juvenile mice, the *decrease* in periodic peak 1 power in *Fmr1* KO mice is clearly a distinct phenotype from the *increase* in broader-band high frequency periodic power and clearly does not support the reviewer’s claim of a “single broad[er] electrophysiological pattern” across multiple frequency bands.

The authors cite published literature justifying their alpha phenotype case. However, in these papers (e.g. Sohal et al., 2009; Cardin et al., 2009; Chen et al., 2017; Veit et al., 2017; Huang 2020) conclusions are drawn because specific manipulations induced effects on restricted frequency bands.

We cited this literature to support our use of opto- and chemo-genetic methods to relate to activity of GABAergic interneurons to alpha-like oscillations in mice, not to describe the “alpha” phenotype in FXS. Additionally, in this literature, the manipulations used did not always induce effects in restricted frequency bands (for example, see Fig. 2d and supp. Fig. 4 of Viet et al, 2017, wherein they employed sustained optogenetic inhibition of PV+ cells and observed a broadband increase in power which they attributed to disinhibition, just as we do in our manuscript in lines 349-55).

The authors then ask: Can the FXS “alpha” phenotype be reproduced by loss of FMRP restricted to cortical excitatory neurons and glia? Again, the evidence points towards a general effect, and not really an ‘alpha’ phenotype. A broad spectral effect resulting from FRMP deletion in pyramidal neurons and glial cells is unsurprising, as glutamatergic neurons and glial cells make the vast majority of the cell population constituting the cortical network from which they record.

We reproduced both the “alpha” and “gamma” phenotypes in the EMX1-*Fmr1* KO mice. As before, the shift in center-frequency of the alpha-like oscillations is unique to this narrowband feature, and distinct from any broader-band power phenotypes. Moreover, the *decrease* in periodic peak 1 power in EMX1-*Fmr1* KO mice is again clearly a distinct

phenotype from the *increase* in broader-band high frequency periodic power, and not just one broad, uniform spectral change in power as the reviewer falsely asserts.

They then look whether brute-force silencing of PV or SOM interneurons alters the LFP of the mouse V1. Silencing PV cells does indeed alter the LFP of V1. This observation is, however, orthogonal to the main topic of the paper. If the authors do not mean to provide a mechanism involving PV-cell role in FXS, the relationship between PV-cell activity and alpha oscillations is merely descriptive and correlative. But again, the main problem, in my opinion, is that silencing PV cells does not affect the b-peak of the alpha-frequency band of the LFP, but it affects many frequencies.

Regarding the relationship between this result and our FXS results, as we clearly outlined in our previous response: *“The results of our interneuron manipulations revealed a separable mechanistic basis for subpeaks 1a and 1b of peak 1 (the alpha-like oscillation). We are the first to report these sub-peaks, as theta oscillations in mice have always been uniformly analyzed over the 3-8 Hz frequency range. Since these subpeaks reveal different FXS phenotypes across development, separably mapping subpeak 1a and 1b onto SOM and PV+ interneurons, respectively, critically substantiated our observation of the two subpeak structure in the FXS data.”*

The reviewer now asserts that, like the “alpha” phenotype, the effect on peak 1b is not a distinct effect from the changes in other frequency bands induced by the manipulation. If there were one consistent spectral effect, the entire effect of the manipulation would be captured in the aperiodic spectrum. Instead, we also see differences in the periodic spectrum after the manipulation, with different effects in low and high frequency periodic signals. Specifically, the decrease in peak 1b power (and lack of change in peak 1a power) contrasts with the increase in high frequency periodic power, so describing these various effects as a single phenotype would undermine the wealth of spectral changes resulting from the manipulation.

Regarding SOM cells, I am not convinced that there is a statistically significant effect following optogenetic silencing. In their response, the authors state: “Although the p-value for the by-animal comparison of peak 1a maximum power was 0.08 following multiple comparisons corrections (Fig. 6r), the genotype difference is still strongly apparent in the separation of the datasets and uncorrected p-value of 0.02.” I find it confusing. Why did they use multiple comparisons to analyze Peak 1a power in two conditions? Why do they

claim significant changes from uncorrected p-values? When do they use corrected- vs. uncorrected p-values?

We made four comparisons across the different power and frequency features of periodic peak 1a and 1b in Fig. 6q-t, accounting for our application of the FDR correction. With guidance from reviewer 4, we have now improved our presentation of these results, please see our response to reviewer 4 above and the updated manuscript text lines 363-71.

Regardless of whether silencing SOM cells does indeed change the LFP power spectrum, and similarly to the case of PV cells, I really don't see how this experiment fits with a pathophysiological investigation of FXS.

Please see our response above related to the relevance of the PV+ manipulation experiments. Additionally, as we stated in our previous reviewer response: *“the experiments allowed us to study the mechanisms of resting-state alpha-like oscillations in WT mice, which to date have not been well-studied, meaning we were addressing a basic science question motivated by the FXS results. In addition to revealing the separable mechanistic basis for subpeaks 1a and 1b, the results of our interneuron manipulations suggested a role for the GABAergic system in these oscillations, motivating our Arbaclofen experiments in FXS mice.”*

Supplementary information, in the present form, is not extended data to results illustrated in the main figures. Instead, they are stand-alone findings that make the paper confusing. The authors' response is quite underwhelming, as this organization does very little to “connect[ing] worlds that are often not in sufficient dialogue with each other”. I strongly encourage the authors to de-clutter the paper, focusing on a simple straightforward message.

As we explained in the previous response, this study is data-rich, and to streamline the narrative and make the manuscript more straightforward we placed some analyses in Supplementary Figures.

Regarding the criticisms of signal processing and statistics:

Spectral resolution. In their response they state: “we used a different definition of frequency resolution based on the tapers (smoothing) used in the multitaper method, which sets the minimum resolvable distance between spectral peak”. Therefore, spectral resolution is determined by the taper bandwidth, and not by the actual (numerical) frequency resolution. The Fourier transform has bins spacing at $1/T = 0.2$ Hz. However, the

taper window (TW) = 5 so the actual resolution (W) is 1 Hz. Hence, peak separation must follow W and not 1/T. Whereas they can obtain a numerical sub-Hz resolution, this would be biologically misleading as sub-Hz values are extrapolated after multi-tapering. This makes sub-Hz precision claims biologically meaningless and thus decimals should not be compared. The separation in peaks 1a and 1b seems very aleatory (I don't see clear peaks in Fig. 3c, inset, black trace; Fig. 3j inset (both black and red trace); Fig. 4b, inset (both black and brown traces); Fig. 7f,p insets (both black and magenta traces); Fig. 8c, inset (both magenta and purple traces)).

The reviewer's critique is factually incorrect as they do not seem to understand the distinction between frequency resolution and spectral resolution. The frequency resolution, related to the Nyquist frequency, gives the bin size of 0.2 Hz, which matches our biological data as we have a sampling frequency of 1000 Hz, an epoch length of 5s, and do not use zero-padding. Spectral resolution, the minimum resolvable distance between spectral peaks, refers to how far apart two oscillatory peaks need to be within a single subject's power spectrum in order to be accurately resolved given our use of 9 Slepian tapers (time-bandwidth product (TW) = 5, spectral resolution 2 Hz) for analysis of periodic peak 1 in freely-moving murine EEG data and 4 Slepian tapers (TW = 2.5, spectral resolution 1 Hz) for analysis of periodic peak 1a and 1b in head-fixed murine LFP data. This does not mean that we cannot compare sub-Hz (decimal) center frequency values for periodic peak 1 across subjects, as the reviewer falsely asserts, but merely that we cannot distinguish between two oscillations less than 2 Hz apart (for EEG) or 1 Hz apart (for LFP) *within a single subjects' spectrum*. In other words, an oscillation can be centered at 3.2 Hz in one animal and 4.0 Hz in another animal, and this is a biologically meaningful comparison, but it would not be meaningful to compare two different oscillations within the same animal centered at these two frequencies. However, since periodic peak 1 is an isolated peak in the EEG data, and subpeaks 1a and 1b are more than 1 Hz apart from each other in the LFP data, even in juvenile animals, our data are effectively resolved within each subjects' spectrum according to our chosen spectral resolutions and meaningfully compared across subjects with sub-Hz precision according to our frequency resolution. Also, since the reviewer is unable to distinguish between the clear subpeaks for the wild-type mice in Fig. 3c, it might help the referee to imagine that they are fitting periodic peak 1 with gaussians, as is done in the SpecParam approach. It is clear that for the black trace in Fig. 3c, and indeed every WT trace in every figure, the peak could not be fit with a single gaussian (i.e., there is more than one subpeak). For some of the averages traces in juvenile *Fmr1* KO (or *EMX1-Fmr1* KO) mice, neither subpeak is particularly clear (i.e., the whole periodic peak 1 is flattened out) – but of course this is an important observation supporting our findings of reduced periodic peak 1 power in KO mice.